# Formation of linear planform chimneys controlled by preferential hydrocarbon leakage and anisotropic stresses in faulted fine-grained sediments, Offshore Angola

Sutieng Ho[1,5,7], Martin Hovland[2,7], Jean-Philippe Blouet[3,7], Andreas Wetzel[4], Patrice Imbert[5], and Daniel Carruthers[6]

[1]National Taiwan University, Department of Geosciences, P.O. Box 13-318, 106 Taipei, Taiwan
[2]University of Bergen, Center for Geobiology, Postboks 7803, N-5020 Bergen, Norway
[3]Fribourg University, Unite of Earth Sciences, Chemin du Musée 6, 1700 Fribourg, Switzerland
[4]University of Basel, Geological Institute, Bernoullistrassse 32, CH-4056 Basel, Switzerland
[5]Total-CSTJF, Avenue Larribau, Pau 64000, France
[6]Formerly Bureau of Economic Geology, University of Texas at Austin, 10100 Burnet Road, 78758 Austin, USA
[7]Fluid Venting System Research Group, Nancy 54000, France

**Correspondence:** Sutieng Ho (sutieng.ho@gmail.com, sutieng.ho@fluid-venting-system.org), Jean-Philippe Blouet (jeanphilippe.blouet@gmail.com, Jean-Philippe.Blouet@fluid-venting-system.org)

**Abstract.** A new type of gas chimneys exhibiting unconventional linear planform which are termed "Linear Chimneys" has been observed on 3D seismic data offshore Angola. Linear Chimneys are occurred parallel to adjacent faults, often within preferentially oriented tier-bound fault networks of diagenetic origin (also known as anisotropic Polygonal Faults, PFs), in salt-deformational domains. These anisotropic PFs are parallel to salt-tectonic-related structures indicating their submission to
horizontal stress perturbations generated by the latter. Only in areas with these anisotropic PF arrangements do chimneys and their associated gas-related structures, such as methane-derived authigenic carbonates and pockmarks, have linear planforms. In areas with the classic "isotropic" polygonal fault arrangements, the stress state is isotropic, gas expulsion structures of the same range of sizes exhibit circular geometry. These events indicate that chimney's linear planform is heavily influenced by stress anisotropy around faults. The initiation of polygonal faulting occurred 40 to 80 m below the present day seafloor and
predates Linear Chimneys formation. The majority of Linear Chimneys nucleated in the lower part of the PF tier below the impermeable portion of fault planes and a regional impermeable barrier within the PF tier. The existence of polygonal fault-bound traps in the lower part of the PF tier is evidenced by PF cells filled with gas. These PF gas traps restricted the leakage points of overpressured gas-charged fluids to occur along the lower portion of PFs and hence, controlling the nucleation sites of chimneys. Gas expulsion along the lower portion of PFs pre-configured the spatial organisation of chimneys. Anisotropic stress
condition surrounding tectonic and anisotropic polygonal faults coupled with impermeability of PFs determined directions of long term gas migration and linear geometries of chimneys. Methane-related carbonates that precipitated above Linear Chimneys inherited the same linear planform geometry, both structures record the timing of gas leakage and palaeo stress state, and thus can be used as a tool to reconstruct orientations of stress in sedimentary successions. This study demonstrates that overpressure hydrocarbon migration via hydrofracturing may energetically more favourable than migration along pre-existing
faults.

# 1 Introduction

Hydrocarbon migration is directly impacted by structures such as faults and salt diapirs (Roberts and Carney, 1997; Talukder, 2012; Plaza-Faverola et al., 2012, 2015). Flow directions in the subsurface and distribution of hydrocarbon leakage sites at the sea floor are pre-configured by such pre-existing structures (Thrasher et al., 1996; Moore et al., 1990). The morphology of structures formed during fluid leakage records the style and intensity of fluid expulsion and thus, is useful for deciphering the fluid-migration history (Roberts et al., 2006; Blouet et al., 2017; Imbert et al., 2017; Imbert and Ho, 2012; Ho et al., 2012b, 2018). As 3D seismic reflection data has played an increasingly important role in visualization and identification of fluid flow features (Heggland, 1997), by conducting seismic analyses for vertical successions of fluid leakage expressions around faults, such as gas chimneys feeding pockmarks and seep carbonates, it is possible to unravel the timing and pathways of migrating fluids and the sealing efficiency of faults (Ligtenberg, 2005; Plaza-Faverola et al., 2012; Ho et al., 2016).

Recent studies from the upper slope of the Lower Congo Basin have revealed the existence of a new type of chimney (Ho et al., 2016; Ho, 2013). Chimneys are usually circular in planform, however the chimneys described in this study are distinctly linear and display an extraordinary parallelism with adjacent faults in map view (Ho et al., 2013, 2016). Chimneys with non-circular planforms were first observed in the 1980's. Hovland (1983) documented chimneys on high-resolution 2D-seismic data from the North Sea exhibiting irregular and elongate planform geometries with rounded summits, and variable widths and lengths ranging between several hundred meters to more than one kilometer. They were interpreted as a result of gas escaping along fractures/faults from the apices of underlying sedimentary folds (Hovland, 1983, 1984). On modern 3D-seismic data, Hustoft et al. (2010) documented chimneys having elliptical cross-sections, and were the first to analyse the planform ratio of chimneys. Hustoft et al. (2010) suggested that the preferred orientation of the long axis of elliptical planforms of chimneys were caused by local stress perturbations associated with adjacent tectonic structures. In contrast to the chimneys described by Hovland (1983), the Linear Chimneys occurring in the Lower Congo Basin are string-like in plan-view, which vary little in width and have blunt terminations often with sharp tips as well as being rooted along and parallel to fault planes. This geometrical arrangement suggests that the near-fault stress field affected the formation of the Linear Chimneys (Ho et al., 2012a). Previously, Ho (2013) and Ho et al. (2016) used intersecting positions of Linear Chimneys and faults to determine the fault's permeability, they suggested that overpressured gas-charged fluids cannot migrate further upwards of the fault plane to produce chimneys to escape. However, the factors that determine the linear planform of these chimneys and their collective orientation have not yet been investigated.

The role of stresses in controlling orientations of venting structures, hydraulic fractures and redirecting fluid flow has been well documented (cf. Nakamura, 1977; Plaza-Faverola et al., 2015). Detailed studies of the relationship between stress state, fault orientation, tectonic structures and injectites have been carried out by Bureau (2014); who demonstrated that sand injectites preferentially intrude pre-existing polygonal faults along the extensional direction of adjacent tectonic structures. Nakamura (1977) studied interactions between orientation of magmatic fluid conduits and tectonic stresses. Nakamura established a conceptual framework relating the orientation of magmatic dykes to regional stress perturbations generated under different tectonic regimes; for instance, linear zones of eruptions occur parallel to fault lines under extensional tectonic regimes, while

zones of eruptions form at high angles with faults in compressive tectonic areas (Nakamura, 1977). Consequently, faulting and near fault stress state can play an important role on the fluids migrations hence the formation and geometric development of fluid-flow structures.

In this case study, Linear Chimneys are associated with networks of tier-bound, small densely-spaced normal faults which have a polygonal organisation in map view. Polygonal networks of discontinuities affecting discrete intervals of fine-grained sediment have previously been linked to diagenetic processes by Berkson et al. (1973). They were first identified as tiered fault systems by Henriet et al. (1982; 1991; 1988) and being investigated in detail by Verschuren (1992). They were later called polygonal fault (PF) systems by Cartwright (1994) (see Clausen et al.; 1999; Goulty, 2008), although other observations show that these faults can host a whole range of different plan form geometries including concentric patterns (cf. Stewart, 2006, Chopra and Marfurt, 2007).

Generally, Polygonal faults are considered as non-tectonic fault systems arising due to compactional-dewatering of very fine-grained sediments during the early stages of burial in passively subsiding sedimentary basins (Henriet et al., 1988). In the classic examples of these fault systems which show "polygonal" fault arrangements and also contribute to their nomenclature, they were characterised by very small differences between the horizontal principle stresses during their formation (Cartwright, 1994; Carruthers et al., 2013). The examples of polygonal faults in this case study show substantial departures from this classic "polygonal" fault pattern (so called isotropic PFs) to very polarized fault arrangements (so called anisotropic PFs) where the tier is deformed by salt tectonic structures or offset by their associated fault systems (Fig. 1; Carruthers, 2012). These faults can display a variety of intricate patterns ranging from tight radial systems around salt diapir to concentric systems within salt withdrawal basins and spiraling concentric patterns above buried pockmarks (Stewart, 2006; Ho et al., 2013). The preferentially aligned faults are many times longer than the regular faults segments with polygonal alignments but are often still confined to the same "tiers". The observations are consistent with a number of other reported examples of preferred fault alignments within networks of polygonal faults (Stewart, 2006; Ghalayini et al., 2016). The preferred fault alignments are indicative of horizontal stress anisotropy at the time of their formation (Carruthers et al., 2013).

Based on seismic observations, the objective of this study is to constrain the relative timing of fluid flow and polygonal faulting thereby offering a fluid migration model for the affected interval. This model will be used as a platform to discuss the interactions between fluid flow, faults and local stress states. Particularly the following questions are addressed: (1) why are chimneys linear in planform and not circular or elliptical as observed elsewhere? (2) why do they occur specifically along certain parts of PF planes?

## 2  Data and methods

The seismic data presented in this study extend across the outer shelf and upper slope of the Angolan continental margin (Lower Congo Basin) (Fig. 1). Two 3D seismic surveys acquired in 2006 on behalf of Total have been used for principal investigation (Appendix 1). The larger of the two surveys covers an area of 1310 km² at about 1,000 m water depth with a dominant frequency of 55-60 Hz and a vertical resolution of approximately 7 m down to about 1s TWT below seafloor. The smaller survey within

this area covers approximately 530 km² with the dominant frequency being slightly higher (70-80 Hz) allowing reach to an improved vertical resolution of 5 m. Both 3D surveys have a bin size of 6.25 x 6.25 m and a map resolution of 6.25 m. They are multi-channels, near-offset and have been post-stack time migrated and zero-phased. The data is displayed in SEG normal polarity where a downward increase in acoustic impedance is represented by wavelets of positive amplitude, as shown on the figures in red. Here, the near offset surveys are used for illustrations as they yield the highest vertical resolution and are optimal for mapping the details of small fluid venting structures. In addition, middle and far-offset volumes (representing the amplitude of the signal received at different angles of incidence) were all used for verifying the presence of studied features to rule out whether or not the studied features are seismic shadows of shallow anomalies or not. Local horizons intersected by fluid venting structures were analysed line-by-line and on arbitrary lines orthogonal to the structures to more accurately map out the linear fluid venting structures. Particularly, studied chimneys were screened for potential artefacts, combining cross-section and map views which are present on the near, middle and far offset volumes.

## 3 Geological setting

### 3.1 Regional setting

The Lower Congo Basin formed during rifting and breakup of western Gondwana followed by the opening of the central South Atlantic (Mascle and Phillips, 1972). Two main phases of sedimentation can be distinguished which broadly correspond to the rift and drift components of the basin's evolution. The rift sequence comprises extensional tilted fault blocks filled with Neocomian-Aptian siciliclastic sediments and overlain by a succession of evaporates (Séranne and Anka , 2005). The drift sequence is composed of Albian carbonates and a Late Cretaceous-Cenozoic succession of siliciclastic sediments. Since the end of evaporite deposition the passive margin sequence has been gravitationally unstable, incrementally translating seaward on Late Aptian evaporites (Duval et al., 1992). Translation was accommodated by upper slope extension and lower slope compression of the post-salt sediment cover (Séranne and Anka , 2005). The 3D seismic survey is situated above the seaward end of this zone of extension comprising an assortment of minibasins and salt diapirs. This paper focuses on the relationship between fluid flow and geological structures in the Neogene-Quaternary upper drift sequence. The principal units are summarised in Figure 2a.

### 3.2 Structural setting of the study area

#### 3.2.1 Salt-related structuration

A large seaward-dipping listric growth fault rooted in the crest of a NW-SE trending salt wall (dashed pink line on Fig. 1) divides the study area into a landward footwall domain and a seaward hanging wall domain (Fig. 2a; Ho, 2013; Ho et al., 2018). On the seaward side of the fault, the Albian to early Cenozoic strata, capped by purple Horizon 23.8 Ma, thickens into a turtle-back anticline (Fig. 2a). These thickness changes mark the first stages of salt-detached extension within the area.

Four, late Tertiary depocentres named Syncline 0, 1, 2, 3 occur along the strike of the salt wall, situated in the hangingwall

of the large listic growth fault (Fig. 1; 2a). These synclines developed during late-stage salt-detached extension in which the NW-SE trending salt wall collapsed forming the large listic growth fault which transects the survey. Synclines 0, 1 and 2 are located adjacent to two salt diapirs (D1 and D2; Fig. 1), which are rooted in the salt wall at depth. Syncline-0 subsided from the Early Miocene (c. 20 Ma) to Messinian (Ho, 2013). Synclines-1, and -2 subsided since approximately the early Middle Miocene (c. 16.4 Ma) until the Miocene-Pliocene (Ho, 2013). Some extensional faults in the SW side of Syncline-2 next to chimney structures were still active during the Quaternary (see fig. 6b in Ho et al., 2012a). The roll-over Syncline-3 in the south of the study area was induced by salt deflation during the Early Pliocene and became inactive in the Late Pliocene (Ho, 2013).

### 3.2.2   Miocene to Quaternary stratigraphy and elements

The fluid flow structures are located within the Middle Miocene to Quaternary strata which is mainly composed of hemipelagites (Philippe, 2000) intercalated with mass transport complexes (Fig. 2b). In particular, the studied chimneys primarily occur within the Upper Miocene and Pliocene deposits within synclines (Fig. 2b). These intervals are deformed by polygonal faulting which conform to two distinct tiers, named here as Tier-1 and Tier-2 (Ho et al., 2012a, 2013, 2016, 2018; Ho, 2013).

The deepest Tier 1 ranges from 70-130 m thick and contains the Late Miocene units whilst the shallower Tier 2, contains the Pliocene units and has a maximum thickness of c. 250 m. Tier-1 has a thicker pinch out toward Diapir-1 while Tier-2 shows a thinner pinch out where polygonal faults become undetectable below 60ms TWT (Fig. 3). These PFs often extend into strata above (e.g. interval A in Fig. 3a). The strata immediately overlying the PF intervals cover the relief of the horst and graben structures below and show constant thicknesses (e.g. interval B-C in Fig. 3a). Pockmarks associated with circular PF hosts can often be observed at the base of PF tiers (e.g. Fig. 3b-c; Carruthers, 2012; Ho et al., 2012a). In Tier-2, a regional imperme-able barrier of Intra-Pliocene age has been identified by its geophysical character and the vast presence of gas accumulations immediately below (Ho, 2013). The stratigraphic positions of venting structures are summarised in Figure 2b.

### 3.2.3   Organisation of PFs in the study area

In this study area, PFs are organised into different patterns in map view such as the isotropic polygonal fault pattern gradually reorganises to a system comprising of longer faults in a certain direction (i.e. referred as anisotropic PFs) with shorter faults orthogonally intersecting them. The shorter faults are the same length as the standard "polygonal" fault segments whilst the longer ones are up to 20 times longer (Carruthers, 2012). These long and short polygonal fault segments are referred to as first and second order PFs throughout this paper.

Preferred fault alignments or "anisotropic fault patterns" within polygonal fault networks have been observed in this study area. Concentric faults surround pockmarks (see fig. 2a in Ho et al., 2013) and are parallel to extensional synclinal faults (Appendix 2b; see also red dotted lines on all maps of Syncline-3 hereafter). Radial faults occur around salt diapirs (Appendix 2c) (Carruthers, 2012) whilst ladder-like fault patterns occur in the center of concentric fault patterns above Syncline-2 (Appendix 2d).

The orientations of the PFs around or above the aforementioned tectonic structures are not unusual as the fault patterns mantle

the expected stress state of the structures (Carruthers, 2012; Carruthers et al., 2013). The direction of maximum horizontal stress around the tectonic structures is indicated by the first-order anisotropic PFs while the horizontal minimum stress is indicated by the second-order anisotropic PFs (e.g. stress ellipses in Fig. 1), and hence different PF patterns are considered as indicators of stress state in the host sediments (Carruthers, 2012; Carruthers et al., 2013).

Throughout this paper we will show that stress conditions and polygonal faulting in this area has had a profound impact on the subsequent phases of fluid flow by defining a number of interim traps. Consequently, it is important to outline the nomenclature used when referring to different scales of stresses and specific parts of the fault planes in this study.

"Regional stress" refers to stress states in the sub-surface driven by the primary tectonic forces which include gravity and the lateral extension and contraction occurring above the regional salt detachment.

"Local stress" refers to stress state at the scale and within close proximity of individual tectonic structures where the regional stress field may be locally perturbed.

"In-situ stress" refers to stress conditions in-place at the location of individual polygonal faults, this is particularly relevant when trying to understand the stress conditions at sites of incipient hydraulic fracture developments which lead to the formation of chimneys.

"Lower footwall", when not specified, refers to the lower part of tilted PF blocks immediately adjacent to the fault which moved upward, or referencing the lower part of horsts in this study area.

"Lower hanging wall", when not specified refers to the lower part of PF graben.

## 4  Observations

Evidence for fluid flow around salt structures is provided by the occurrences of chimneys, pockmarks, depressions, positive high amplitude anomalies (PHAAs) which are acoustically hard (increase in acoustic impedance) interpreted as methane-derived authigenic carbonates, and negative high amplitude anomalies (NHAAs) which are acoustically soft (decrease in acoustic impedance) interpreted as free gas (Coffeen, 1978; Petersen, 2010; Plaza-Faverola et al., 2011; Ho et al., 2012a). These structures are characterised by a linear-to-circular geometry in plan view (Fig. 2b).

### 4.1  Linear Chimneys

#### 4.1.1  Acoustic properties of Linear Chimneys and terminations

Chimneys have been observed worldwide in seismic data (cf. Loseth et al., 2011; Berndt et al., 2003; Hustoft et al., 2010; Plaza-Faverola et al., 2010; Ho et al., 2016). Seismic chimneys are represented by narrow vertical zones characterized by either stacked amplitude anomalies, pull-up, push-down or distorted reflections (Heggland, 2005; Hustoft et al., 2007, 2010; 30  Petersen, 2010; Løseth et al., 2001; Loseth et al., 2011). In the study area chimneys are often associated with high-amplitude patches and shallow depressions, all of these pile up to form vertical successions (see Ho et al., 2012a). Linear Chimneys are

typically expressed as "squeezed elongate columns" of acoustic distortion zones in seismic data (Fig. 5a), in plan-view they appear as linear amplitude anomaly zones being 10s to 100s m wide and having an aspect ratio of 1:4 (Fig. 5a; Ho et al., 2012a). Linear Chimneys may terminate up- or downwards into NHAA (e.g. Fig. 5b-c), or upward into linear flame-like patterns of PHAA (see seismic section and amplitude map Fig. 6). They may also terminate upwards into linear, elongate or sub-circular shallow depressions on the modern seafloor (Fig. 7). These three elements can be combined to form 3 key variations of vertical stacking sequences (Fig. 8; see also Appendix 3):

- Type-1 Linear Chimneys terminate upwards into linear, PHAAs within depressions, which are shallow flat-bottomed with relief in the range 3-5 ms TWT (Fig. 8). The acoustic columns defining the chimneys are often associated with velocity pull-up effects.

- Type-2 Linear Chimneys terminate upwards into columns of linear NHAAs (Fig. 8). The chimney body is also characterised by push-down reflection zones.

- Type-3 Linear Chimneys terminate upwards into linear PHAAs with depressions and downwards into linear NHAA columns (Fig. 8). Linear Chimneys of this type are usually not represented by any reflection distortion zone.

The NHAA columns in Type-2 and Type-3 are situated in the lower part of the PF tier and are capped by the Intra-Pliocene regional barrier (see seismic lines in Fig. 9 and 10).

The topmost termination of a chimney is easily distinguishable when associated with pockmarks or PHAAs (cf. Heggland, 1997; Judd and Hovland, 2007; Cathles et al., 2010); whereas identifying the lower termination is challenging due to signal perturbations that increase with depths (Hustoft et al., 2007, 2009). Apart from the downward terminations of Type-3 chimney that can be clearly distinguished due to the NHAA column, the other two types are poorly constrained. Hustoft et al. (2007, 2010) suggested that the base of the chimney is marked by the disappearance of distorted seismic reflections. In this study, the lower tip of chimneys is considered to be located at the level where columns of distorted seismic reflections start to branch out in opposite directions or where distortions disappear (Fig. 9a).

### 4.1.2 Linear Chimneys and fault patterns

In the study area, Linear Chimneys mainly occur within the Pliocene PF Tier-2 (Fig. 2b; Ho, 2013) which are parallel to PFs that have preferential directions (Fig. 5a). Both elements are often parallel to adjacent tectonic faults or salt structures (Fig. 1; 9c). Although Linear Chimneys are often parallel to the first-order PFs, some do not show preferred orientations close to the NNE edge of Sycline-2 (Fig. 10a) where concentric and uni-directional PF arrays intersect. At this location PFs are more isotropically arranged (Fig. 10b). Another exception occurs above Syncline-1, where linear venting structures are parallel to the second-order PFs and the eastern edge of Syncline-1 (Fig. 11).

Few types of gas-charged fluid migration features are found within anisotropic PF networks. In the interval of PF Tier-2, in map view, a kilometric-scale PF area is filled by negative high amplitude patches in Syncline-3 (Fig. 9b), where NHAA lumps are observed to reminiscent the PF pattern. The whole NHAA area is limited laterally by the extensional fault of Sycline-3 and vertically by the Intra-Pliocene horizon, below which Linear Chimneys of Type-2 are observed (Fig. 9a).

Linear Chimneys intersect fault planes in different positions within PF Tier-2. A catalogue and a statistical analysis comprising counts of how common each intersection position, has been made by examining 209 detected chimneys (Fig. 12; see also Appendix 3; sourced from Ho, 2013; Ho et al., 2016). The Linear Chimneys intersecting PFs can be split into two main populations based on the number of their positions (Fig. 12): (1) The first population (54%) have downward terminations intersecting the lower part or basal tips of single or conjugate PFs, and rise from the lower footwall of tilted PF blocks or horsts, (2) the second population (19%) stem from (around) the intersection of pairs of conjugate PFs, and occur along the middle of the PF grabens (hanging wall).

Population (1) and (2) represent 73% of the total number of chimneys (see right column in Fig. 8 for summary). In the case of population (2), the Linear Chimneys may also intersect the lower part of the PFs, but the seismic resolution and distortion prevents an accurate determination of their position. Smaller populations include chimneys whose body intersects the middle portion of PFs footwall and hanging wall (9%); and chimneys occur in the middle of PF blocks (7%). The remaining 10% of chimneys intersect at other various positions (Fig. 12; Appendix 3). Furthermore, among the 73% (Fig. 12), 23% and 8% of the chimneys terminate downwards into negative bright spots in the PF's footwall or hanging walls, respectively; these sub-populations all belong to Type-3 Linear Chimneys. Consequently, one-third of the chimneys are associated with free gas stored in the lower part of PF blocks, while the rest only have apparent roots in the lower part of the PF tier or deeper.

### 4.1.3 Radial high-amplitude depression networks along syncline-related faults

Although most linear venting structures occur in PF Tier-2, some exceptions occur. For example, a radial network of a leakage system at a kilometer-scale was found along syncline-related extensional faults in a deeper Late-Middle Miocene interval devoid of PFs (for details see Fig. 13). This complex network is composed of interconnected linear depressions associated with PHAAs that overlie a network of big Linear Chimneys (Fig. 13a-b). These Linear Chimneys are characterised by push-downs (Fig. 13c) of which most have horizontal lengths around or in excess of a kilometre with the longest ones occurring along the strike of extensional faults (Fig. 13a).

## 5 Interpretations and discussions

The geometrical coincidence of Linear Chimney and PFs implies a relationship between both of these structures. To decipher the genetic relationships the following aspects need to be discussed: 1) the relative timing of PFs and Linear Chimney formations, 2) the gas-charged fluid migration pathways to the nucleated location of chimneys, 3) the mechanisms of preferential gas accumulation, 4) factors that control the linear planform of the chimneys.

### 5.1 Timing of polygonal faulting

Analysing timing of polygonal fault formation is essential for the discussion of whether pre-existing PFs affected fluid migration pathways, i.e. chimneys. The relationship between the timing of PFs and Linear Chimney formation can be constrained by several lines of evidence. Polygonal fault nucleation is widely considered to occur during the early stages of fine-grained sedi-

ment compaction (see Goulty, 2008). Authors like Berndt et al. (2012), Ostanin et al. (2012) and Carruthers (2012) suggested that PFs formed in shallow sub-seafloor sediments and ceased propagating when they tip out on the seafloor. Polygonal faults in the Neogene-Quaternary deposits of Lake Superior, Hatton Basin and Vøring Basin indicate that their growth is very recent and could occur to the present day seafloor (Berkson et al., 1973; Jacobs, 2006; Berndt et al., 2012; Laurent et al., 2012). Recently Sonnenberg et al. (2016) confirmed that PFs grew close to the seafloor with evidence of fault scarps filled by onlapping syn-sedimentary strata. The non-uniform topmost terminations of PFs indicate upward propagation after PF initiation (Berndt et al., 2012).

Within the study area, new evidence supports that PFs grew in sediments very close to the palaeo seafloor. Ho et al. (2013) documents that in the syn-sedimentary growth wedge of Tier-2, buried ca. 50 ms below the modern sea floor (see Fig. 3; Ho et al., 2013), PFs disappeared progressively as the tier's thickness decreased below 60 ms TWT towards the pinch-out. This means that PFs started to grow below seafloor at shallow depth: minimum 60 ms TWT (faulting during the tier deposition), or, maximum 110 ms TWT (faulting at present day). Similarly, the timing evidence of PF faulting in Tier-1 is shown by onlapping reflections on the both side of a dome underlain by a circular PF-bounded horst (interval C in Fig. 4a, see also 4b-c). Knowing that the onlapping strata are located 80 ms above Tier-1, this dates the latest activity of the PFs subsequent to Tier-1 deposition. It can be observed that the particular PFs bounding the circular horst are significantly longer than most other PFs, and propagate largely above Tier-1 (into interval A in Fig. 4a). Thus, it is likely that most PFs formed during Tier-1 deposition, and some were reactivated once after the tier was buried (below interval A) at shallow depth (15 ms below the seafloor). To conclude, based on literature and our seismic observations, the top-most boundary of both PF tiers represent the approximate timeline of when the main tier ceased to form.

## 5.2 Formation of Linear Chimneys

Seismically recorded "gas chimneys" are commonly considered to be the result of hydraulic fracturing of an impermeable interval (Pyrak-Nolte, 1996; Heggland, 2005; Loseth et al., 2011; Hustoft et al., 2007, 2010; Cevatoglu et al., 2015). Hydraulic fractures develop when pore pressure exceeds the sum of the minimum lateral stress and the tensile strength of the sediment above and propagate upwards perpendicular to the direction of the minimum lateral stress (Phillips, 1972; Cosgrove, 1995; Hustoft et al., 2010; Løseth et al., 2009; Loseth et al., 2011). Because the geological significance of chimneys has already been well discussed in many previous studies (cf. Løseth et al., 2001; Berndt et al., 2003; Hustoft et al., 2010; Plaza-Faverola et al., 2010; Ho et al., 2016), we focus here on their timing, and geometrical development in interaction with PFs.

### 5.2.1 Timing of chimney's formation related to PFs

Timing of chimney's formation is suggested to be recorded by their associated pockmarks/depressions and methane-related carbonates, which formed at chimney's topmost terminations when hydrocarbon-charged fluid reached the palaeo seafloor. Chimneys connected to pockmarks have been suggested to have formed during catastrophic blow-out events on the seafloor (Judd and Hovland, 2007; Hustoft et al., 2010). An analogue of a modern outcrop was observed when a pockmark 40 m in diameter and 7 m deep formed above a chimney while overpressured water was expulsed after 5 ½ months, from a deeper

reservoir (Loseth et al., 2011). During an experiment on CO2 injection in reservoirs, a 10 m long chimney terminating in a 4.5 m wide and 6 m deep pockmark on the seafloor developed within 48 hours at an onshore test-site in Scotland (Cevatoglu et al., 2015). These studies demonstrate that chimneys terminating into pockmarks or depressions can form within days. Similarly, PHAAs at the top of chimneys, interpreted as methane-related carbonates (Hustoft et al., 2007; Petersen, 2010; Plaza-Faverola et al., 2011; Ho et al., 2012a) usually precipitate less than a meter below the sea floor (Regnier et al., 2011), can be considered as a time marker for gas migration through chimneys to attain the palaeo-sea floor. Because PHAAs and the associated chimneys extend exactly from the linear gas accumulation below (see upward and downward terminations of Type-3 chimney in maps; Fig. 10a-b), and because the gas accumulations are compartmentalized by the anisotropic PF cells and respect the planform of PF cells (see Fig. 10c), it implies that the PF networks must have formed prior to the gas accumulations, and hence, modulate the planform development of chimneys and the subsequent fluid features.

It could be argued that the chimneys emanating from the lower part of a polygonal fault plane formed by overpressured gas expulsion at the upper tip of proto-PFs, which were still in their developmental stage. This assumption is, however, inconsistent with the fact that many chimneys are modern, and currently active as indicated by PHAAs and pockmarks at their topmost terminations on the present day seafloor (Fig. 7), while the fault planes have already fully developed since the end of the Pliocene. The nucleation point of the chimneys must therefore correspond to a level from which the fluid could not migrate further along the fault plane, and hence, it forced the gas to open a new migration path i.e. chimney.

### 5.2.2 Levels of chimney nucleation and locations of multi-layered gas reservoirs within PF tier

As the nucleation site of linear chimneys is directly linked to the site of gas accumulations, we first investigate the stratigraphic location of gas accumulations by tracing the gas migration pathway prior to the accumulations. This is done by analysing the chimney's downward terminations. Type-3 chimneys (31%) initiated within the PF tier as indicated by high negative amplitude columns at their downward termination (Fig. 10c) which are interpreted as residual gas accumulation. In contrast, the downward terminations of the major population of chimneys (Type-1) cannot be determined with precision because of signal attenuation downward. However, they still appear to root in the lower part of the tier or its base, suggesting that overpressured gas-charged fluids occurred around the lower boundary of the tier, which most probably leaked and emptied the reservoirs leaving none or only weak seismic signals. Therefore, Type-3 chimneys are interpreted as an earlier stage of Type-1, before their gas exhausted. Now we investigate how gas had migrated specifically into the lower part of the PF tier or below. Because PFs root at different of levels of depths and the presence of bright spots occurs at different strata (within or below the lower fault tier) (cf. profiles in Appendix 4), it is suggested that gas below the PF tier migrates via the long roots of PFs into different permeable layers within the tier and forms multi-layered reservoirs (Fig. 9a).

We do not rule out the possibility that gas was already present within the carrier bed before polygonal faulting. However, the seismic data clearly shows that the timing of tectonic faults and PF overlapped (Appendix 4a). In many cases tectonic faults postdate PF initiation or formation, and it has been demonstrated that tectonic faults are the main fluid migration paths for fluid into the shallow interval in the study area (Imbert et al., 2017; Ho, 2013). The traps within PF tier are small and would not take much time to charge them. Gas migration can occur quickly and the chimneys or seep carbonates record the very recent phases

(End of Pliocene) of overpressure within the tier. If hydrocarbon was present in the carrier bed prior to polygonal faulting the succession of shale above PFs would be very thin (significantly > 200m). It is unlikely that this thin succession of shale would have enough seal integrity. It is more likely that the seal formed after PFs when the overburden was thicker and more compacted.

As the exact stratigraphic levels of gas sources and migration pathways to the base of chimneys cannot be identified, based on the region in which chimneys are rooted, we propose the following scenarios when gas migrated upwards from deeper sources: (1) Gas was trapped in strata along sealed tectonic faults below the PF tier, (2) gas migrated laterally and reached carrier beds immediately below the PF tier and intersected by long PFs then accumulated there (Fig. 14a, Appendix 4), or (3) gas migrated along the lower portion of the PFs to reach permeable layers inside the lower tier (Fig. 14b). These three processes either

happened solely or in combination with each other as a series of steps.

In conclusion, the rooting position of the majority of chimneys suggests that, before the chimneys nucleation, gas migrated to and accumulated preferentially in the lower part or at the base of the PF tier.

### 5.2.3   Mechanism of gas trapping in the lower part of the PF tier

As supported by the statistical analysis presented herein, over 54% of chimneys stem from the region around the lower PF

footwall, therefore, we infer that over 54% of the time gas accumulated in the footwall at the base of chimneys. It is also the same for the 19% of chimneys that stem from the lower PF grabens (hanging wall). As a result 73% of the total time gas preferentially accumulated in the lower part of PF blocks, so we investigate the cause of this phenomenon. We suggest that two hypotheses in combination account for the mechanism of preferential gas accumulation in the lower PF footwalls of tilted blocks, horsts and lower hanging walls/grabens: (a) the presence of an impermeable regional seal and (b) impermeable portion

of fault plane); while two other hypotheses determine together the preferential gas migration to the lower PF footwall: (c) the differential strain in fault blocks, and (d) the stratigraphic position of permeable layers in fault blocks; finally one hypothesis for graben hanging wall: (e) the increase of local permeability.

a) Impermeable barrier. The seismic record documents that gas is present in the lower part of PF Tier-2 over a vast area, below the regional impermeable, Intra-Pliocene barrier (Ho, 2013). The Intra-Pliocene barrier corresponds to the topmost boundary

of free gas accumulations, and does not parallel the seafloor (blue dotted line in Fig. 9a). As a result, this impermeable barrier does not likely represent bottom simulating reflectors (BSR) and is, hence, interpreted as of purely depositional origin.

b) Fault seal. Persistent occurrences of gas accumulations in the lower part of the PF tier below impermeable barrier, regardless of faulting offsetting it, likely indicate that the lower portion of PF plane is not hydraulically communicated with the upper one. Otherwise gas would use the upper fault plane to migrate further to the upper fault tier (Ho et al., 2016; Ho, 2013). Therefore,

the upper portion of PF fault plane above the regional impermeable barrier is likely impermeable at least during gas migrations (Ho et al., 2016), and the downward limits of the impermeable fault zones are possibly non-uniform and can vary or extend beneath the Intra-Pliocene barrier. This hypothesis is well demonstrated, for example, by the vast distribution of gas-filled PF blocks below the Intra-Pliocene barrier in Sycline-3 (Fig. 9a).

It can be argued that, sediments in the lower PF tier are more permeable and lead gas preferentially accumulate in such place.

This possibility is disregarded because of the similarity between the lithologies in the upper and lower part of the PF tier as indicated by Total's internal well reports, regardless the permeability measurement of the host sediments is unavailable.

c) Differential strain. Shear strain resulting from extension and normal faulting affects the hydraulic properties of rocks adjacent to the fault surface (Barnett et al., 1987). Extensional faulting induces significant shear strain and dilatancy (Zhang et al., 2009) which consequently enhances the porosity and permeability of the wall rocks in shallow buried depths (Barnett et al., 1987). Numerical modeling demonstrates that the lowest shear stresses occur in the footwall block near the basal tip of a normal fault and that the greatest shear stresses occur in the upper part of hanging wall blocks (Fig. 15a; Zhang et al., 2009; Welch et al., 2009). These results match the conceptual model of Barnett et al. (1987) for unlithified shallow buried sediments, which shows that the lower parts of footwalls and the upper part of hanging walls are in a state of compressional strain, compared to the top of the footwall and base of the hanging wall in shallow depths (Fig. 15b). As Tier-2 was buried only a few tens of meters when PFs formed, it was very likely not lithified and the lower part of the footwall blocks could have experienced dilatation (Barnett et al., 1987). Therefore, the highest permeabilities would be expected to occur in the footwall of a normal fault near the basal fault tip where gas accumulation is expected to occur. In fact, the majority of Linear Chimneys emanate from the lower parts of the footwall, where gas columns (NHAA) are observed (Fig. 10c).

d) Stratigraphy high. Another explanation for the preferential accumulation of gas in the footwall blocks of the faults is purely geometric: with normal faults, the footwall block is upthrown with respect to the hanging wall, and its series usually raise or tilt upward along the fault (Fig. 15c). As a result, upward migration of gas tends to fill the footwall side of the faults.

e) Fractures increase premeability. For the second major population of chimneys (19%) that stemmed from the middle of grabens (PF hanging wall), it is likely that the outbreak point of overpressured fluid was located in the lower part of the graben. In the hanging wall, deposits are likely under a compressional regime (Barnett et al., 1987; Welch et al., 2009). Thus, gas will not preferentially migrate into such a location, however a controlling factor is needed to guide the direction of gas migration: fracturing in the bottom of graben leads to an increase in permeability facilitating the trapping of gas (Fig. 14b-ii; Ho et al., 2016). This phenomenon happens when graben sediment moves downward along curved, steepening upward faults during extensional faulting (Cloos, 1868; Fossen and Rørnes, 1996; Bose and Mitra, 2010). Alternatively, the lower parts of the graben are subjected to compression where a compressional fold forms a structural trap.

The combination of the above elements is suggested to induce the formation of PF fault-bound traps in the lower part of a PF tier.

### 5.2.4   Nucleation of Linear Chimneys

A conceptual model for the formation of Linear Chimneys is proposed below. The majority of Linear Chimneys stem along the surface of the lower PF footwalls at various positions (Fig. 12), suggesting gas-charged fluids could not migrate along the upper portion of PFs while impermeable (at the moment when chimneys formed). The permeability of small faults in fine-grained marine sediments varies upon the changes in stress and resultant strain around faults (cf. laboratory experience of Kaproth et al., 2016), which can likely explain the impermeability along the upper part of PFs. In literature, numerical models of Nunn (2003) shows that fluid pressure might not be high enough to maintain low effective stress in the upper fault zones. Therefore,

the upper part of the fault remains closed. Other modeling results show that it is possible for the lower part of PFs to appear permeable and critically stressed in the contemporary stress field while the upper parts are neither permeable nor critically stressed (Wiprut and Zoback, 2000; Zoback, 2007). In the anisotropic stress area (salt tectonic area), stress generated by the overpressured fluid in host rocks leading to propagation of planar fractures in PF hanging walls, this likely indicates that fluid

pressure was not high enough to open the upper fault plane, but only high enough to overcome the minimum horizontal stress plus the fracture strength of the fault blocks (Delaney et al., 1986; Kattenhorn et al., 2000). Therefore, once the gas trapped in the lower part of the footwalls became overpressured (Fig. 16a-b), hydraulic fractures propagate from the footwall to pierce the overlying strata and breach the impermeable barrier, and as a result, the chimneys were initiated and originated along the lower part of polygonal fault planes.

We would like to emphasise that apart from overpressured fluid (gas) creating new fractures, overpressured gas may also pass through, filling pre-existing sub-vertical cracks/fractures in the hanging wall bottoms along the main fault surface (fig. 2 in Gaffney et al., 2007). Pre-existing vertical fractures occur in hanging wall orginated from movements of normal faults have previously been demonstrated by analogue models of van Gent et al. (2010). Fluids may open and extend pre-existing sub-vertical cracks/fractures in the hanging wall only if the pressure required for the fluid entering into the hanging wall fractures

was less than the one for creating a new fracture (Gaffney et al., 2007). Pore pressures in the PF bound traps decrease after the fractures propagate or extend, and the residual gas in the traps may re-equilibrate with lithostatic pressure (Zoback, 2007). Consequently, some free gas can remain in the lower part of the PFs at the downward termination of Linear Chimneys (Fig. 10c).

For chimneys originating within the lower part of PF grabens, gas might be compartmentalized in the damaged graben by

the impermeable portion of the PF which was likely extended downward beneath the Intra-Pliocene barrier; therefore, gas was not able to flow into the adjacent horsts (Fig. 14b-ii). Consequently, hydraulic fractures initiated in the graben centre and propagated upward along the central axis (Fig. 16c).

For chimneys that do not intersect with any fault i.e. occur in the middle of PF fault blocks, the illustrated model by Løseth et al. (2009) can be used as a referential analogue (see fig. 21 in Løseth et al., 2009); a lateral contact point between the edge of

the gas accumulation and the upper limit of the tilted storage-layer, in the middle of the tilted block, formed a hydrocarbon spill point from where gas chimney nucleated and propagated upward (Løseth et al., 2009). This type of spill point is commonly occurred in structural traps.

### 5.2.5   Results of chimney's linear planform geometry relating to fault orientation

The linear planform of chimneys and their evident spatial relationship to anisotropic polygonal faults suggest that gas migration

and hydraulic fracture propagation are controlled by the alignments of anisotropic PFs. Anisotropic PFs follow the orientations of salt tectonic structures indicating that the PFs are heavily influenced by the stress states resulting from salt activities (Carruthers, 2012). The presence of faults can perturb the surrounding stress field and affect the adjacent fracture propagation (Rawnsley et al., 1992; Kattenhorn et al., 2000). Thus, degree of horizontal stress anisotropy and dominant direction of horizontal intermediate stress play a determinant role in both formation and geometry of anisotropic PFs, and hence planforms of

chimneys.

In Syncline-0 polygonal faults are absent, yet the kilometric-scale Linear Chimneys are still present (Fig. 13a). Here, Linear Chimneys are parallel to deep-seated tectonic faults resulting from salt movement, the horizontal stresses are not equal as the intermediate principal stress exceeds the minimum one (Cosgrove, 1995). The gas pressure was likely not strong enough to overcome the intermediate stress so the hydraulic fractures opened in parallel with it, and against the direction of the minimum horizontal stress (Cosgrove, 1995). As a result, the final chimneys are linear in planform and follow the strike of adjacent faults. This example clearly demonstrates that chimneys propagate towards the direction that resulted from the perturbation of horizontal anisotropic stresses induced by the tectonic faults (cf. Nakamura, 1977).

In the smaller scale of polygonal faulted blocks, Linear Chimneys and anisotropic PFs are often aligned, such as in Synclines-2 and 3 (Fig. 10a; 9c; 5). However, in a particular location above the ridge of Syncline-2, Linear Chimneys are aligned with a pseudo-isotropic (less-anisotropic) PF network enclosed in a zone between two (strong) anisotropic PF patterns, one is parallel to the edge of Syncline-2 and the other has a "ladder"-like pattern in the center of Syncline-2 (Fig. 10a-b). In this specific location although the PF pattern is similar to isotropic polygonal faulted areas but the stress magnitude remains greater because of the tectonic extension (Carruthers, 2012). Given that in such an enclosed pseudo-isotropic PF area, chimneys are still linear and all aligned parallel to their rooted PF and do not show strong preferred orientation (Fig. 10a); this particular example leads us to conclude that at tier-fault scale, the in-situ anisotropic stress of the nearest PFs has major influence on the orientation of Linear Chimneys than the local tectonic fault stress field. Nevertheless, as the majority of Linear Chimneys are aligned parallel to both tectonic and polygonal faults, as Linear Chimneys do not occur in areas where classical isotropic PFs are solely present, therefore the combination of both anisotropic stress fields of tectonic and polygonal faults is suggested to be the main cause of linear planform chimneys with preferential orientations.

Finally, the lateral propagations of the kilometric-scale Linear Chimneys rarely impeded by faults are oriented roughly parallel to them and the chimneys can reach much greater lengths (Fig. 13). In contrast, chimneys within polygonally faulted areas are much shorter horizontally (> 300m) (Appendix 3). This is because the distance for which hydraulic fractures can propagate laterally along a specific trajectory is limited by faults.

In conclusion, the examples above demonstrate that 1) when tectonic faults are solely presented (without PFs) the planform and orientation of chimneys are affected only by the stress field of tectonic faults; 2) while in areas where PFs occur, tectonic stress controls the orientation of anisotropic PFs, and the in-situ stress of the PFs controls the orientation of Linear Chimneys.

### 5.2.6 Model of fluid migration and Linear Chimney's formation

Linear Chimney formation can be summarised in 6 steps (Fig. 17).

1. During the Pliocene, anisotropic PFs formed and developed under the influence of an anisotropic stress field induced by adjacent (salt-) tectonic structures.

2. Gas-charged fluids migrated vertically from deeper intervals along tectonic faults, and laterally into the permeable beds below or at the base of the PF tier (Fig. 17a).

3. Gas-charged fluids migrated upwards along the root of PFs, then flowed into the lower part of the tier, and filled the high-

est permeable layers in the horst or the fractured apex of grabens where the permeability was higher than in the undamaged sediment (Fig. 17a-b). The pressure of gas-charged fluid was not strong enough to allow gas to intrude the upper part of PF plane (which is referred as impermeable). Further upward migration of the gas-charged fluids within strata was prevented by the Intra-Pliocene impermeable interval.

4. Overpressure of gas-charged fluids attained the threshold value for hydraulic fracture propagation but was insufficient to reactivate the fault.

5. Hydraulic fractures (i.e. chimneys) propagated upward from the lower part of the PF footwall or hanging wall (Fig. 17c), throughout the end of the Pliocene to the Quaternary. These fractures were affected by the stress field around the closest fault and developed a linear planform parallel to adjacent faults (along the direction of the intermediate principal stress).

6. The linear outlet of chimneys on the seafloor was eroded by gas venting, producing a linear depression in which methane-derived authigenic carbonates precipitated and are expressed by PHAAs in seismic data (Fig. 17d).

## 5.3   Implications for petroleum exploration

### 5.3.1   Reconstruction of hydrocarbon leakage history by using Linear Chimneys

This analysis of Linear Chimneys has revealed information about palaeo activities of buried hydrocarbon systems, especially how gas-charged fluid interacted with pre-existing geological structures while migrating upward to the subsurface. Based on the analysis of linear venting structures, we attempt to reconstruct the hydrocarbon leakage regime in this study area. Linear venting structures and gas concentrations occur predominantly in the synclines indicating they are sites of active fluid flow (Fig. 9b; 10b). The reason why gas preferentially concentrates within syncline in the Pliocene PF interval in this study area

may because of coarser grained sediments trapped in the syncline depocenters during that period. It is also known that synclinal faults cut down to deep turbidite channel reservoirs in this study area (Monnier et al., 2014). Venting structures occurring around the extensional faults of synclines suggest that these faults served as initial leakage pathways for gas-charged fluids to migrate upwards into Tier-2. If the amount of gas exceeds the accommodation volume of the faults, gas will migrate horizontally into shallow carrier beds at the base of PF Tier-2 and then use the deep-rooted PFs as further leakage pathways into the PF tier (Fig.

18a). This explains why gas accumulations occur within PF Tier-2 above the center of Syncline-3, mimicking the geometry of the polygonal cells (traps) (Fig. 9b). Within the anisotropic PF network in all syncline location (e.g. Fig. 10b; 9c), the preferential orientation of linear gas accumulation and hydraulic fractures (i.e. Linear Chimneys) suggests that the direction of gas flowing and escaping within the tier was likely guided by anisotropic stress condition.

In contrast, where anisotropic PFs are absent no Linear Chimneys occur. Therefore, gas migrations are likely unaffected by the

surrounding stress state because the horizontal principle stresses are too weak or too similar and instead gas may migrate in random directions, until they reach a permeable bed or mechanically weak zone to break through (Fig. 18b).

To summerise, the direction of fluid leakage in areas of anisotropic PFs can be predicted by analysing fracture and fault directions (Ho, 2013; Ho et al., 2013).

### 5.3.2 Reconstruction of palaeo stress directions

*Linear chimneys as stress indicators*

We have shown that the propagation and resulting morphology of chimneys are receptive to perturbations in magnitude, directions and differences of the horizontal principle stresses. The ability to date the formation of such systems makes Linear Chimneys potential indicators of palaeostress conditions. Normal faults propagate parallel to the intermediate principal stress, while hydraulic fractures also open in parallel to the direction of intermediate principal stresses and against the minimum principal stress (Cosgrove, 1995). For example, in Syncline-1, the orientation of the first-order PFs implies that the direction of the intermediate compressive stress during PF's formation was initially following the curvature of the northern edge of Syncline-1 (Fig. 11). However, the subsequently formed Linear Chimneys are rather paralleling to the curvature of the eastern edge. Because hydraulic fractures open in parallel to the intermediate principal stress and as their alignments also indicate the direction of intermediate stress, therefore, at the moment overpressured gas-charged fluids escaped via hydraulic fractures (i.e. Linear Chimneys formed), the intermediate stress direction was likely switched from the northern curvature to the eastern curvature. Thus, the horizontal stress field re-oriented during gas leakage after PF formations. Next to the NE side of Syncline-1 an extensional fault set that is observed to parallel the Linear Chimneys (Fig. 11) was re-activated during Plio-Quaternary (red stars in Fig. 2a; Ho, 2013). Because these tectonic faults were active during the same time as the linear conduits formed (in Pliocene to the beginning of Quaternary), it is plausible that the re-orientation of the stress fields in Sycline-1 resulted from the movement of these faults. In conclusion, comparing the direction of first-order PFs and the direction of Linear Chimneys is useful for diagnosing the evolutional history of stress fields in the past.

*Lineaire PHAAs as stress indicators*

We have shown that the plantform of chimneys was modulated by the stress field of faults, and that the kilometer-scale Linear Chimneys are parallel to the tectonic faults in Syncline-0 (Fig. 13a); these chimneys with their lateral tips connect to each other and constitute a complex Linear Chimney network at 9 Ma. Their top is marked by a radial-depressional network formed due to further leakage. Methane-related authigenic carbonates that precipitated within the depressional network formed another complex PHAA network and highlighted the radial geometry of underlying chimney network (Fig. 13b). Therefore, the subsequent flow structures associated with the chimneys that have the same planform also appear to be useful to determine the palaeo principal stress directions.

## 6 Conclusions

The anisotropic stress attributed to perturbations of the regional stress field by faults and salt diapirism, controls the orientation of PFs, which in turn impacts gas-charged fluid accumulation, migration, leakage pathways and ultimately the geometry of gas leakage conduits and associated expulsion features at the seafloor. The mechanism of Linear Chimney formation is summarised as follows:

1) Fluid expulsion features making the upper termination of chimneys at the palaeo sea floor (pockmarks, depressions and seep

carbonates) date chimney formation from the End Pliocene to the Present. Polygonal faulting initiated in the shallow depth range from 50 to 100 ms TWT below the seafloor during Early Pliocene pre-date Linear Chimneys.

2) PF blocks form fault-bound gas traps in the lower part of PF tiers.

3) The location of these traps determines the site of gas leakage and hence, the nucleation site for vertical chimneys.

4) Linear Chimneys nucleating along the lower part of polygonal fault planes document that gas-charged fluids did not migrate along the upper portion of PF planes, which, therefore, appear to be impermeable.

5) The linear planform of chimneys is mainly determined by the orientation of the intermediate principal stress around the closest fault. Overpressured gas-charged fluids break through the host rock by pushing aside the host rock towards the direction of minimum principal stress, consequently Linear Chimneys developed aligned and parallel to the intermediate principal stress, and hence tectonic and/or polygonal fault strike.

6) In isotropic stress fields, under the same spectrum of fluid expulsion dynamics, the morphologies of chimneys and associated fluid expulsion features at the sea floor (depressions, pockmarks, seep carbonates bodies) are circular, while they are linear in anisotropic stress fields surrounding tectonic faults, salt structures and in anisotropic PF networks.

7) In-situ stress fields of isotropic PFs alone are not sufficient to induce Linear Chimneys, anisotropic tectonic stress fields must be involved.

8) In areas experiencing a transitional of two stress fields, Linear Chimneys follow the trend of less-anisotropic PFs rather than the nearby tectonic structures. Therefore, the development of Linear Chimneys is interpreted to have been predominantly affected by the in-situ stress field of anisotropic PFs (which are dominated by the anisotropic tectonic stress).

9) Linear Chimneys can be used as a tool to reconstruct previous stress directions in the same way as using preferential orientated PFs.

*Competing interests.* No competing interest

*Acknowledgements.* We thank Total S.A. for providing data, funding and its partners for publication permission, and the Ministry of Science and Technology of Taiwan for the grant MOST1052914I002069A1. Our work is based on and extended from Chapter 6 of S. Ho's PhD. The scientific work was fully carried out in Total-S.A.-France and completed under its direction. S Ho thanks Benoit Paternoster for his supervision on Geophysics since 2007. S. Ho thanks J.A. Cartwright for his great interest in this work, Mads Huuse for reviewing S. Ho's PhD, Cardiff University for the partial PhD funding. Thanks for the uncountable supports/advice from Timothy Byrne, David Hutchings, Quentin Vannelle, Ludvig Löwemark and Char-Shine Liu. A special thanks goes to Sebastian Czarnota Konrad for the English proof reading service. We thanks A. Plaza-Faverola and two reviewers for their helpful comments. This work had been previously submitted to Marine-Geo1ogy on 23rd January 2017 and retrieved by us in 2018 due to extensive delay in the revision process.

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

# 5.3Ma horizon dip map

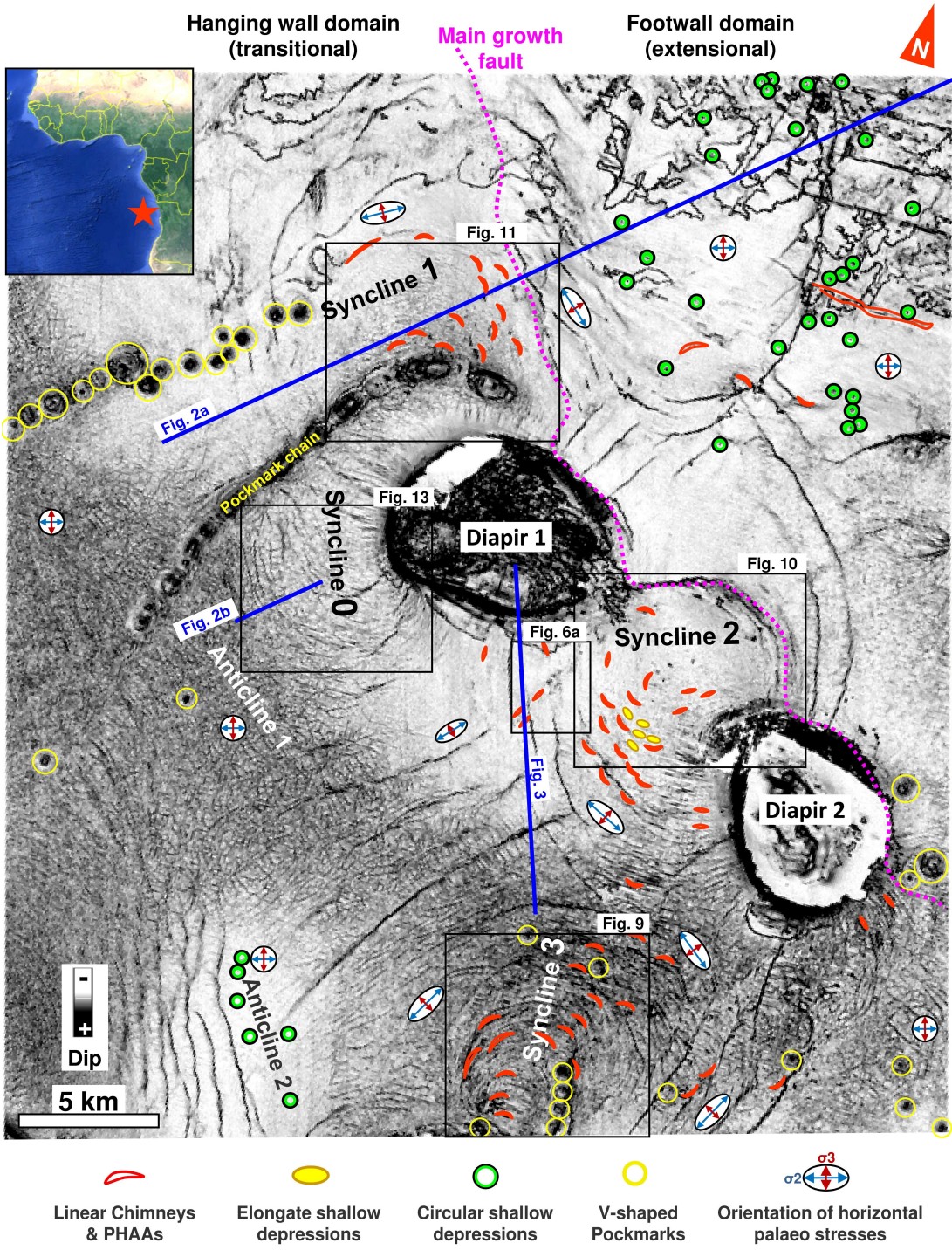

**Hanging wall domain (transitional)**     **Main growth fault**     **Footwall domain (extensional)**

Fig. 11

Syncline 1

Fig. 2a

Pockmark chain

Fig. 13

Syncline 0

Fig. 2b

Anticline 1

Diapir 1

Fig. 10

Fig. 6a

Syncline 2

Fig. 3

Diapir 2

Fig. 9

Syncline 3

Anticline 2

Dip

5 km

| | | | | |
|---|---|---|---|---|
| Linear Chimneys & PHAAs | Elongate shallow depressions | Circular shallow depressions | V-shaped Pockmarks | Orientation of horizontal palaeo stresses |

σ3

σ2

**Figure 1.** Dip map of referential horizon 5.3 Ma showing distributions of fluid expulsion structures across the study area. Sub-types of fluid expulsion features are shown diagrammatically (see legend below main figure). Palaeo-stress ellipses show relative directions and magnitudes of the horizontal principal stresses and are constructed from the planform geometry of the polygonal fault networks (see section 3.2.3 for more information). Blue axis and red axes on stress ellipses indicate the palaeo-orientation of the intermediate and minimum stresses, respectively. Location of seismic survey is indicated by a red star on insert map.

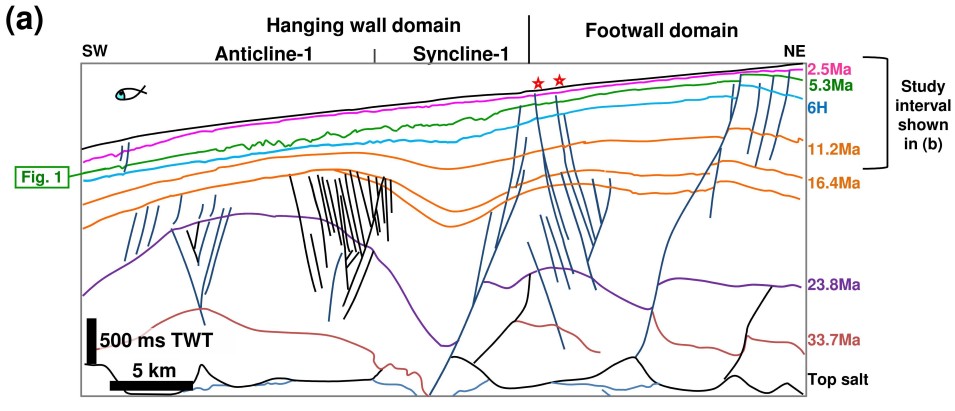

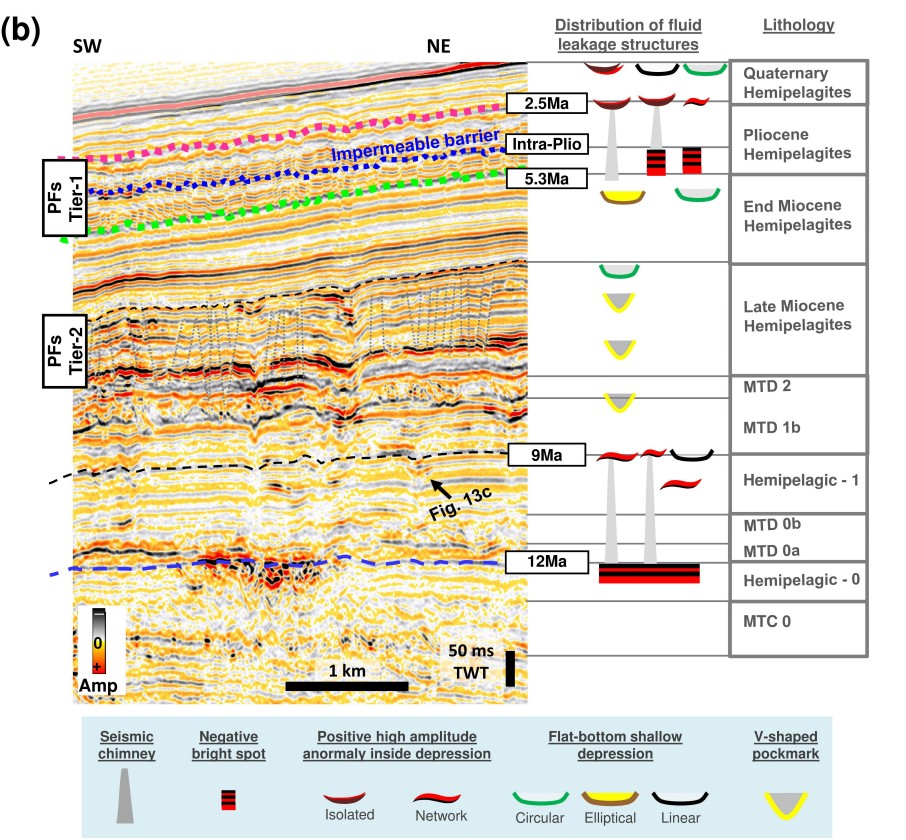

**Figure 2.** Geological setting. a) Interpreted cross-section through the study area. See Figure 1 for location. Red stars indicate faults that were reactivated during the Quaternary. b) Types of fluid expulsion structures and their distributions within the stratigraphic framework are shown to the right of seismic section. Horizons pre-fixed "H" and their approximate ages are shown. Figure adapted from Ho et al. (2012a) and Ho (2013).

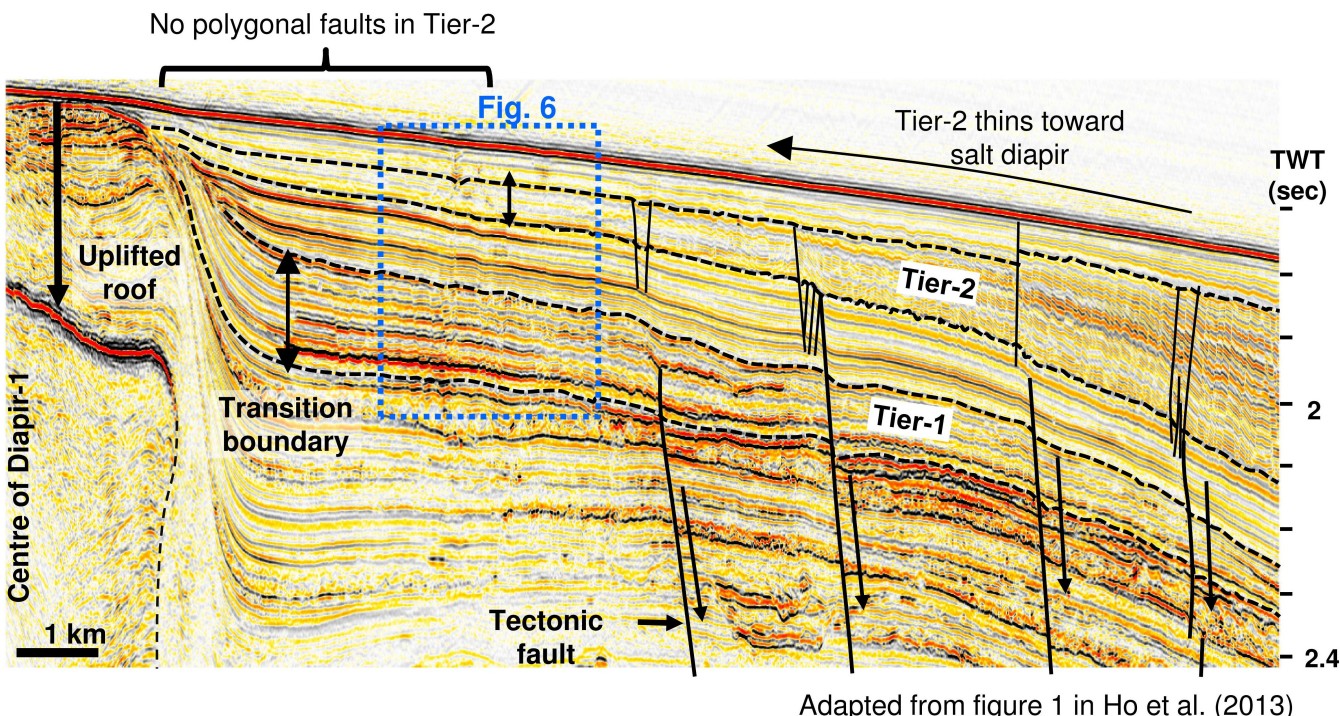

**Figure 3.** Arbitrary seismic line showing the pinch-out of polygonal fault tiers (defined by black dotted lines) against the SE flank of Diapir-1. Line location is shown on Figure 1. The polygonal faults disappear beyond the black vertical arrows, progressively towards the pinch-out of the tier at the transitional boundary where the wedge thickness starts to be less than 60 ms. Note that PFs are absent toward the pinch-out of Tier-2, but are present at the same location in Tier-1 below, where this tier reaches its maximum thickness (Ho et al., 2013). This may provide additional support for the theory of minimum thickness determining PF growth (Carruthers, 2012). This observation can serve as a reference example on PF growth. Image adapted from Ho et al.(2013).

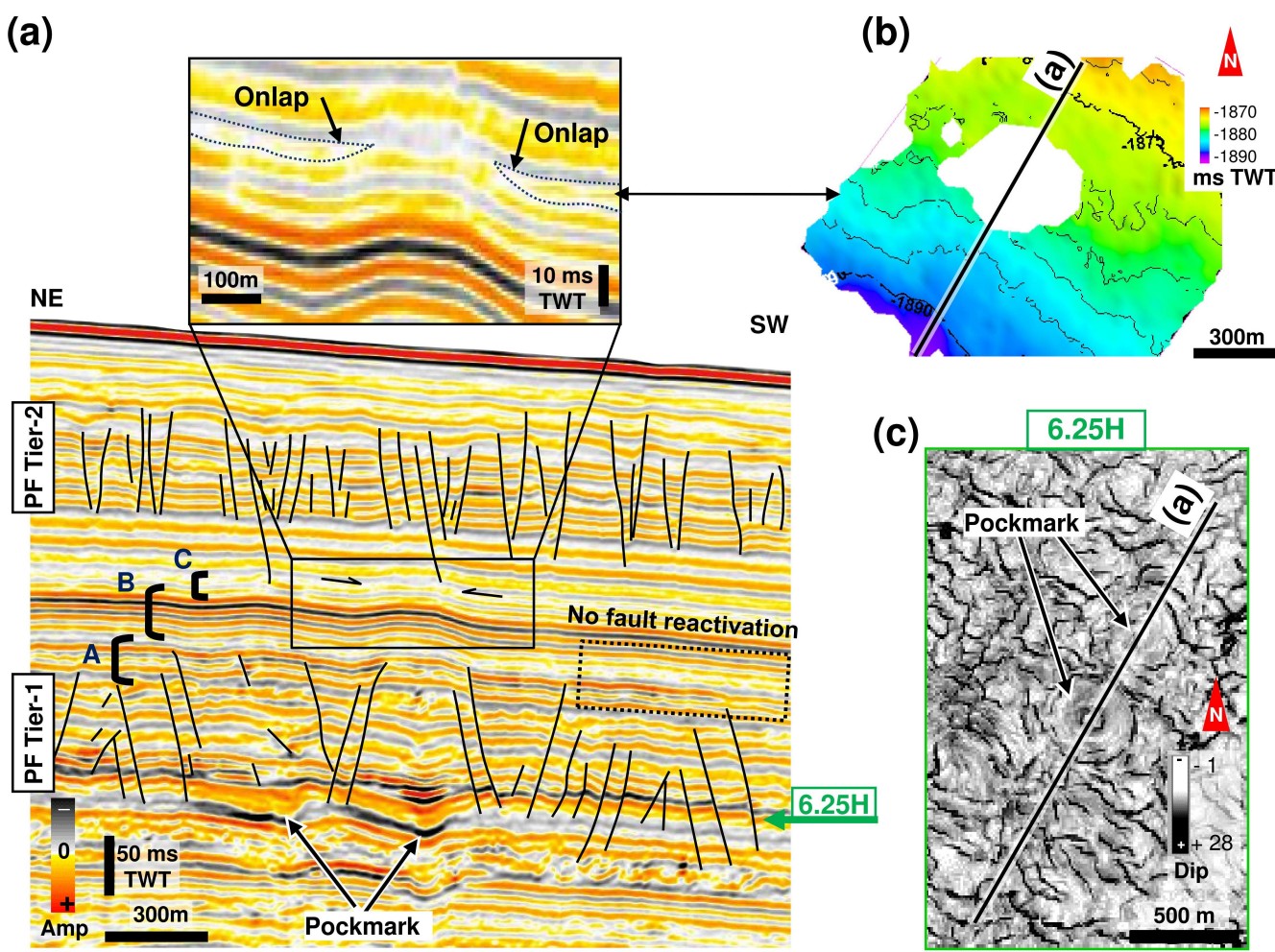

**Figure 4.** Direct evidence for the time when PFs activity ceased. a) Profile transecting outward-facing concentric faults (defining a horst block) above a buried pockmark (annotated) in PF Tier-1. A minority of fault's upper tips exist in interval A above Tier-1 (interval A); the first non-faulted strata (interval B) is folded above the horst with an isopachous thickness; the topmost reflection (in interval C) pinches-out on the positive relief above the horst as indicated by black arrows. b) Two-way time map of the onlap termination of horizon (in interval C) against the horst. c) A dip map at the base of Tier-1 showing concentric PF around the circular pockmark shown in (a).

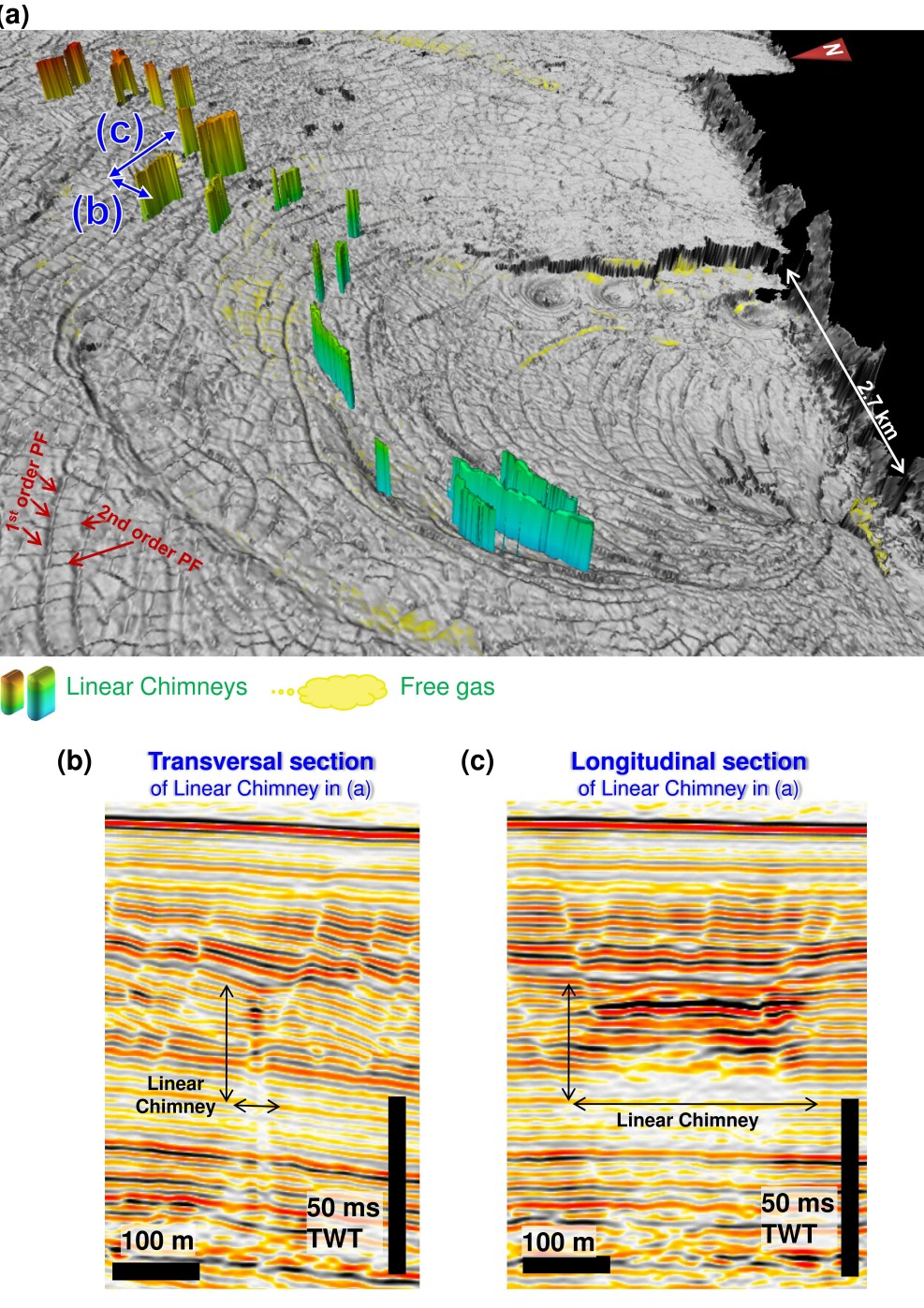

**Figure 5.** Morphology of Linear Chimneys. a) 3D visualisation of Linear Chimneys in Syncline-3 parallel to preferentially and concentrically orientated polygonal faults parallel to syncline edge. The 3D chimney bodies are issued from the 3D mapping of the high amplitude anomaly columns on seismic sections. b) and c) Transverse and longitudinal seismic sections through Linear Chimney in (a).

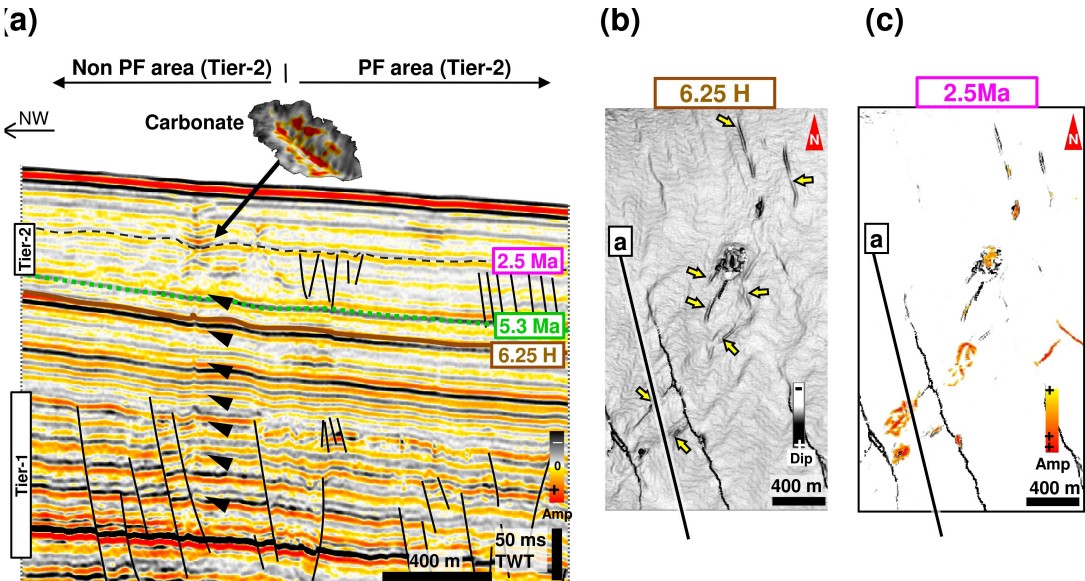

**Figure 6.** Group of Linear Chimneys with flame-liked PHAAs at upward terminations. a) Seismic section shows the Linear Chimneys emanating from Tier-1 and terminating upward in the upper Pliocene units. This section is a close-up of the thinning wedge studied in Ho et al. (2013). The full profile is shown in Figure 3. Line location is shown in (b) and (c). b) Dip map of Horizon 6.25H showing the linear planform of chimneys (yellow arrows). c) Isolated, positive high amplitude anomalies at the top boundary of Tier-2. They occur at the topmost termination of Linear Chimneys and are interpreted as methane-related carbonates.

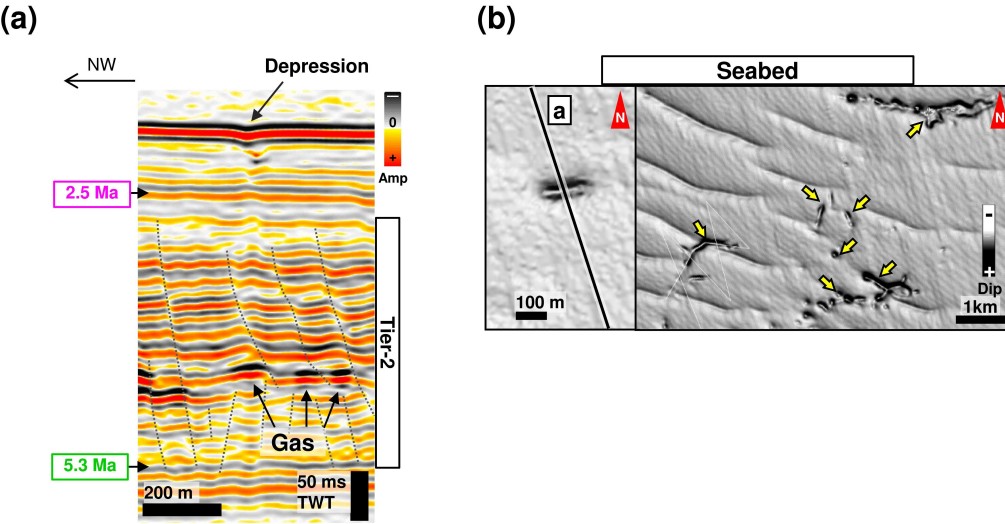

**Figure 7.** Dip map showing linear depressions at the present day seafloor. a) Seismic section shown on map (b) is across one depression, which occurs at the topmost termination of a Linear Chimneys emanating from Tier-1. b) The linear depressions at issue are indicated by yellow arrows and locally interfere with regularly spaced furrows of likely sedimentary origin.

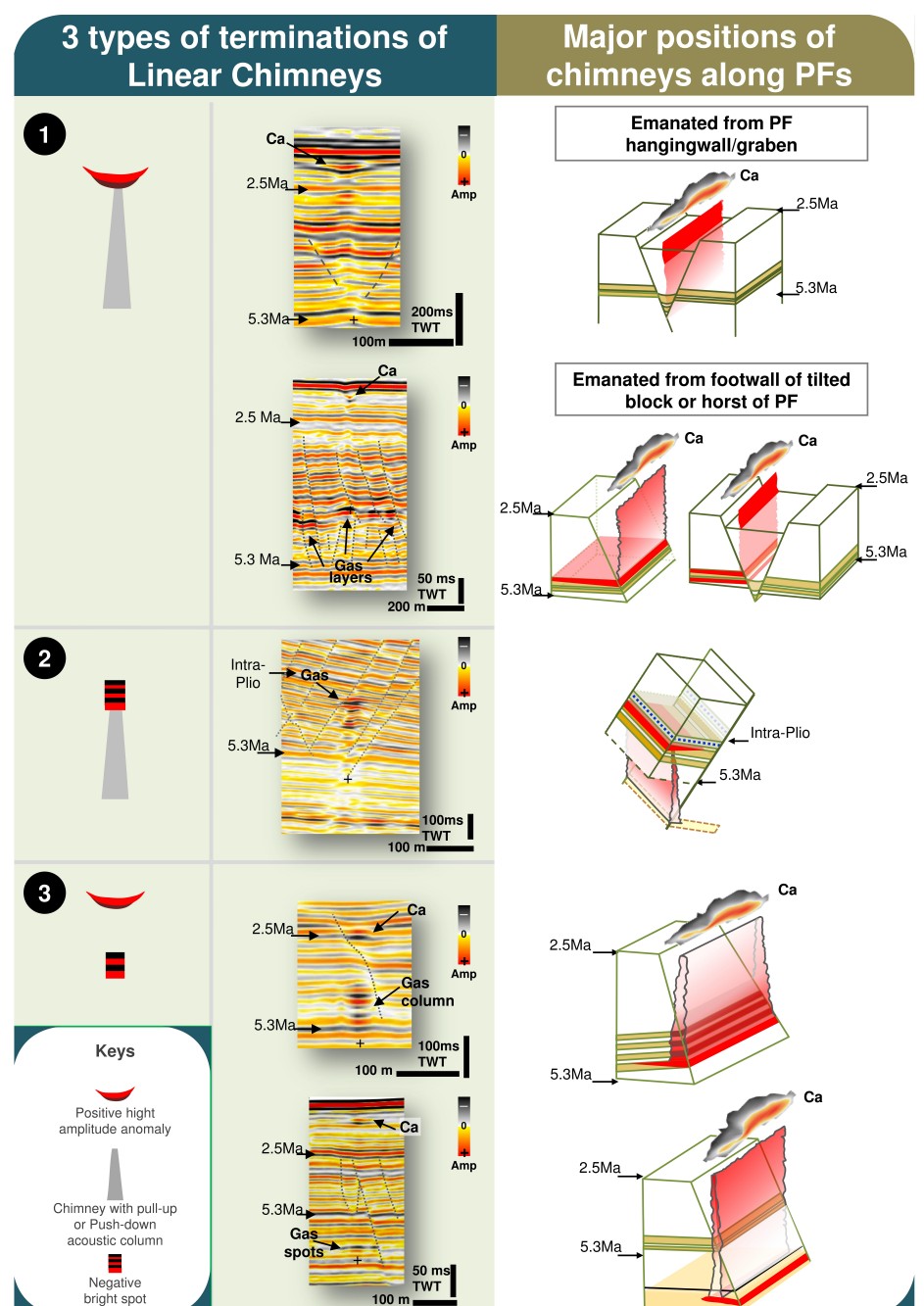

**Figure 8.** Classification scheme showing types of terminations of Linear Chimneys. Left column: shows symbols which represent the three groups of Linear Chimneys with different up- and downward terminations. Middle column: shows the seismic images of the chimneys represented by the symbols. The apparent bases of chimneys are marked with crosses. Right column: shows the 3D interpretation of the major intersecting positions between the polygonal faults and the Linear Chimneys which shown in the adjacent seismic images.

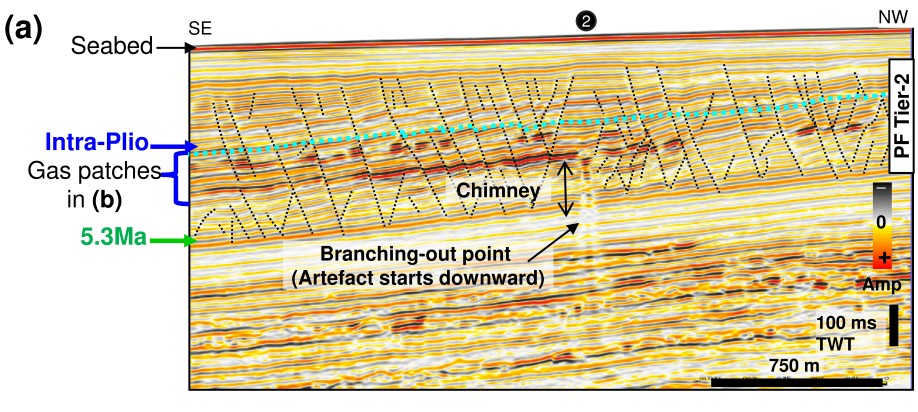

**(a)**

SE     ❷     NW

Seabed

PF Tier-2

**Intra-Plio**

Gas patches in **(b)**

**5.3Ma**

Chimney

Branching-out point
(Artefact starts downward)

0

+

Amp

100 ms
TWT

750 m

**(b)**

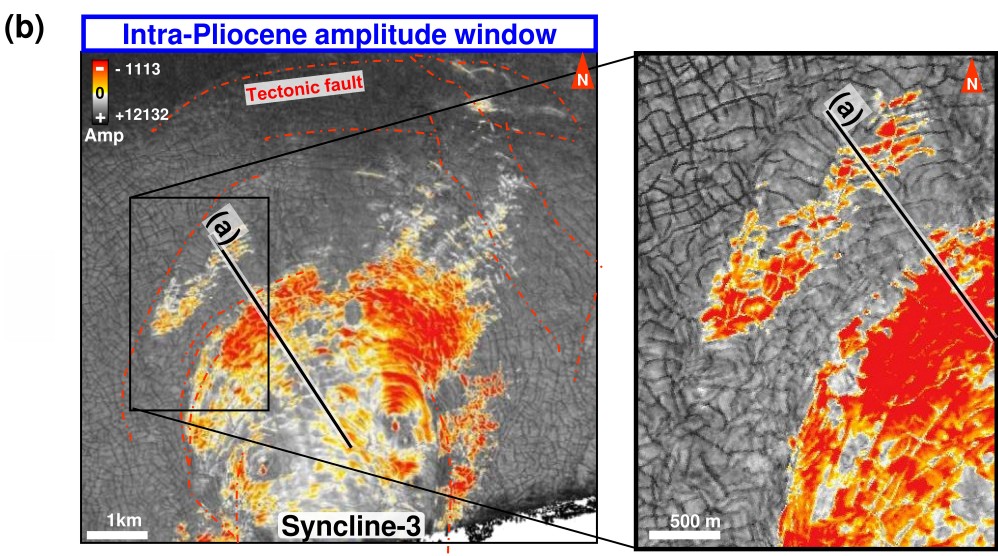

**Intra-Pliocene amplitude window**

- 1113

0

+12132

Amp

Tectonic fault

N

(a)

Syncline-3

1km

N

(a)

500 m

**(c)**

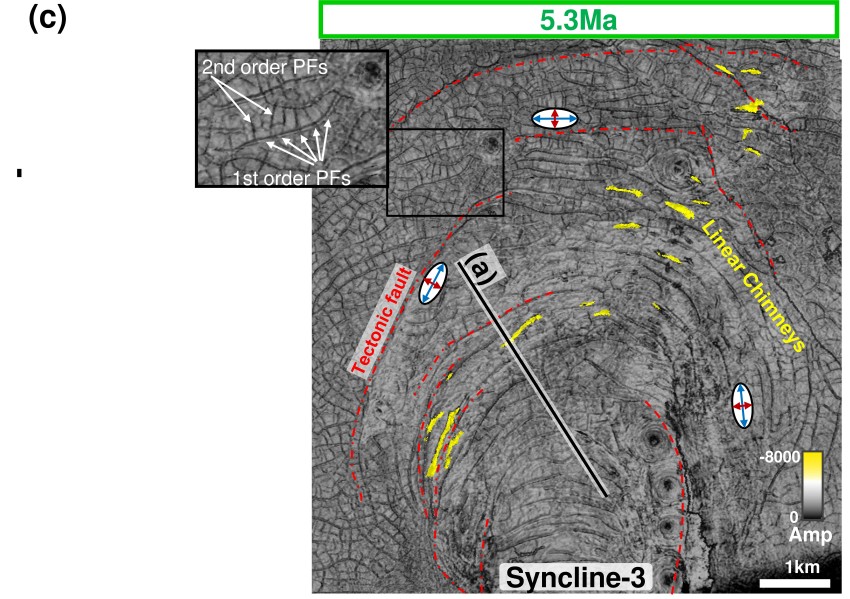

**5.3Ma**

2nd order PFs

1st order PFs

Linear Chimneys

Tectonic fault

(a)

Syncline-3

-8000

0

Amp

1km

**Figure 9.** Linear Chimneys of Type-2 and PF cells filled by gas in Syncline-3. a) Seismic section showing type-2 Linear Chimneys below gas-filled layers (expressed by negative high amplitude anomalies) within PF blocks capped by the regional impermeable barrier "Intra-Plio". Numbered black-filled circles indicate the type of chimneys. Line location shown on maps in (b) and (c). b) Amplitude extraction of a window in the middle of Tier-2 over syncline 3 showing gas-filled PF cells forming a km-scale gas accumulation laterally bound by extensional faults. Images sourced from Ho (2013; Ho et al., 2013). c) Overlaying of two amplitude maps shows the distribution of Linear Chimneys along the 1st order, anisotropic PFs that are parallel to synclinal tectonic faults (indicated by red dotted lines). The map of a negative high amplitude reflection within isolated chimney body (indicated by yellow colour) is superimposed on the amplitude map of horizon 5.3Ma at the base of PF Tier-2. Blue axis and red axes on stress ellipses indicate the palaeo-orientation of the intermediate and minimum stresses, respectively.

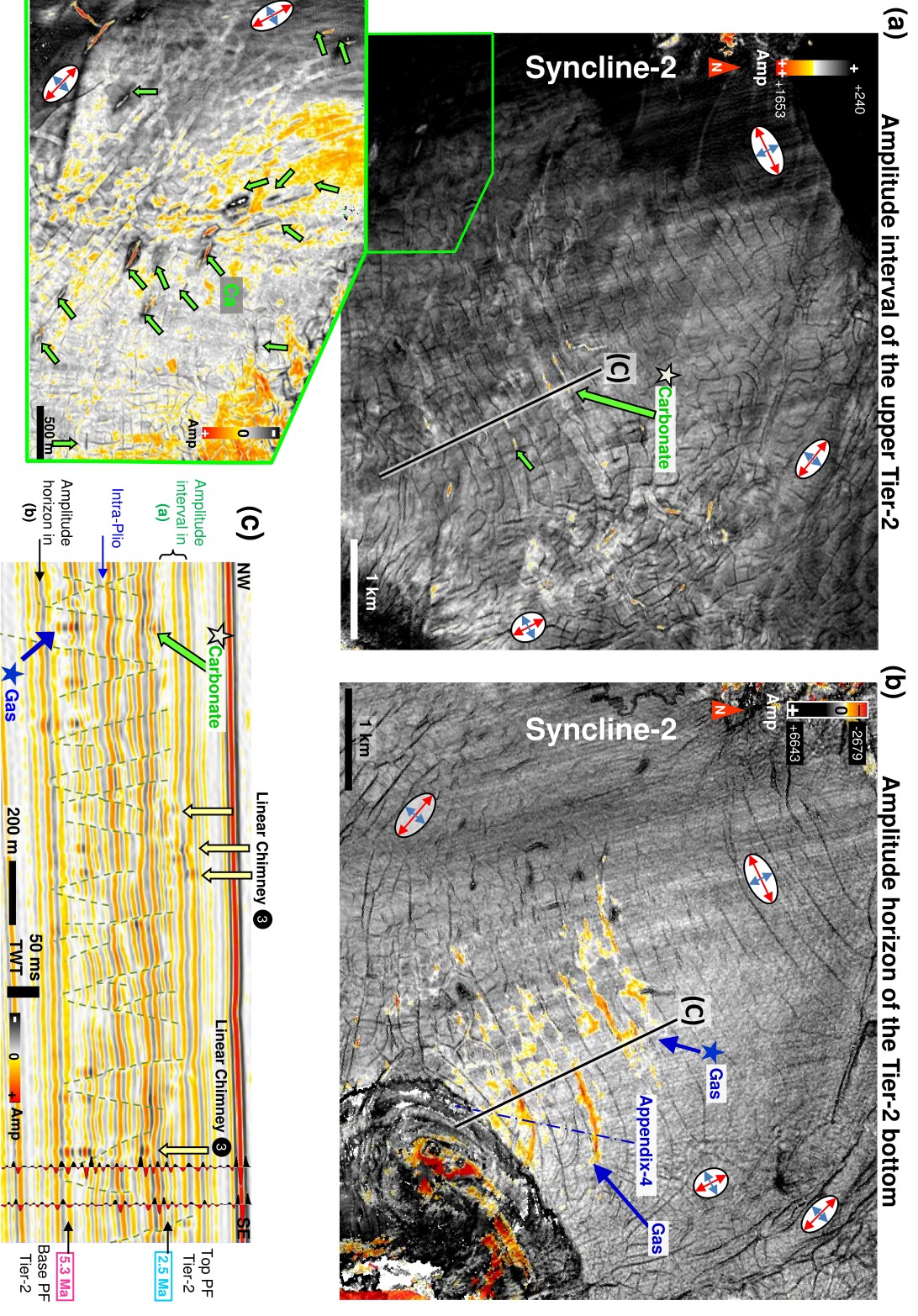

(a) Amplitude interval of the upper Tier-2

Syncline-2

Amp ++ +1653 / + +240

(C) Carbonate

1 km

Amp 0 + 500 m

Gas

(b) Amplitude horizon of the Tier-2 bottom

Syncline-2

Amp ++ +6643 / + 2679

1 km

(C)

Gas

Appendix-4

Gas

(C)

Amplitude interval in (a)

Intra-Plio

Amplitude horizon in (b)

Gas

Carbonate

NW

Linear Chimney ③

Linear Chimney ③

200 m

50 ms TWT

- 0 + Amp

Top PF Tier-2

Base PF Tier-2

5.3 Ma

2.5 Ma

SE

**Figure 10.** The linear planform of the topmost and lowest termination of chimney Type-3, and alignment of the Linear Chimneys parallel with polygonal faults in withdrawal Syncline-2. a) RMS amplitude extraction of a window in the upper portion of PF Tier-2 showing linear carbonates (red = Positive High Amplitude Anomalies - PHAAs). Exceptional examples of Linear Chimneys without preferential orientation can be observed close to the NE edge of syncline (see carbonates in the right side of the image). See brackets in (c) for the vertical location. Blue axis and red axes on stress ellipses indicate the palaeo-orientation of the intermediate and minimum stresses, respectively. b) Amplitude map of horizon at the base of PF Tier-2 showing the linear gas accumulations (red = High Negative Amplitude Anomalies - NHAAs). Note that NHAAs are aligned parallel with overlying PHAAs shown in (a). c) Seismic section showing positive bright spots (Ca for Carbonate) above and negative bright spots below Linear Chimneys. Numbered black-filled circle indicates the type of chimney. Green arrows indicate the position of carbonates, while yellow ones indicate locations of Linear Chimneys.

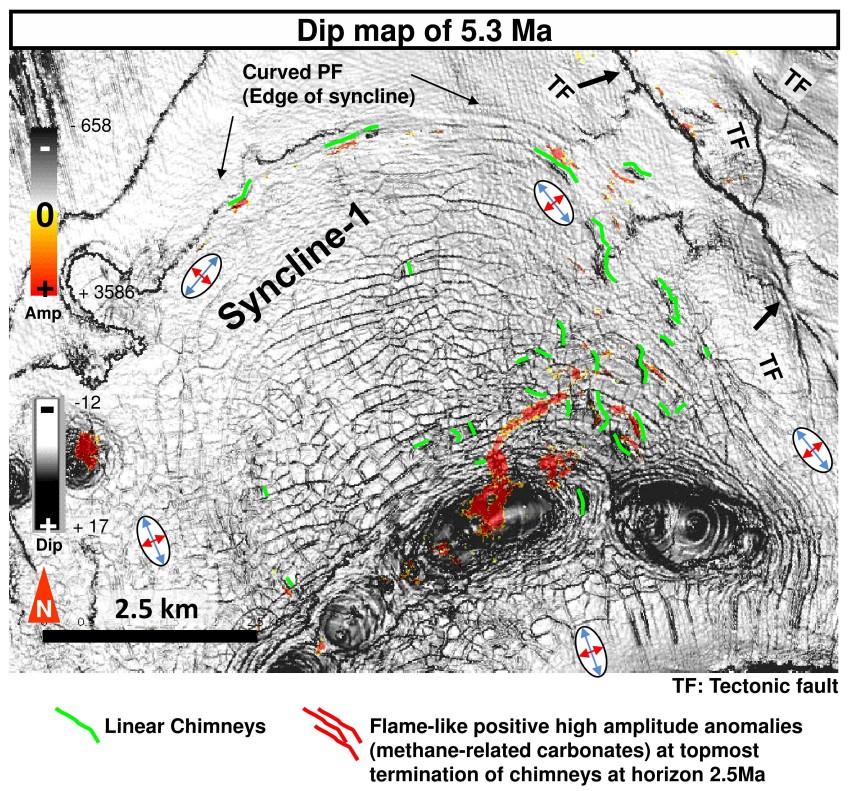

Dip map of 5.3 Ma

TF: Tectonic fault

**Figure 11.** Dip map of key horizon 5.3Ma (base of PF Tier-2) showing distribution of linear chimneys in Syncline-1. Isolated positive high amplitude anomalies (PHAAs) at the topmost terminations of Linear Chimneys at horizon 2.5Ma (interpreted as methane-related carbonates) are superimposed on the dip map. Green lines highlight the locations of Linear Chimneys underlying the carbonate. Red amplitudes are flame-like PHAAs. Blue axis and red axes on stress ellipses indicate the palaeo-orientation of the intermediate and minimum stresses, respectively.

# Percentages of chimneys-PF intersecting position
## (209 measurements)

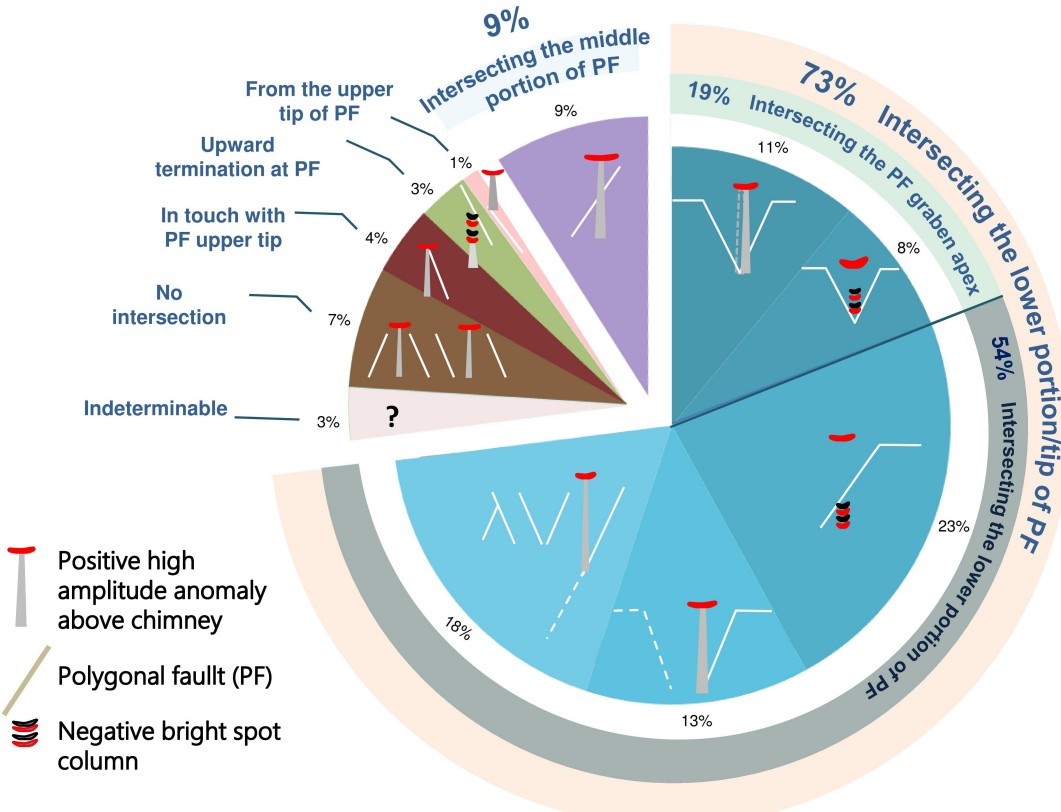

**Figure 12.** Pie charts showing the percentage of chimneys intersecting or emanating from different parts of fault planes or adjacent fault blocks. The position of the chimney-fault intersections are illustrated with cartoons in each pie segment. See key at bottom left for drawing codes. Image modified from Ho (2013).

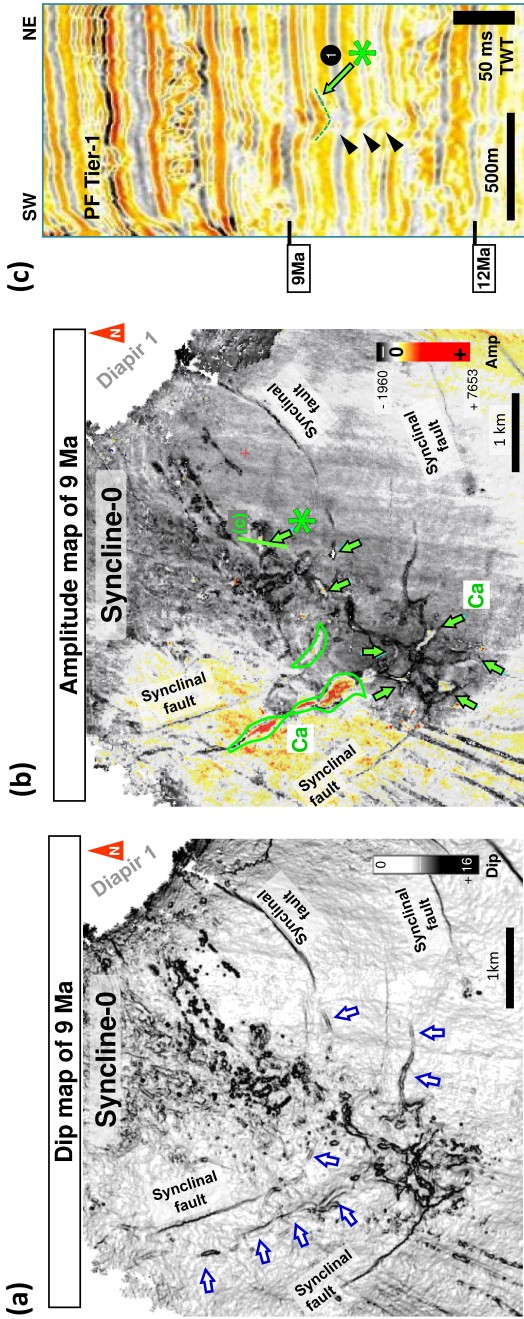

**Figure 13.** Kilometer-scale Linear Chimneys along the tectonic faults of Syncline-0. a) Dip map and b) amplitude map of horizon 9 Ma showing the geometry of a radial network of high positive amplitude depressions that overlays a big Linear Chimney network. Green star locates the PHAA depression shown in (c). c) Zoom of seismic profile in Figure 2b showing a PHAA within the radial network overlying a chimney. Black arrows indicate the chimney. Numbered black-filled circle indicates the type of chimney. See green dotted line in (b) for location.

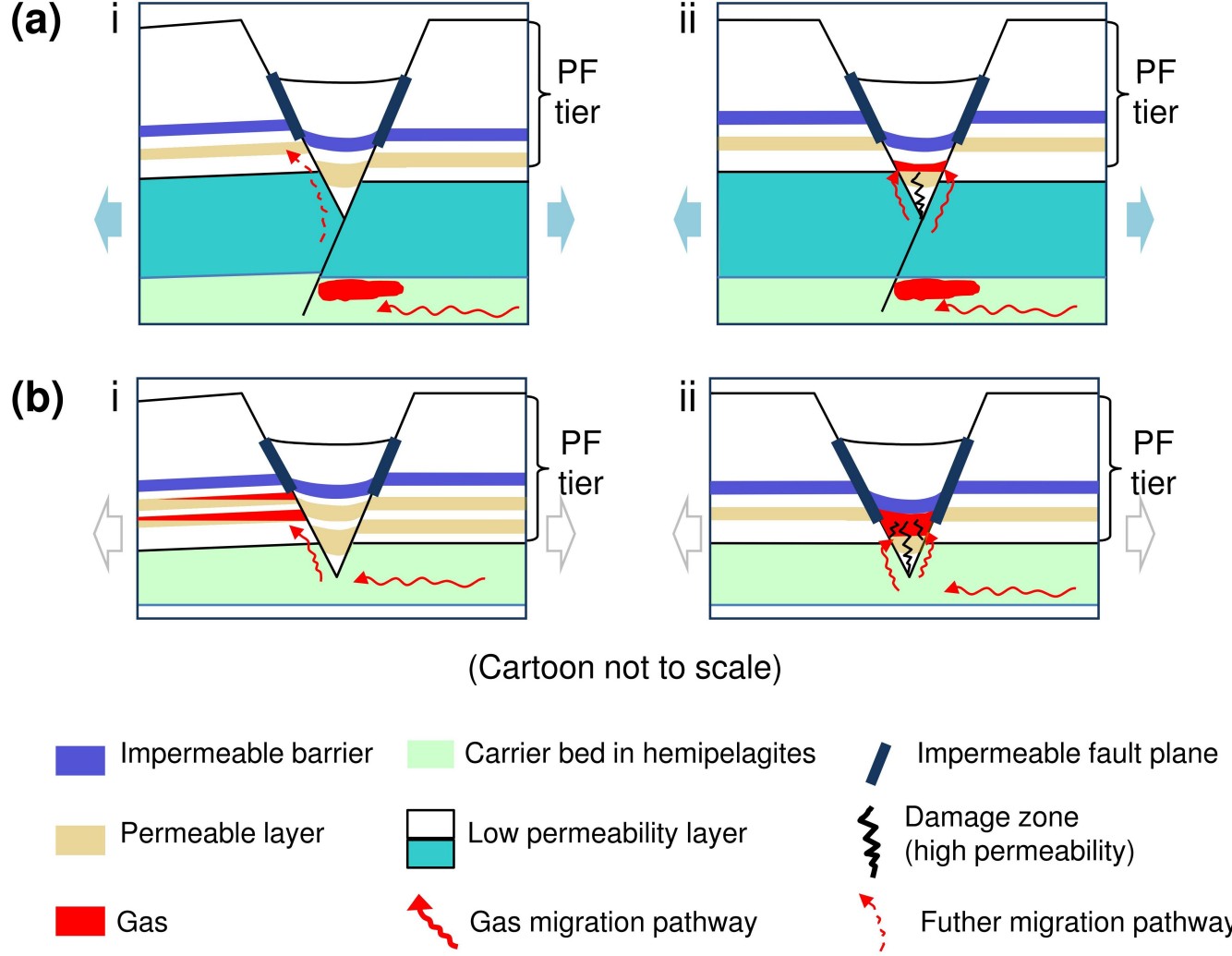

(Cartoon not to scale)

**Figure 14.** Conceptual model for free gas migration into the (multiple) permeable layer(s) in the lower part of PF Tier-2 from an underlying carrier bed. Bold black lines denote the segments of PFs interpreted as impermeable. Cartoon not to scale. a) Case where the carrier bed occurs beneath the basal tier surface of the PF tier (the carrier bed thickness can extend beyond the bottom of the cartoon). First-order PFs have propagated beneath the regional tier surface and have intersected the carrier bed. (i) Further gas migration into the permeable layer of the footwall and (ii) into the hanging wall. Gas migration into the hanging wall apex is likely because of the increase of permeability induced by apex damaging. b) Cases where the carrier bed occurs at the base of a polygonal fault tier. (i) Gas migration into the permeable layer in the footwall via the permeable portion of polygonal fault plane. (ii) Gas migrates directly into the overlying permeable layer in the fractured hanging wall when it is juxtaposed against the carrier bed.

**(a)**

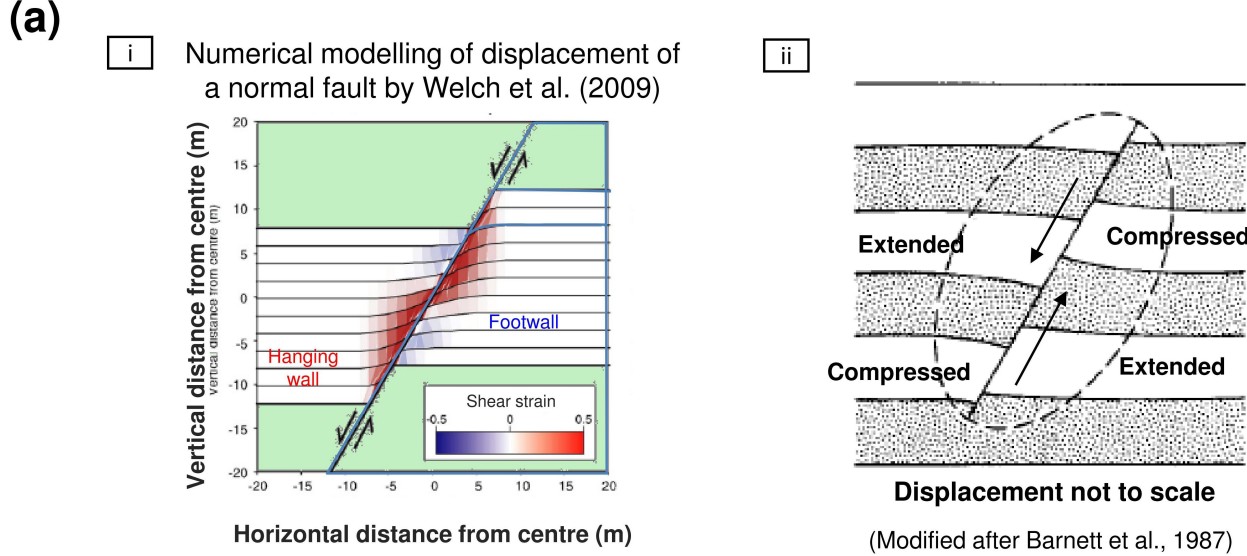

**(c)** Higher stratigraphic position

**(b)** Impermeable barrier

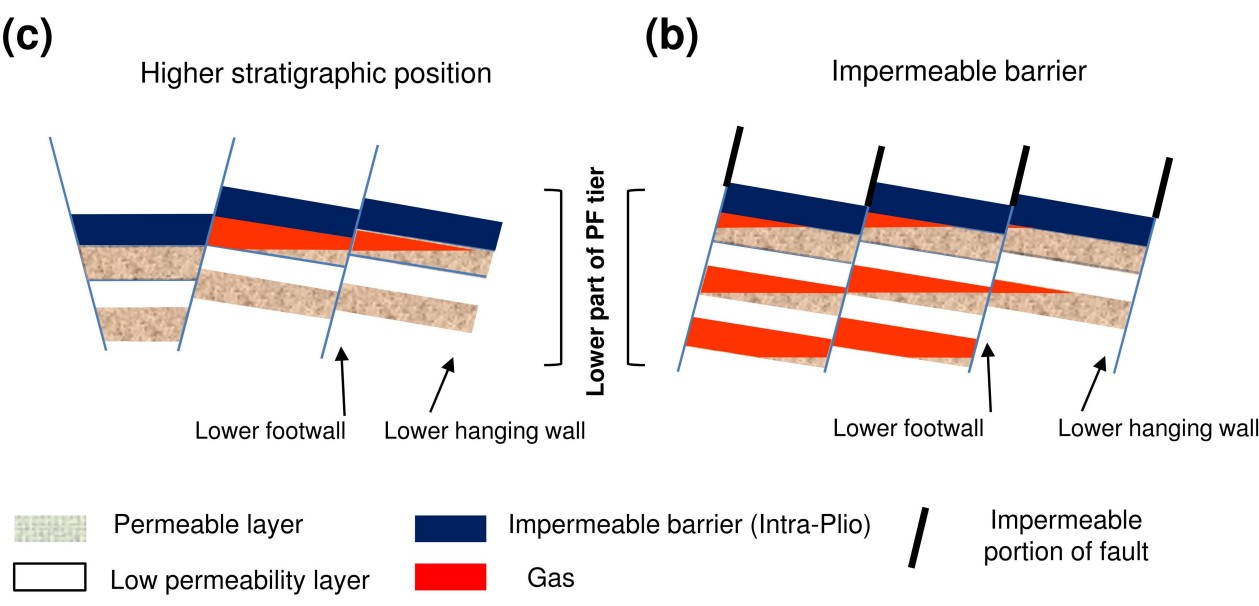

**Figure 15.** Causes of gas retention within the lower part of PF blocks. a) Stress distribution around a normal fault. i) Magnitude of shear strain around normal fault (Modified after Welch et al, 2009). ii) Displacement of horizontal beds by a normal fault and regions of relative extension and compression in the vicinity of a normal fault in shallow unlithified sediments (Modified after Barnett et al., 1987). b) Higher stratigraphic position of permeable layers in PF footwalls (in compare to hanging wall) leads preferential gas accumulation in such locations. c) An enough impermeable barrier in the middle of the PF tier prevent further upward migration of free gas.

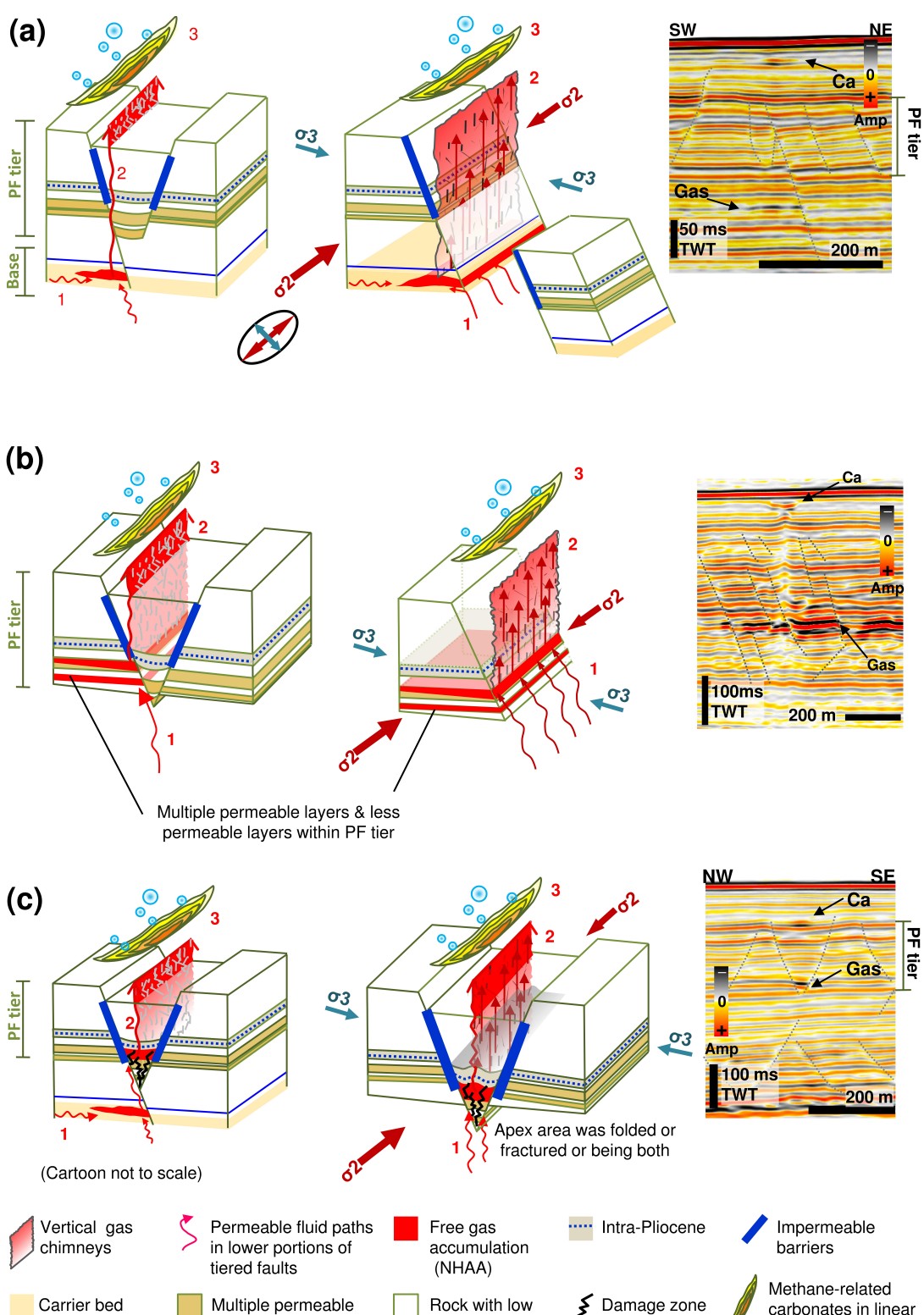

Multiple permeable layers & less permeable layers within PF tier

(Cartoon not to scale)

Apex area was folded or fractured or being both

Vertical gas chimneys

Permeable fluid paths in lower portions of tiered faults

Free gas accumulation (NHAA)

Intra-Pliocene

Impermeable barriers

Carrier bed below PF tier

Multiple permeable layer (simplified)

Rock with low permeability

Damage zone

Methane-related carbonates in linear depression (PHAA)

**Figure 16.** Left column: Conceptual models illustrating alternative gas migration pathways through the PF tier. Right column: Seismic lines showing the critical observations. Free gas on seismic profiles is expressed by negative high amplitude anomaly (NHAA) while methane-related carbonates are expressed by positive high amplitude anomaly (PHAA). a) Below the PF tier where Linear Chimneys originate from; free gas is trapped by fault-bound traps formed by tectonic faults or deep-rooted PFs. b) Gas accumulation in the PF-bound traps in the lower part of PF footwalls; to notice the possible occurrence of multiple permeable layers filled by gas. c) Free gas trapped by lower hanging wall of PFs in which accommodation space was created by fractures or fold during subsidence.

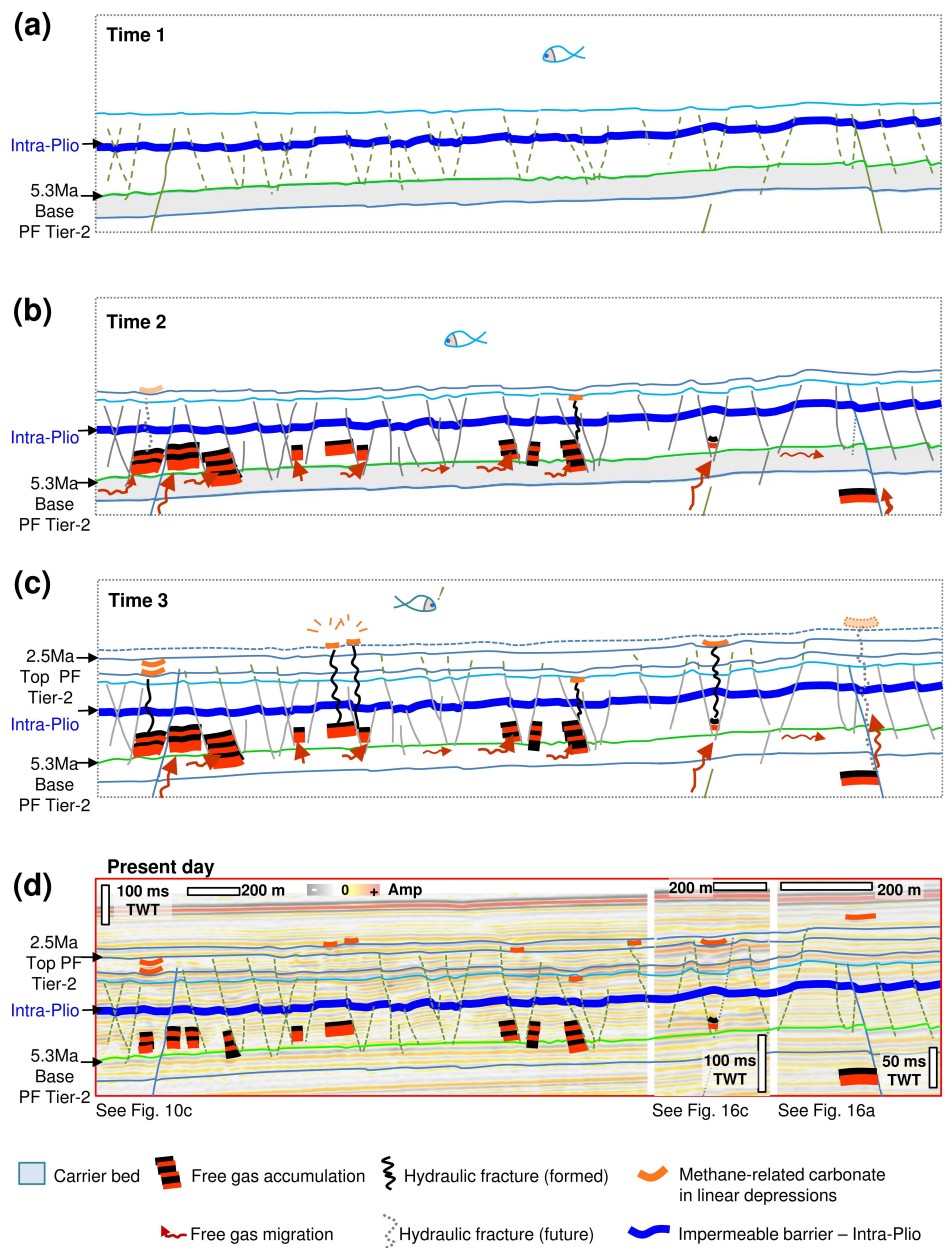

**Figure 17.** Conceptual model for the time steps of gas migration into the PF tier and formation of Linear Venting Systems. a) Initiation of PFs and deposition of impermeable barrier (Intra-Pliocene). b) PF tier and associated gas traps formed. Gas migrated along tectonic faults and reaching the carrier bed below the PF tier and then flowed into the PF-bound traps via roots of PFs. c) Overpressure in PF-bound traps induced hydraulic fracture which propagated vertically to the seafloor. Continuous gas expulsion induced the formation of depressions and methane-related carbonates above the fractures. PFs continue to propagate upwards during sedimentation. New generations of Linear Chimneys and depressions formed. d) The present day state. The whole tier is buried. The drawing of the final stage overlays the corresponding seismic sections.

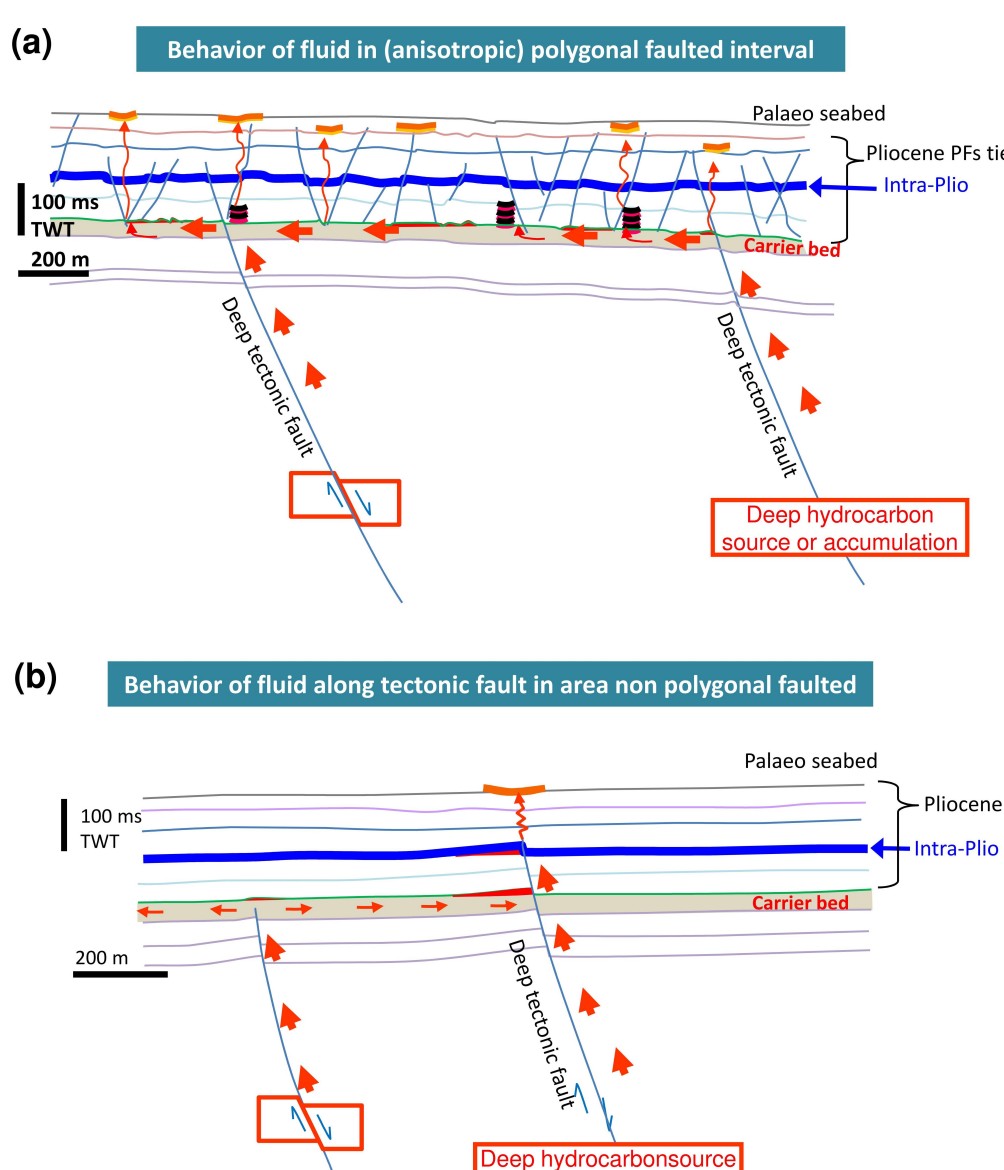

**Figure 18.** Conceptual models for two styles of gas migration into interval with or without polygonal faulting. a) Drawings show the trajectories of gas migration pathways branching out along the bottom of PF tier. b) Migration pathways focused along tectonic fault within the Pliocene interval.

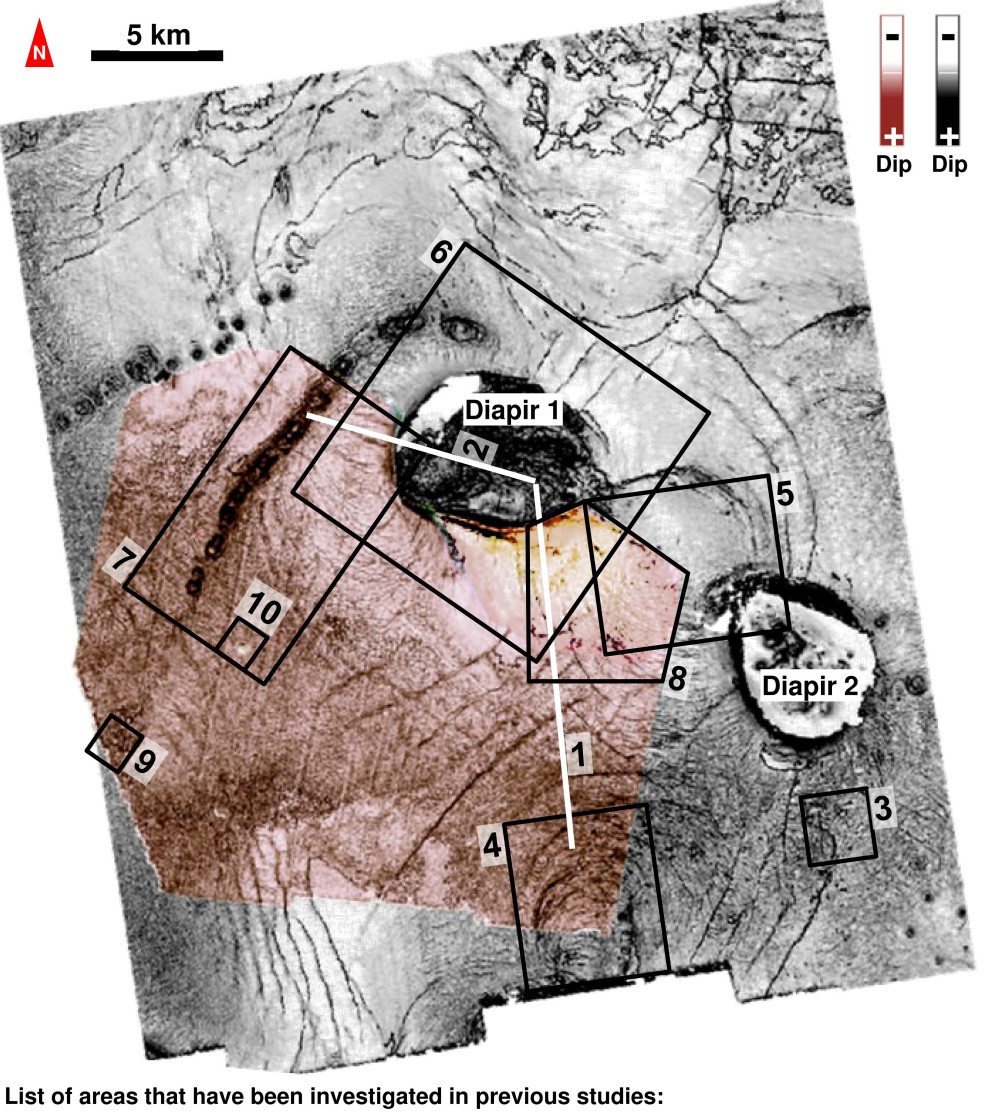

**List of areas that have been investigated in previous studies:**

1) Fig. 1 in Ho et al. (2013) (see also Appendix-5)
2) Fig. 1 in Ho et al. (2013)
3) Fig. 2a in Ho et al. (2013)
4) Fig. 2d in Ho et al. (2013) Fig. 2b in Ho et al. (2012b)
5) Fig. 2c in Ho et al. (2013) Fig. 2b in Ho et al. (2016)
6) Fig. 3d in Ho et al. (2013)
7) Fig. 2a in Ho et al. (2012b)
8) Fig. 3b in Ho et al. (2012a)
9) Fig. 7b in Ho et al. (2012a)
10) Fig. 8b in Ho et al. (2012a)

The high resolution survey corresponds the pink area.
The regional survey (grey area), which Ho (2013) and this study based on, was uniquely available and accessible in Total, France.

**Figure A1.** Appendix 1. The areas investigated in previous studies (Ho, 2013; Ho et al., 2012, 2013, 2016) are shown on the dip map of horizon 5.3Ma. Superposition of the high resolution survey (pink area) and the regional survey (grey area).

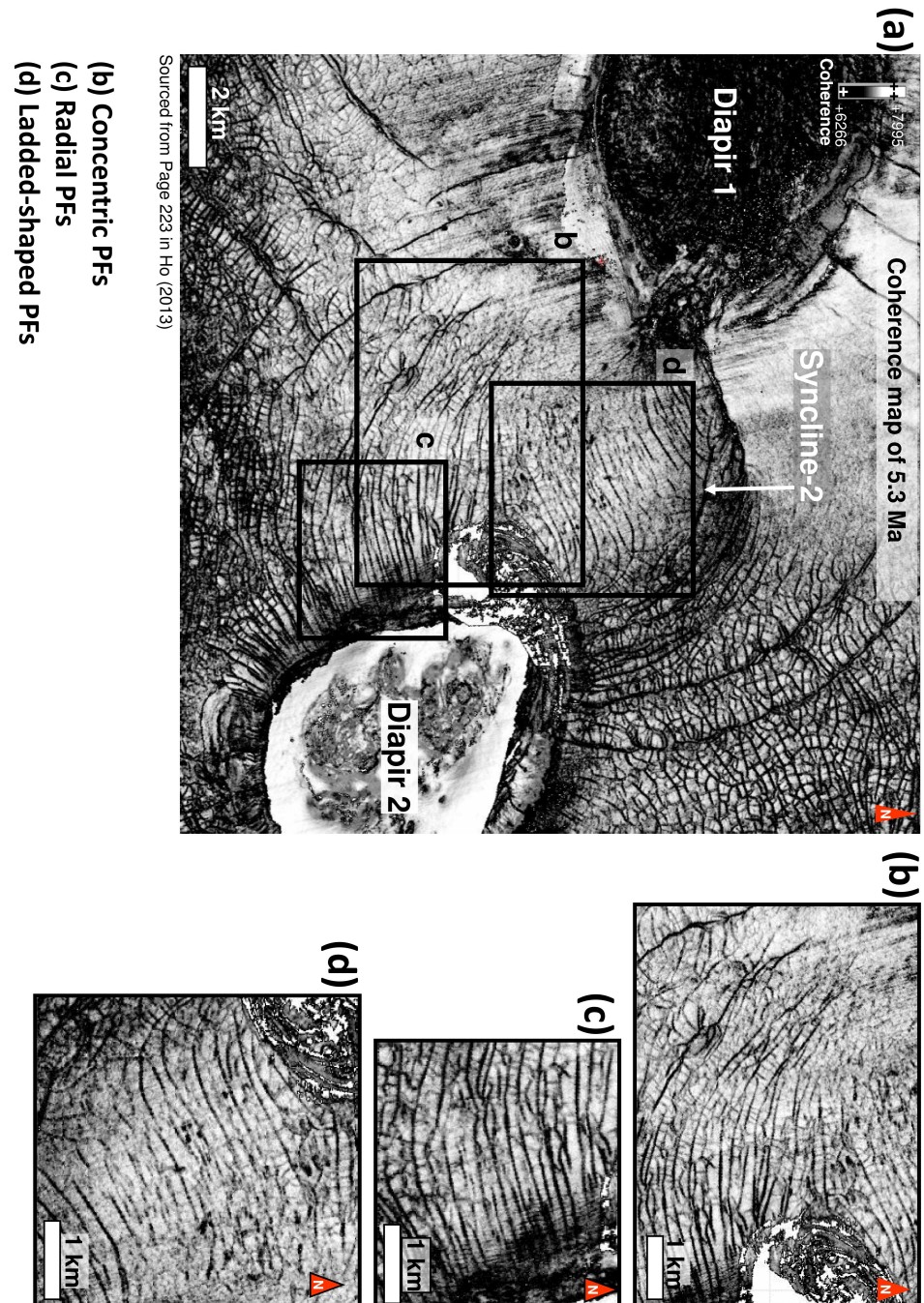

**Figure A2.** Appendix 2. Different patterns of anisotropic PF networks in which Linear Chimneys are found. a) Coherence attribute extracted onto the basal surface of PF Tier-2 showing the geometry and preferential alignment of PFs in withdrawal Syncline-2 and around Diapirs 1 and 2. b-d) Close ups of the map in (a). Locations of maps are shown by coloured squares. Sourced from Page 223 in Ho (2013).

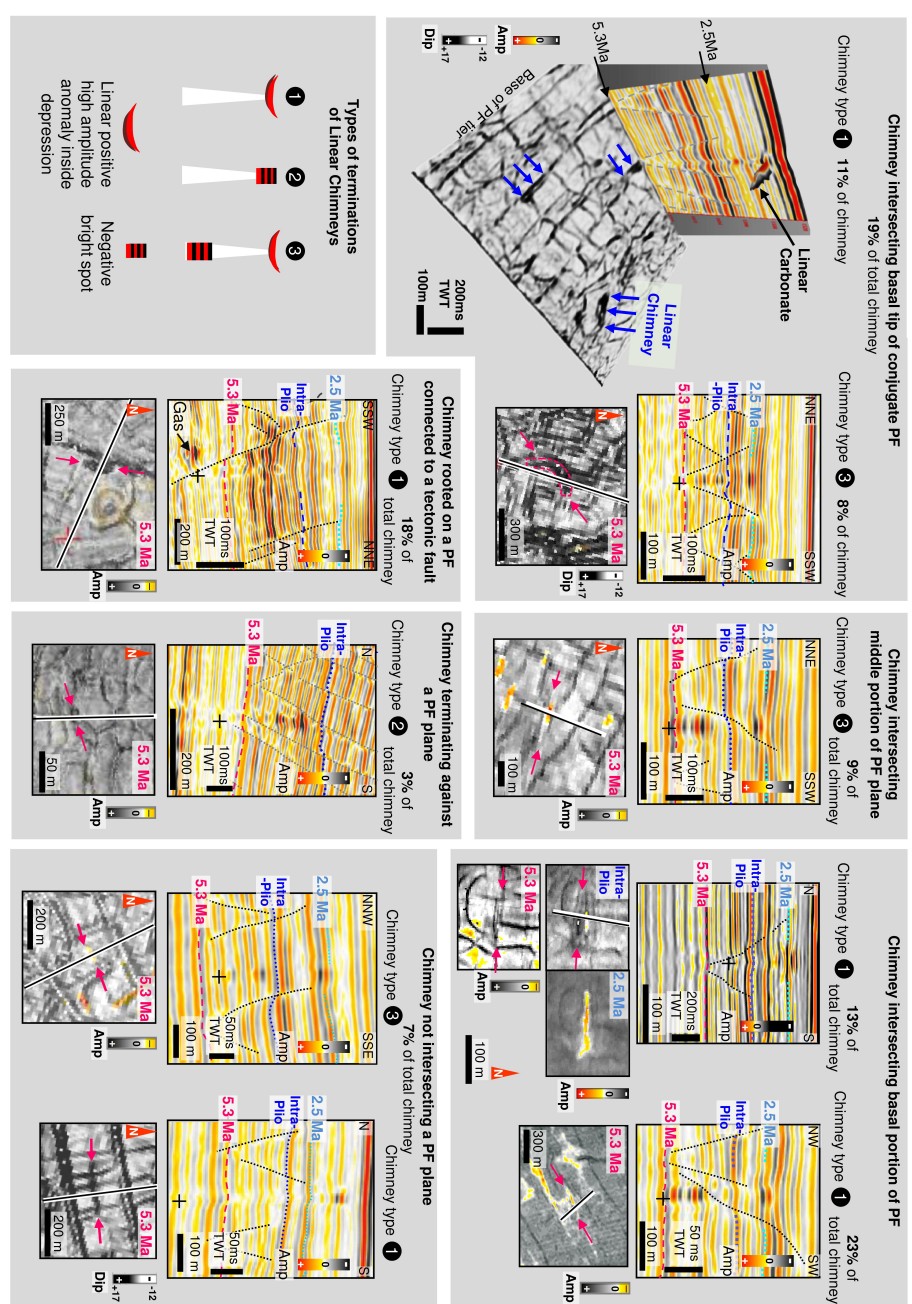

**Figure A3.** Appendix 3. Examples of different groups of chimneys intersecting and emanating from different parts of polygonal faults (PFs). The linear vents of Type-1, 2, 3 are labelled with black circles. The percentages correspond to the numbers of chimneys intersecting fault planes at specific positions (see pie chart in Fig. 12). The planform dimensions of the chimneys are shown on maps below each section. The apparent bases of chimneys are marked with crosses. See description in section 4.1.1.

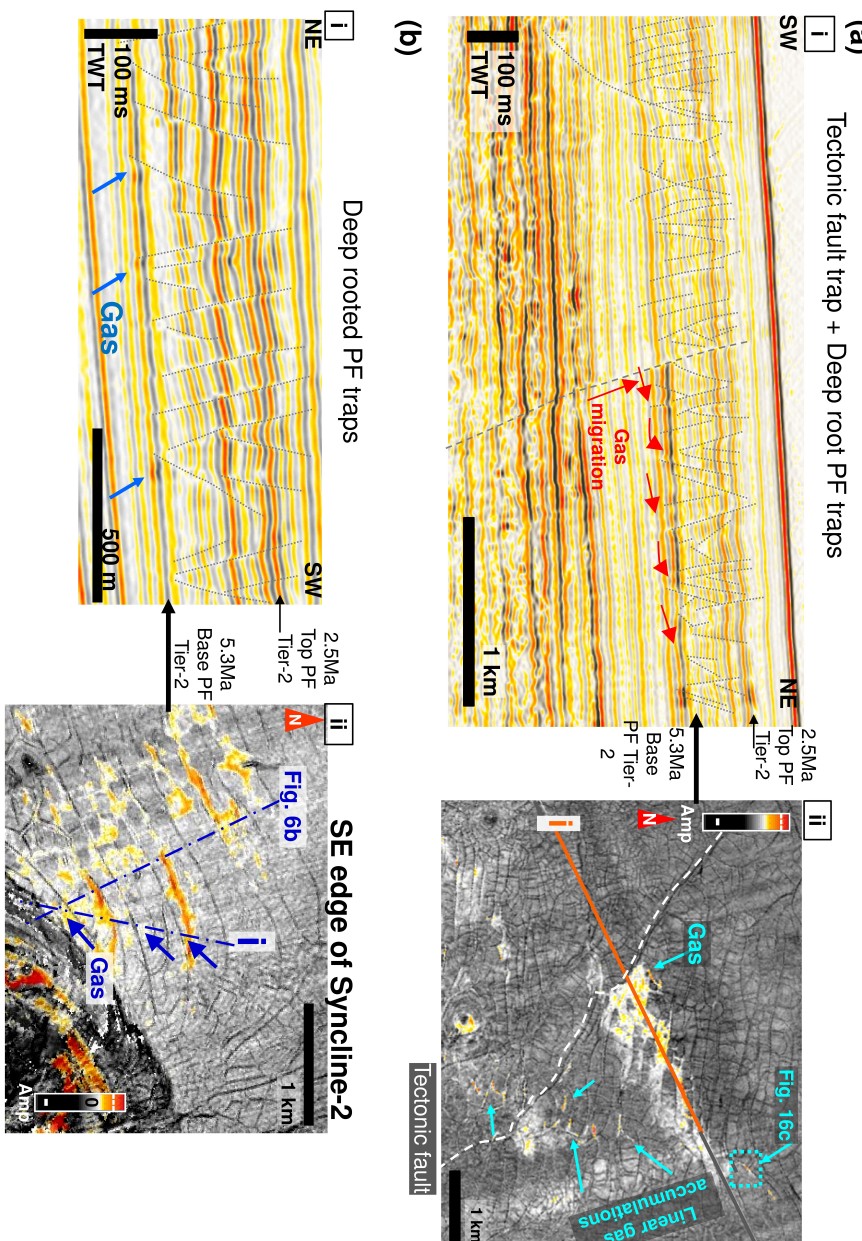

**Figure A4.** Appendix 4. Linear patches of negative high amplitudes interpreted as gas accumulation at the base of PF Tier-2. a) Free gas interpreted to have migrated along a tectonic fault and flowed preferentially into a permeable interval at the base of Tier-2 in the higher elevation (i.e. tectonic fault trap) as shown on seismic profile (i). Note that the deep-rooted PFs extend deeper than the tier (forming PF traps). (ii) The amplitude map along the base of Tier-2 shows that gas had actually filled the whole PF cell traps. b) Gas occurs around the bottom of deep-rooted long PFs as shown on (i), and exhibit linear planforms on the amplitude map of Tier-2's base (ii).