# Peer review of "Formation of linear planform chimneys controlled by preferential hydrocarbon leakage and anisotropic stresses in faulted fine-grained sediments, Offshore Angola"

_Solid Earth, 2018_

## Referee Comment (RC1) · A. Plaza-Faverola (Referee) · 10 Jun 2018

The paper investigates the correlation between the distribution of near-surface fluid migration features and different kinds of polygonal and tectonic faults at a deep marine setting offshore Angola. The tectonism is primarily associated with salt diapirs. The authors document a range of fluid migration features in the area from which the most particular ones are linear chimneys. They propose a conceptual model that explains the orientation and distribution of gas chimneys by alteration of local stresses adjacent to polygonal faults. This paper include a compilation of observations published in the

framework of Sutieng Ho's PhD thesis, but here the authors focus on the effect of physical stresses on fluid dynamics. The data available for the study as well as the maps showing the fluid flow features are of high quality. The observations and thorough analysis of the geological controls on fluid flow in the region are highly relevant for the understanding of chimneys and shallow gas dynamics at other margins. The aims of the study are concrete and the introduction provides an excellent compilation of relevant references regarding stresses associated with fault planes. The cited studies goes from pioneering geomechanical concepts in the 80s to recent laboratory and modeling studies about shear and stain on sediments adjacent to fault planes. I must say, however, that the paper is very difficult to read, mainly because of the amount of observations presented in figures that are overloaded with insets, sub-parts and details. Some of the interpretation can be challenged. I came up with a few main points that I believe could be addressed to improve the manuscript. A list of detail comments annotated while reading through the paper is also provided.

Main points:

• Polygonal faults: the core of the paper is the relation of shallow gas accumulations and chimneys with the presence of polygonal faults. Although the authors do reference key studies related to polygonal faults (identified in the area but also globally), I think that a little bit more emphasis on describing the main aspects of such faulting is needed in the paper. I usually think of polygonal faults as those faults forming "polygons" in fine-grained sediment sequences. I realize that in this paper there are also radial faults associated with the synclines next to the salt diapirs that are as well referred to as polygonal? Are these termed polygonal faults because the mechanism of formation is similar to the other ones? How exactly do they form? Do they really reactivate? Is it really meaningful to talk about foot wall or hanging wall if the faulted blocks can be one or the other depending on the reference fault plane? They do not form sub aerially right? Section 3.2 could concentrate all the relevant details about the types of polygonal faults observed here and what exactly are the main characteristics of such

faults.

• Stress distribution at polygonal fault planes: The use of the principle of particle motion and the distribution of stress at each faulted block is great for explaining migration pathways and leakage distribution. However, I wonder hoe meaningful is the terminology of footwall and hanging wall for polygonal faults. In cases where there are several parallel polygonal fault planes, one block may be the hanging wall with respect to a block on one side but it would be the footwall of the next fault. It is difficult to see that the shallow gas accumulations are exclusively at the lower part of the foot wall of polygonal faults (if I understood correctly the interpretation by the authors). I do see high amplitudes on what could be a hanging wall or a footwall depending on which fault segment for the polygonal set I choose. So I think it makes more sense to talk generally about focused regions of higher shear stress and dilation depending on the relative motion of the faulted blocks with respect to each other. In general, the discussion about stresses is hard to follow. I think it would help to use only one figure to project all the relevant stress vectors inferred at the local zones of fluid leakage, together with regional stress vectors from, for example, salt-tectonics. Where do the blue and red vectors in figure 4 and 6 come from? Are these measured orientations of principal stresses or inferred? Figure 10 says the vectors are local + in situ. What does in situ mean in this context, how are these stresses estimated? Information about regional and local stress fields are estmated in the region is missing. Also the figure captions should explain what those vectors signify.

• Linear chimneys: Are the "linear chimneys" really chimneys? What is the definition of chimney used here? Aren't these features fractures/small scale fault planes where the fluids literally escaped through and formed authigenic carbonate that together with trapped gas in the system creates the blanking in the seismic? Is there evidence of breciation, or hydrofracturing in the regions interpreted as linear chimneys? Actually, the illustration of the chimney features is not that great in the figures. And this brings me to the next concern. In the data section the authors mention that different stacks were

produced grouping angle of incidence. It is indicated that the seismic data presented in the paper is from the near offset stack. The seismic profiles shown in figures 3 and 4 are from which stack version? The character of what is indicated as chimney in panel e of figure 4 reminds me how gas chimneys look in undershooting data, namely along a stack using just a selected range of offsets. Two vertical blank regions appear to each side of a high amplitude zone; the separation between the two vertical blank regions increases with depth. The feature appears then as a double line on the maps. This is a bit confusing for a reader that is not familiar with the processing of the data. I wonder whether showing the character of the chimney on the full stack is more intuitive and straight forward. If the double pattern of the so called linear chimney is not due to processing, it would be interesting to her what causes such a particular feature (figure 4 a and b).

• The PHAAs are interpreted as carbonates. Why would they always be associated with depressions rather than mounds? By analogy with carbonate mounds at present day seafloor, buried authigenic carbonate concretions within chimneys can get buried and appear as mounded features in the middle of a cavity with onlap of reflections at the flanks. The interpreted buried carbonate concretions don't show positive morphologies in this case? Can it be related to the resolution?

• The figures are of high quality. However, even if they are over loaded with insets, some times they lack explanations of features that seem relevant for the interpretation. For example, panel b in figure 4 shows a white band braking through a high amplitude reflection. What is that feature? It is really hard to link all the different insets. In figure 5 I don't manage to identify where is the feature pointed with a yellow arrow on 5ai, on 5aiii. Is it correct that the seismic is NW to the right? It is important to find a way of simplifying these figures. Figure 5 could be split into 2 figures. I would suggest selecting 7-8 figures to show the main observations and to illustrate the conceptual models. Despite all the figures in the main text the authors still refer to appendix figures for observations that are key for the paper. The figures in the main text could be used

in a more efficient way.

Please find below my notes while reading through the text; it includes a few typos. The L refers to the paragraph number and the P to the page number.

- L20/P3: minimum and maximum offsets? I assume these data sets are multi channel with long offsets? It is kind of important to provide this info before it is mentioned that the amplitudes vs. angle were used for verifying the seismic character of the observed features (a sort of undershooting?).

- L25/P3: typo: to map the linear. . .

- L20/P4: figure 6b referenced before 3, 4 and 5? Check the flow of the figures. It is difficult to see from figure 2b what is stated here: that studies chimneys occur primarily in syncline areas. Maybe indicate the syncline structure in 2B and relate better 2a and 2b?

- L25/P4 typo: relief instead of relied?

- L10/P5 typo: check the unit used is it 10 to 100 s, ms or m?

- L5/P6: The linear features shown in figure 4a-b are indeed strange features. Are these really along polygonal faults? (PFs?). Polygonal faults usually do not have a preferred orientation, but on the contrary, they consist of fault segments oriented covering the whole azimuth range (closing polygons), right? The linear features seem to follow the circular structures to the north of the syncline. There seems to be an overarching control on the orientation and distribution of these features rather than the polygonal faults as such. I guess I am missing a clear definition of what the authors are referring to as polygonal faults. For examples, are types 1-3 described by the authors termed polygonal faults because they all formed due to dewatering of fine-grained sediments? See main comment.

- L20/P6: typo, 19% FORM

- L30/P6: Consider making two different figures here. The figures have so many parts that it is actually a struggle to go through and understand everything. Is the statistical part of the figure really relevant? What do we do with the fact that 54% are intersecting the lower portion of PF? What seems most relevant in this paper is to get compared the orientation of the fluid flow related features with respect to the orientation local and regional faults and fractures, right?

- L10/P7: Check the use of tenses in this paragraph.

- L20/P7: Is the evidence by Sonnenberg et al., 2016 related to the polygonal faults in this present study? In that case, it would help to see a sentence hinting what is the observation that works as evidence. I got this advice recently and I kind of see now the need for bringing into the current study the key observations rather than referring the reader too often to the previous studies. This degrades the flow of the reading and makes difficult to follow the paper.

- -L30/P7: do polygonal faults really reactivate? How is the accommodation of such movement if the fault planes can converge to each other rather than been parallel? Aren't these kind of faults associated with diagenetic processes and are hence a kind of one-time event?

- -L20/P8: 60 m deep and 4.5 m wide pockmark??? That is quite deep compare to the with, it is almost a conduit rather than a pockmark.

- -L30/P8: so there is active gas release at the seafloor at present? Or you mean active in the sense that there is gas filling the near-surface systems through the gas chimney structures?

- -L30/P8: this model is hard to digest here since there are so many faults and pre-existing weak planes that one would think that the fluids would find preferential pathways without much effort and hence gas chimneys would not be favor?

- L5/P9: I could not find figure 6b. In general, it is difficult to find in the figures some

of the observations regarding gas chimneys. Again, the figures could be simplified by selecting only key examples.

- L20/P9: The use of appendix figures to illustrate what seems to be the main conceptual model of the paper is not ideal. One figure in the main text should be enough to illustrate description of the model for fluid migration and development of chimneys. Figure 8a doesn't illustrate this, or did I miss something? When you mention PF tier here is it 1 or 2? It is very easy to get lost while reading, I think it is due to the fact that there are so many figures overloaded with details.

- L25/9: typo: the PRESENCE of...And please revise this paragraph. These aspects are not necessarily ruling out each other and a combination of them could be a precondition for explaining your observations. Consider reformulating the paragraph.

- L30/9: where do we see the gas accumulating at the foot wall? The foot wall of a major tectonic fault or are you referring to small faulted compartments resulting from the polygonal faulting? If so, is it really meaningful to talk about footwall if the blocks are somehow both footwalls and hanging walls with respect to each other?

- L5-10/P10: Why would these areas be subjected to "relative" compressional strain? What do the authors mean here? Would it be more appropriate to say "less" compressional state rather than relative? In analogy with the particle motion maps for earthquakes (focal mechanisms) one would expect that the lower part of the hanging wall and the upper part of the footwall would experience more compression while dilation would dominate the upper hanging wall and the lower footwall (which is indeed consistent with Barnett et al., )

- L10/P10: again, no figure 6d and also the interpretation that the gas is accumulated in the footwall of polygonal faults is not easy to digest. A block can be considered a footwall or a hanging wall, depending on which fault plane is used as reference. I can see high amplitudes in both hanging and footwalls in figure 6c for example.

- L20/P11: Typo: BY. . .

- L20/P12: Again, the model of shear stress distribution through the four quadrants of the faulted blocks is very sounding for explaining the distribution of gas into more permeable zones. However, where these more permeable zones are entirely correlated with footwalls and hanging walls in these polygonal fault system is hard to assimilate. Is it really necessary to use this terminology? See main comment.

- L25/P12: So the chimneys grow episodically? You foresee that the growth occurred in several episodes of reactivation of the system? It is important to describe this more explicitly in order to be able of comparing to systems from other margins with comparable settings.

- L5/P13: typo: may BE because..

- L10/P14: typo: blocks

- L25/P14: point 6 in the discussion. Check the grammar here. The sentence has a problem. When are the chimneys circular in isotropic stress fields? And when are they linear, isotropic and anisotropic? Clarify.

- Figure 1: great figure. The use of a dip map to show all the elements of the study works extremely well, we can see the flanks of the salt domes and even the fine scale faults and fractures. However, isn't the present day bathymetry important to understand the stress regime?

- Figure 2b – typo: linear positive high amplitude. . .It is a bit difficult to read through the symbols of this figure. The idea of overlapping the symbols related to different fluid flow features on the seismic profile is great. However, it is not easy to see the actual seismic feature (in particular the high amplitude anomaly depressional network). I assume the location of the symbols just indicate the interval where each feature is observed rather than the actual feature? Maybe this can be hinted in the caption (as a help for the reader). Is the seismic section a compressed version of the geological section in 2a or

only part of that transect?

- Figure 5: Can you really tell that the high amplitude analogies in inset ii are gas accumulations, without any clear sign of polarity change?

- Figure 6: Not convincing with the positive and negative bright spots interpretation. If you think it is key to differentiate between carbonates and gas for the discussion you may need to show a better indication for this, perhaps using wiggles and zooming into the anomalies?

- Figure A7 typo: Gas MIGRATION into the hanging wall apex WAS likely because of the increase (check the sentence in any case)

---

## Referee Comment (RC2) · Anonymous Referee #2 · 13 Jun 2018

**1. GENERAL COMMENTS**

The paper contains novel scientific concepts and ideas about the interaction between polygonal faulting and chimneys within the scope of SE. Substantial conclusions are reached and presented around the orientation of linear chimneys with respect to polygonal faulting, and how these features can be used to work out timing of hydrocarbon migration. Scientific methods and assumptions seem valid and are outlined, I suggest a couple of ideas below which would improve this further. The description of the results obtained is clouded by figures, I suggest how these can be re-ordered and simplified

to better convey the results. The authors do credit related work and clearly indicate their contribution. The title clearly reflects the contents of the paper. The abstract is not as clear as it might be due to poorly constructed sentences, however in principle it summarizes the contents of the paper and I make suggestions to improve the grammar.

The overall presentation of the authors results and ideas is badly let down by the quality of the figures and of the referencing to the figures in the text. I suspect some of the figure references in the text are "broken", with the figures cited in the text not always supporting the argument. This makes the paper hard to read (and review). It appears to me that the authors do have the necessary material to back up what they write in the paper, however these are in need of substantial reordering. I suggest how this can be done below.

The language suffers from typos and a few examples of poor grammar, I suggest how this can be rectified in detail below. I am satisfied that mathematical formulae etc are correct. As I suggest above, the number of figures can be reduced. I suspect that the second seimic survey referred to in the text does not add anything, but it is difficult to tell. The number and quality of references appears to be appropriate, although I have not checked this in detail.

Given the scientifically interesting ideas and discussion, I consider this paper worthy of publication in SE, however the presentation needs some potentially substantial work first on presentation first.

In this review I run through how I would lay out the figures, and share my notes on how the text can be further improved.

2. FIGURES

I suggest a significant overhaul of the figures, cutting out what does not contribute to the story. By doing this the number of figures can be slightly reduced, but all of the figures can be significantly decluttered, making it easier for the reader to see and

understand the interesting conclusions the authors have reached.

DATA and METHODS

Fig 1: Simple map highlighting position of the two seismic surveys to the coastline and the wide geological environment described under "Regional Setting"

Fig 2: Sample seismic line on each dataset through a feature of interest, showing what this looks like on full, near, mid and far offset data, demonstrating the resolution of the data etc

GEOLOGICAL SETTING

Fig 3: A super-regional seismic line (if possible), or as regional line as you can manage across the two datasets with penciled in interpretation if data confidential, showing the rift, sag and passive margin sequence and geometry. Include an inset zooming into the interval of interest, highlighting the different units.

Fig 4: A map highlighting the features of interest (perhaps after your existing figure 1) with: 1) a multi-segment seismic line going through each of the synclines and diapirs. Use this seismic line to support statements made about timing of syncline subsidence etc 2) a short seismic line or lines (two maximum) highlighting the other features described in section 3.2

OBSERVATIONS AND DISCUSSION

Fig. 5: A seismic line highlighting a PHAA and NHAA. These should be clearly labeled and not covered by interpretation.

Fig 6: A map and one seismic section through one of each of the three different types of chimney highlighting the properties of the chimney on seismic (again interpretation and labels should not obscure the raw data). For the chimneys with no clear base, show one with branching and the other with distortions

Fig 7: Three different maps highlighting each of the three anisotropic PF array types
you recognize in Tier 2 and one map of an isotropic PF array in Tier 2. Also show how the linear chimneys interact with each of the three anisotropic PF array types you describe in Tier 2 and one map showing how chimneys interact with isotropic PF array in Tier 2

Fig: 8: Three seismic lines showing the three different ways chimneys interact with polygonal fault planes

Fig 9: Something similar to your Fig 7b

Fig 10: Figure containing one map and one section in support of observations in 4.1.3

Fig 11: Maximum of two seismic lines illustrating the relationships described in section 5.1

Fig 12: Your Fig 9

Fig 13: Your Fig 10

Fig 14: Your Fig 11

Fig 15: Your Fig 12

This should reduce the number of figures from 19 (including your appendices) to 15, also ensure the remaining figures all serve a purpose and clearly illustrate what is described in the text. You will hopefully find doing this that the figures are less cluttered with less insets (as per the other reviewers review) and that these will support your very interesting text much better.

Please bear in mind the following hints when reproducing these figures:

(a) on amplitude maps, please ensure your color scale is centered on zero (so that it is clear that red/orange is negative and black/grey positive or vice versa) At the moment, for example on your existing figure 4C, it is impossible to tell what parts are anomalous (negative amplitude) and which are positive amplitude.

(b) Please ensure labels do not obscure the features of interest, and if something is traced onto a feature of interest (for example your existing figure 2b), that there is an uninterpreted seismic line beside it

Once these figures have been made, please then check the text very carefully to ensure that figure references in the text match these new figure numbers.

3. TYPOS / NOTES READING THE PAPER

Unless otherwise noted, replace the equivalent text with the text I give in "inverted commas"

P1 L2: "Angola. These features are termed "Linear Chimneys"."

P1 L3: "Hydrocarbon migration"

P1 L4: Remove "the" (second word in line)

P1 L7: Replace "e.g." with "such as"

P1 L11: "The initiation of polygonal faulting occurred 40 to 80 m"

P1 L12 "The majority of Linear Chimneys nucleated in the lower part of the PF tier below an impermeable layer within the tier. The filling of lower parts of the polygonal fault tier demonstrate the presence of pore space within the lower part of the tier. The PF gas traps restrict the leak points. . ." NOTE it is possible to have porosity / gas without significant permeability

P1 L17 ". . .polygonal faults coupled with. . ."

P2 L3 "flow directions in the subsurface and the distribution. . ."

P2 L5 ". . .structures formed during fluid leakage records the style. . ."

P2 L8: Replace "leakage" with "leak"

P2 L23: ". . . documented chimneys having elliptical cross-section and described the

planform ratio of these chimneys for the first time."

P2 L25 replace "orientations" with "orientation"

P2 L26: Reference in brackets

P2 L28 "... align parallel to these"

P2 L32 "However, neither factors that determined the linear planform of these chimneys nor the reason why gas charged fluid migrated into the PF tier have been investigated."

P3 L1 "It has been documented that"

P3 L5: "interactions between the orientation of magna fluid conduits and tectonic stresses. Nakamura established"

P3 L6: "...different tectonic regimes, noting, for instance, that aligned..."

P3 L11: Delete "Based on seismic observations"

P3 L12: "in shallow buried sediments based on seismic observations, thereby"

P3 L18-19: Statement not supported by fig. 1

P3 L19 "The seismic data has a"

P3 L21 "The dominant frequency is slightly"

P3 L24: Please describe which angles are covered by the near, mid and full stack

P3 L26: "rule out whether the studied features are"

P3 L29: "... to map the linear fluid venting structures as accurately as possible"

P3 L30 "are present on the near, mid and far offset volumes"

P4 L4: Figures cited do not support text

P4 L8: Add reference

P4 L14: Which figure in Ho 13?

P4 L20 Ho et al 2012a, fig 6a in this paper OR Fig 6a in Ho et al 2012a?

P4 L26 Replace "Relied" with "Relief"

P4 L27: Replace "Isopach" with "Constant"

P4 L 28: "Below which a large number of gas accumulations are interpreted" (also explain why these are interpreted to be gas accumulations)

P5 L2: Spell out that PHAAs are acoustically hard (increase in acoustic impedance) and NHAs are acoustically soft (decrease in acoustic impedance). Use PHAA and NHAA or PHA and NHA, don't use what you currently have (PHAA and NHA). Section 4.1.1 I struggled to find the observations in the text in the figures - some figure references may be wrong. See my separate comments on how to tidy up the figures to make things easier for the reader Section 4.1.2 I struggled to find the observations in the text in the figures - some figure references may be wrong. See my separate comments on how to tidy up the figures to make things easier for the reader

P7 L30: Do not see the described feature on the referenced figure

P8 L2: Same comment as P7 L30

P8 L19: Sentence does not make sense to me

P8 L30: Replace "Some might hypothesize" with "It could be argued" Section 5.2.2 Explain why strong soft anomalies are interpreted as gas. Has the seismic been balanced correctly?

P9 L28: "1) the presence of an"

P9 L33: Figure reference incorrect

P10 L14: Fig 6D does not exist

P12 L23: Sentence does not make sense

P13 L8: ". . .. Study area may be because" The notes which follow on the figures are some observations, these may or may not be relevant given my recommendation to reorder and rearrange the figures.

Figure 1: A multi-segment seismic line showing the labelled geometries would be helpful

Figure 2: Features of interest on the seismic line obscured by illustrations

Figure 3: "A few faults propagate above the Tiers-2 (interval A)": This is confusing, interval A is above Tier 1, not Tier 2

Figure 4: Please center amplitude maps on 0. It is difficult to tell which parts of the amplitude maps are negative or positive at the moment.

Figure 5: (A) (i)/(ii) and (iii) should be swapped around (show seismic line first, and then maps of given horizon). Likewise for figure 5B

Figure 6: Labelling of maps is confusing, please label the horizon used for figure 6b on figure 6c. Amplitude limits of inset to figure 6a are not labelled.

Figure 7: Figure caption does not match figure, check this.

Figure 8: Cannot see location of seismic line on figure 8b. Amplitude map should be centred on 0.

Figure 9: "Low permeable layer" should be "low permeability layer"

Figure 10: Poor grammar in figure caption, suggest rewriting. Not sure what middle block diagram between block diagram to left and seismic line to right adds in a, b and c?

Figure 11: I like this figure, but no caption for figure 11d.

Figure 12: I like this

Figure A1: It is good to have a summary map like this, however I cannot see the grey

area this study is based on.

Figure A2: This seems to be a key figure, not sure what it is doing in appendices. Ensure amplitude maps are centred on 0 so it is clear what is anomalous.

Figure A3: Would a reference to Ho 2013 suffice instead of reproducing this here?

Figure A4: This seems to be a key figure, not sure why it is in appendices. Ensure amplitudes centred on 0

Figure A5: Where is "appendix A"? Do you mean "Appendix 1"? This seems to be a key figure, why is it in appendices?

Figure A6: The amplitude map sin a(ii) shows PF cells filled by amplitude anomaly which IS INTERPRETED to represent gas fill. Ensure amplitude maps centred on 0. Figure A7: I like this, is this a key figure?

4. OTHER NOTES I have not had time to check the reference list for typos. Given the number of typos I have identified, I recommend that the paper be thoroughly proof-read prior to publication by a number of different people to ensure the paper is typo free. Given the figures need reordering, redoing and representing in a different order, I have not reviewed the figures or figure captions in any further detail other than listed above.

5. CONCLUSION I like the paper and feel it should be published, subject to the alterations I suggest above. I hope my comments are constructive and useful to the authors in helping them improve the quality of the presentation of the material.

---

## Referee Comment (RC3) · Anonymous Referee #3 · 2 Jul 2018

This paper investigates a new variety of chimneys which exhibit linear planform and are aligned parallel to adjacent anisotropic Polygonal Faults (PFs) and salt tectonic structures. This paper demonstrates that the formation of Linear Chimneys are as a result of hydrocarbon migration influenced by anisotropic stresses around the adjacent anisotropic PFs, whose developments were perturbed by the anisotropic stress field of salt tectonic structures. The genetic relationship between tectonic faults, stress perturbation, PFs, hydrocarbon migration and the geometry of hydrocarbon leakage indicators has been demonstrated with detailed illustrations. This paper is based on

research results in Ho's PhD thesis, and detailed analysis of chimneys and PFs on 3D seismic data used in exploration. This paper contains new concepts of hydrocarbon migration in polygonally faulted interval. The authors have nicely demonstrated the concrete role of PFs in hydrocarbon migration processes. They have put forth the conceptual model of preferential orientated PFs forming fault bound traps which control the location of hydrocarbon accumulation and overpressured gas leakage, and hence dictating the hydraulic fracturing point i.e. nucleation location of gas chimneys.

I am glad to see there has been such an advancement over what has been already published on this topic by Ho et al. (2013) & (2016). I believe that the authors must have put in an enormous amount of effort to develop this study, and the results in this paper are definitely worth to be published, however some improvements can be made, and the suggestions below could be helpful.

Paper structures The introduction is well covered especially for the historical part, however, for audiences who are not familiar with PFs, it is worth to mention briefly how PFs form and how PFs can serve as palaeo stress indicators. I believe that the latter has already been well investigated in the published works by the last author and so brief explanations and references to these works will be enough. Otherwise, later in the discussion section, it may appear too critical to jump directly into discussions of PF formational timing, although I like the way how the authors lay out and discuss the timing relationship between PFs and chimney developments.

Figures There are figures for each key observation that back up the hypothesis provided by the authors, but some times some figures are just too "busy" and make the reading difficult. The number of figures will not be an issue as soon as it is well presented and not overloaded on any single page. If it is unavoidable to show a complex set of sub figures then it will be better to separate them in two different figures. The three types of chimneys indicated by drawing in Fig. 2b should also be shown on seismic lines in a separate figure, with the corresponding drawing, before Fig. 4. For example, a simplified version of the catalogue of chimneys in Appendix 2 can be added

before Fig.4. For Fig. 4, it may need some re-organisation. The sub figures (a), (b), (d) can be one single figure, (c) and (e) can be put into a new figure. Reordering Fig. 4 and 5 could be considered. Type 1 chimney should be shown before Type 2 in the figures. If it is possible make a diagram or a table, etc, to integrate the three types of chimneys with Fig. 7a. The seismic line in 3D view in the far right of Fig. 7a is quite a representative figure, it could be shown earlier right after Fig. 3. The presentation of Figure 6 is ok, but it will be better to reorder the sub figures by labelling the insert in (a). Enlarge the figure into an entire page and it should make the reading easier. Appendix 3 and 5 could go into the main figures. Appendix 5 can be integrated in the geological section while appendix 3 can be put in the description of PF patterns.

Technical aspects There are few scientific questions which could use more clarity. 1) In the proposed conceptual model, the authors have suggested that gas could not migrate further upward the PF plane, the reason for that could simply be the regional seal retains the gas in the lower part of PF tier but has nothing to do with the permeability of PFs? Could it be purely some lithological effects, such as permeable layers occur rather in the lower part of the PF tier and layers in the upper tier layers are less permeable or impermeable? 2) If the more permeable (lower) part of PF tier juxtaposes against the impermeable (upper) part of PF tier, such as the model in Loseth et al. (2011), perhaps this could explain the formation of fault bound traps occurring particularly in the lower part of PF tier? 3) Is there any well data which can be used to indicate the lithology of PF tier? 4) In the statistics there are 7% of Linear Chimneys which are not intersecting with any PFs and occur in the middle of the tilted PF blocs. Is the conceptual model of impermeable faults intersecting impermeable layer to form fault bound trap still work for these chimneys? 5) Is the hypothesis of the upper part of fault is impermeable uniquely based on the number of chimneys emanated along the lower part of PFs? How if chimneys emanated along the upper part of fault (above the regional impermeable layer) or at the upper tip? In the statistics, there is only 1% of chimneys emanated from the upper tip of PFs, and no chimneys emanated along the upper part of PF plane, does it mean that among all the studied PFs are only 2 of them

permeable?

I was happy to read this study with many new insights and i hope that my comments and suggestions above are useful for improvements.

---

## Author Comment (AC1) · 30 Jul 2018

To Anonymous Referee #2

1. Author's response to general comments

The authors would like to take this opportunity to thank the reviewer's for their extensive review and helpful insights into the readability of the paper. The reviewer's recommendation of revising the structure of figures has been taken on board. Whilst we do not agree with all changes we acknowledge that figures and clarity in explanations and

reference to the figures is a common theme from all reviewers and detailed below are areas where we think a compromise can be met. We hope the response below meets the reviewer's requests and improves the overall manuscript. The grammar will be revised again by our British co-author and all other authors to ensure the overall quality.

2. Figures

Reviewer 2 - Fig 1: Simple map highlighting position of the two seismic surveys to the coastline and the wide geological environment described under "Regional Setting"

Authors – We take on board the reviewers comments and will make some changes to resolve the clarity issues. The main areas we will focus on is splitting and moving figures to convey single or a few points per figure to make the final message clearer.

The suggestion of showing the survey along the coastline will only double the volume of the figure number. Since we do not have the authorisation to show the exact location we have used a red star to indicate its approximate location. The details of the wider geological context is contained in the "Setting" section of the manuscript.

The purpose of Figure 1 is to show the distribution of fluid leakage structures, tectonic structures and the spatial relationship between both. As it does not affect the interpretation in this study, we regard it unnecessary to change Figure 1 or add an extra information pertaining to survey location.

Reviewer 2 - Fig 2: Sample seismic line on each dataset through a feature of interest, showing what this looks like on full, near, mid and far offset data, demonstrating the resolution of the data etc

Authors – We think that an image showing a feature of interest should be in the description section of the manuscript and should come after the geological setting.

For Figure 2 it is more appropriate to show a regional line across the survey in Figure 2, because it shows the main structures. Due to the image resolution of the regional line in Figure 2a, we have included an inset zoom-in to show the detailed stratigraphy

in Fig 2b.

We do not have authorisation to use data other than the near offset survey for publication. The far-stack data has been examined internally and the authors are confident that the chimneys are not artefacts. On this basis we think the near-offset data is adequate.

We will add representative seismic figures after the Stratigraphy section to show studied chimneys, as suggested by Reviewer 3.

Appendix 2, 4, 5, 7 will be modified and move into key figures as suggestion in the note and as the partial suggestion by Reviewer 3.

GEOLOGICAL SETTING

Reviewer 2 - Fig 3: A super-regional seismic line (if possible), or as regional line as you can manage across the two datasets with penciled in interpretation if data confi-dential, showing the rift, sag and passive margin sequence and geometry. Include an inset zooming into the interval of interest, highlighting the different units.

Authors – The study is focused on the Neogene succession within the localised 3D seismic volume. Whilst regional context is important the data available for the study does not allow production of a super-regional line and we regard the deep intervals, especially the rift and sag sequence has having no bearing on the interpretations here other than that the fluids inevitably come from depth. All of the deformation (structures) shown in this paper are either compaction related or are detached at the base of the Early Cretaceous evaporites as shown in figure 2.

Reviewer 2 - Fig 4: A map highlighting the features of interest (perhaps after your existing figure 1) with: 1) a multi-segment seismic line going through each of the synclines and diapirs. Use this seismic line to support statements made about timing of syncline subsidence etc 2) a short seismic line or lines (two maximum) highlighting the other features described in section 3.2

Authors – The intricate details pertaining to the timing of minibasin subsidence and salt diapirism are beyond the scope of this study however, in broader terms their timing is relevant to some of the interpretations. We feel the seismic line in Fig 2 is adequate to document the gross structural evolution of this part of the basin and we will look to add the necessary detail within the text. We will try to move Appendix 5 and put it before Figure 2b and 3.

OBSERVATIONS AND DISCUSSION

Reviewer 2 - Fig. 5: A seismic line highlighting a PHAA and NHAA. These should be clearly labeled and not covered by interpretation.

Authors – We will look to revise figures where this occurs. We are confused by this comment.

Reviewer 2 - Fig 6: A map and one seismic section through one of each of the three different types of chimney highlighting the properties of the chimney on seismic (again interpretation and labels should not obscure the raw data). For the chimneys with no clear base, show one with branching and the other with distortions.

Authors – An extra group image which shows different types of chimneys will be added after the figures int the Geological Setting section.

Reviewer 2 - Fig 7: Three different maps highlighting each of the three anisotropic PF array types you recognize in Tier 2 and one map of an isotropic PF array in Tier 2. Also show how the linear chimneys interact with each of the three anisotropic PF array types you describe in Tier 2 and one map showing how chimneys interact with isotropic PF array in Tier 2

Authors – We feel this is adequately shown on figure 1 and that the subsequent zoom in maps show the necessary detail in each geometry. Clear reference to each of the zoom in maps within the appropriate part of the text should be adequate for the reader to visualise these variations in more detail. A previous paper by the authors focuses

purely on the geometric variation and the reader will be referred to this paper within this manuscript.

Reviewer 2 - Fig: 8: Three seismic lines showing the three different ways chimneys interact with polygonal fault planes Reviewer 2 - Fig 9: Something similar to your Fig 7b

Authors – For the suggested organisation of Figure 8 and 9, we will split the Figure 7a and 7b into two.

Reviewer 2 - Fig 10: Figure containing one map and one section in support of observations in 4.1.3 The map for the section 4.1.3 (Figure 8), the dip map is showing the depressional geometry of the topmost termination of Linear Chimneys and the amplitude map is showing the PHAA developed along the strike of the depressional linear geometry. Reviewer 2 - Fig 11: Maximum of two seismic lines illustrating the relationships described in section 5.1

Authors – We will try to look at the figures and try to accommodate the suggestion.

Fig 12: Your Fig 9 Fig 13: Your Fig 10 Fig 14: Your Fig 11 Fig 15: Your Fig 12

Reviewer 2 - Please bear in mind the following hints when reproducing these figures: (a) on amplitude maps, please ensure your color scale is centered on zero (so that it is clear that red/orange is negative and black/grey positive or vice versa) At the moment, for example on your existing figure 4C, it is impossible to tell what parts are anomalous (negative amplitude) and which are positive amplitude.

Authors – We will move the "0" into the right place on the scale bar. We will add in the Methodology that, the red-orange colour indicates positive polarity and black-grey negative. The "0" on some of the amplitude scale bar had been displaced by the image editor when the images were shrink to fit on the page size.

Reviewer 2 - (b) Please ensure labels do not obscure the features of interest, and if something is traced onto a feature of interest (for example your existing figure 2b),

that there is an un-interpreted seismic line beside it. Once these figures have been made, please then check the text very carefully to ensure that figure references in the text match these new figure numbers.

Authors – We will look in to this matter and make sure it is not the case. Figure 2b is the unique case, all other figure's labels do not cover the area of interest. Symbols on the seismic line in Figure 2b are to represent stratigraphic positions of the studied features, because it is impossible to show all studied features on one single seismic section. We are going to improve Figure 2b as suggested by other reviewers as well.

3. TYPOS / NOTES READING THE PAPER

Reviewer 2 - Unless otherwise noted, replace the equivalent text with the text I give in "inverted commas"

Authors – The authors will ensure these corrections are made in the list of comments.

---

## Author Comment (AC2) · 30 Jul 2018

**Answers to comments of Reviewer 1 A. Plaza-Faverola**

**Answer Part 1**

(Due to the word limitation of the online system, the answers to comments are divided into two parts)

[Figure]

**1. Author's response to summary**

The authors would like to take this opportunity to thank the reviewer's for their extensive review and helpful insights into the readability of the paper and raising specific technical questions and queries. A major theme in the reviewer's comments is on clarity of explaining the key results, summarising the main themes and simplifying and/or clarifying figures and their connection to the text. The authors propose that the manuscript now go through a revision phase to address these broader issues and in doing so complying with the reviewer's general comments. Below are specific answers to the reviewer's questions. Review questions are in bold and the authors responses are below. Hopefully the authors have addressed the reviewer's concerns with adequate responses.

**2. Main points:**

**â ËŸA′c Polygonal faults: the core of the paper is the relation of shallow gas accumulations and chimneys with the presence of polygonal faults. Although the authors do reference key studies related to polygonal faults (identiïfied in the area but also globally), I think that a little bit more emphasis on describing the main aspects of such faulting is needed in the paper. I usually think of polygonal faults as those faults forming "polygons" in fine-grained sediment sequences. I realize that in this paper there are also radial faults associated with the synclines next to the salt diapirs that are as well referred to as polygonal? Are these termed polygonal faults because the mechanism of formation is similar to the other ones? How exactly do they form? Do they really reactivate?**

The reviewer's questions are all valid and the authors will work to address the wider issue of clarity with regard to faulting. The classification and debate on the genesis

of polygonal faults has evolved quite a bit over the last decade or so and with improvements in the resolution and coverage of 3D seismic data geoscientists have been offered atypical examples of the fault system to study and a chance to explore the finer details of these systems. One such detail is in the geometry which across a number of basins now show a departure from the classic "polygonal" pattern that has been observed previously. This has implications for understanding variations in the in-situ stress state at the time of faulting (expanded later).

At the time that polygonal faults were first mapped in 3D seismic data, for example in the North Sea (cf. Cartwright, 1994), the first question to resolve, was the origin of the faulting and the "polygonal" geometry was central to that discussion. Summarising; the presence of discrete tiers of normal faults in a passively subsiding, largely non-tectonic basin, was paradoxical since uniform 1-d compaction does not typically coincide with normal faulting. The "polygonal" geometry also suggested a non-tectonic setting as the wide range of fault strikes indicate horizontal stress isotropy which are typical of horizontal or only very gently dipping strata undergoing uniform compaction. Polygonal faults can also form where there is isotropic horizontal extension above a rising and widening, symmetrical dome but this setting clearly does not account for such vast systems in the North Sea and other places. It was later postulated that syneresis (shrinkage) caused volumetric contraction during burial and compaction of the host sediments which resulted in the next extension that produced the fault systems. Because this driver for faulting occurred in an environment of largely horizontal stress isotropy the faults developed as polygonal planforms. Other more sophisticated mechanisms have been put forward recently such as diagenetic processes. Nevertheless at the time of discovery, the name polygonal aptly described the geometry of the faults but as more systems were recognised their name was also referring to the mystery process that formed them.

There is no reason why the process generating polygonal faulting should be restricted to non-tectonic basins so what should happen in settings where there is a phase of reactivation, diapirism or gravity tectonics? One may expect that the horizontal stresses would not be isotropic in these settings and that compaction and contraction-related faults would form with preferred alignments. A PhD study undertaken by one of the authors (Carruthers, 2012) investigated two basins where such environments exist and put forward a comprehensive list of justifications for the presence of non-polygonal, a sort of hybrid "polygonal" faults. The obvious issues that arise from this work is the usage of terminology because most geoscientists are now familiar with the fault family "polygonal faults" but the examples known to date are increasingly non-polygonal. In this paper, we acknowledge previous work on polygonal faulting on timing, genesis and interpretation but will use the revision phase of the manuscript to clarify the terminology of polygonal faults (faults having formed during early compaction and contraction of very fine-grained sediments) but which have preferred alignments. Responding to the reviewers question regarding reactivation; there is no reason to treat reactivation of "polygonal" faults as different from any other fault systems. If a stress is applied to the fault is great enough to overcome the strength of the host-media and propagate the fault tips the fault will propagate. In some of the classic North Sea examples of polygonal faults, and even here, polygonal faults are organised in tiers. The nature of tier boundaries, especially the upper ones, are still not well understood. There have been some indications in some basins to argue for an upper mechanical boundary to upward fault propagation. Most of the time, polygonal faults define the seal package in petroleum basins and rarely return detailed well information from the interval, the few that have suggest the mechanical boundary hypothesis is at least not globally applicable. There is better evidence to indicate that the uppermost tier boundary is a paleo-seabed horizon at which upward propagation of polygonal faulting ceased but because most of the system occurs in non-tectonic basins were not reactivated expect in specific locations.

**Is it really meaningful to talk about foot wall or hanging wall if the faulted blocks can be one or the other depending on the reference fault plane?**

Hanging wall and footwall are basic properties of faults so we believe there is no reason why they are not relevant to this particular fault family. We use "graben, horst and tilted blocks" to describe the shapes of fault blocks, and use "hanging wall and footwall" to describe the direction of displacement of the fault plane. Hanging wall and footwall can co-exist in a single titled block; for example, the side of fault juxtaposes against the adjacent down shifted block is considered as footwall, while the opposite side of fault juxtaposes again the adjacent up lifted block is considered hanging wall. Our reference to hanging wall and footwall is with respect to a particular fault so we appreciate that when discussing a specific example we should make this clearer on figures. With reference to the statistical analysis we feel this is less relevant because our findings indicate that fluid migration is sensitive to the elevation of bedding across such small-scale fault blocks.

**They do not form subaerially right? Section 3.2 could concentrate all the relevant details about the types of polygonal faults observed here and what exactly are the main characteristics of such faults.**

The authors acknowledge the need for a summary and overview of the faults families and their characteristics at the outset of the paper. The authors will include the necessary additions during the revision phase of the manuscript.

Answering the specific technical question from the reviewer; polygonal faults form in the very shallow sub-surface during compaction and contraction of often very-fine grained hemipelagic sediments probably in base of slope (deep-water) settings. At key phases during the propagation of faulting (usually at the end of faulting) the upper tips were exposed at the seabed.

**â ËŸA′c Stress distribution at polygonal fault planes: The use of the principle of particle motion and the distribution of stress at each faulted block is great for**

[Figure]

**explaining migration pathways and leakage distribution. However, I wonder how meaningful is the terminology of footwall and hanging wall for polygonal faults. In cases where there are several parallel polygonal fault planes, one block may be the hanging wall with respect to a block on one side but it would be the foot-wall of the next fault. It is difïñĄcult to see that the shallow gas accumulations are exclusively at the lower part of the foot wall of polygonal faults (if I under-stood correctly the interpretation by the authors). I do see high amplitudes on what could be a hanging wall or a footwall depending on which fault segment for the polygonal set I choose. So I think it makes more sense to talk gener-ally about focused regions of higher shear stress and dilation depending on the relative motion of the faulted blocks with respect to each other.**

This question has been raised previously and answered previously. Blocks relative to fault plane – specific mention will be made within the text.

**In general, the discussion about stresses is hard to follow. I think it would help to use only one ïñĄgure to project all the relevant stress vectors inferred at the local zones of ïñĆuid leakage, together with regional stress vectors from, for example, salt-tectonics.**

Given the length of the manuscript at present we feel we can convey all the necessary points by better editing of text and fig captions rather than produce more figures. We feel the paper is already on the long side. We acknowledge the difficulties to navigate the section referenced by the reviewer and will look to improve.

**Where do the blue and red vectors in ïñĄgure 4 and 6 come from? Are these measured orientations of principal stresses or inferred?**

The stress ellipses are not absolute stress vectors. They are used to indicate the direction and relative changes in magnitude of the horizontal stresses (Sigma-2 and

[Figure]

Sigma-3). In fault systems produced during 1-d compaction and burial the largest compressive stress (Sigma-1) is vertical and the intermediate and least principal stresses are horizontal. The intermediate principal stress (Sigma-2) is parallel to fault strike. In classic examples of polygonal faults the wide-range of the fault strikes indicate over large areas there is no consistent direction of Sigma-2 suggesting overall horizontal stress isotropy. Where the same system of faults are clearly polarised in a dominant direction (such as radial or concentric around salt diapirs) the direction of Sigma-2 is predominantly in one direction.

The radial and concentric alignments of faulting here arise due to stress perturbations related to gravity salt tectonics but the dominant fault characteristics are the same as the layer-bound normal faults with polygonal geometries. We therefore, as explained earlier, consider them the same fault family and having formed by the same dominant process and at the same time but under the influence of local tectonic-driven stresses. This follows previous studies such as (Carruthers, 2013, Carruthers et al, 2013, and Ho et al, 2013).

The variations in faulting and thus changes horizontal stress isotropy to anisotropy are best shown on the regional base map which shows the most variations. We have thus taken the opportunity to show the stress ellipses here first.

We acknowledge that this could be better explained in the manuscript and on figures and will look to improve during the revision phase of the manuscript.

**Figure 10 says the vectors are local + in situ. What does in situ mean in this context, how are these stresses estimated? Information about regional and local stress fields are estmated in the region is missing.**

We address some of this question above. In situ refers to the stresses in the host-sediments at the moment of and during faulting. The geometry of polygonal faults offer a basic understanding of the dominant direction of the intermediate

compressive stress. Where fault strikes converge or diverge for example toward or away from a salt diapir it may be possible to indicate that there is a relative increase or decrease in the magnitude of the Sigma-2 but this is the limit to which this technique can be used to indicate horizontal stress variations. Once a polygonal fault has formed, the regional stress state may be further perturbed by the now formed fault plane, and the perturbed stress field around polygonal fault is what we refer as in-situ stress field. We will look to clarify all references to stress in the necessary sections.

**Not sure this is accurate, see comment above. Section 1, second paragraph. Also the figure captions should explain what those vectors signify.**

Agree we will change and clarify better.

**â ËŸAʹc Linear chimneys: Are the "linear chimneys" really chimneys? What is the definition of chimney used here? Aren't these features fractures/small scale fault planes where the fiuids literally escaped through and formed authigenic carbonate that together with trapped gas in the system creates the blanking in the seismic?**

The authors feel a comprehensive classification and overview of terminology and definitions is already present in the manuscript – see specific references below. However, we will ensure this information is explained as clear as possible in the revision phase of the manuscript.

In section 4.1.1 we have described the seismic characters of chimneys while in section 5.2 we have given geological interpretation of these chimneys.

In section 4.1.1. we described that on the study seismic survey, chimneys are represented by: 1) stacked of high amplitude anomalies (either positive or negative or both), 2) pull up reflections and 3) push down reflections. These three types of features can

be "or not" overlaid by positive high amplitude anomalies which were interpreted in the discussion section as authigenic carbonates.

In section 5.2. we interpreted these chimneys as hydraulic fractures created by over-pressured fluid.

Since there are previous studies which have well studied the basic characters of chimneys so we did not repeat the discussion in this paper.

**Is there evidence of breciation, or hydrofracturing in the regions interpreted as linear chimneys?**

The relevance of the question is uncertain but we assume it refers to establishing evidence of overpressure and hydro-fracturing rather than migration through faults.

The vertical stacks of high amplitude anomalies showing a linear plan form (either with positive or negative polarity, interpreted here as seep carbonates or free gas respectively) occur at different stratigraphic levels along the same vertical axes. This supports the existence of vertical fluid migration zones and hence the Linear Chimneys.

These vertical stacks of linear fluid venting systems cross multiple stratigraphic intervals and are taller than the polygonal fault tiers suggesting these systems are not simply migrating through faults. Furthermore, their vertical form is quite different from the 40-60 degree dip range typical of the polygonal fault system (and other extensional faults here). There vertical forms are best explained by overpressure and development of linear chimneys influenced by stress anisotropy around polygonal faults.

**Actually, the illustration of the chimney features is not that great in the figures. And this brings me to the next concern. In the data section the authors mention that different stacks were produced grouping angle of incidence.**

**It is indicated that the seismic data presented in the paper is from the near offset**

**stack. The seismic profiles shown in figures 3 and 4 are from which stack version? The character of what is indicated as chimney in panel e of figure 4 reminds me how gas chimneys look in undershooting data, namely along a stack using just a selected range of offsets. Two vertical blank regions appear to each side of a high amplitude zone; the separation between the two vertical blank regions increases with depth. The feature appears then as a double line on the maps. This is a bit confusing for a reader that is not familiar with the processing of the data. I wonder whether showing the character of the chimney on the full stack is more intuitive and straight forward. If the double pattern of the so called linear chimney is not due to processing, it would be interesting to her what causes such a particular feature (figure 4 a and b).**

All images shown in this manuscript are from multi-channel, post-stack, near-offset seismic data and were processed by the geophysical department in Total S.A. There is not any undershooting data used in this study.

In the methodology we mentioned that we also checked the presence of chimneys in the middle and far-offset surveys (all of them have different incident angles) to verify the authenticity of the studied chimneys. Since the near-offset survey best illustrates the chimneys we have used this survey in this project within Total S.A. Use of the full-stack data was beyond the scope and contract conditions of this study.

We agree that it is not easy to visualise Linear Chimneys that are parallel to PFs within the PF network due to the figure resolution in Figure 4a. That was the reason why we used Figure 4b to show the planform geometry Linear Chimneys. Regardless, the branching shadows below the chimneys are artefacts. We agree that it is can be confusing for readers who are not familiar with seismic processing. In order to show the planform geometry of Linear Chimneys, we have mapped high amplitude reflections which cross the body of bright spot chimney columns, and have superimposed it on a dip map of PF (see Figure I here).

**âËŸA′c The PHAAs are interpreted as carbonates. Why would they always be associated with depressions rather than mounds? By analogy with carbonate mounds at present day seafloor, buried authigenic carbonate concretions within chimneys can get buried and appear as mounded features in the middle of a cavity with onlap of reflections at the flanks. The interpreted buried carbonate concretions don't show positive morphologies in this case? Can it be related to the resolution?**

Carbonate mounds with onlap reflections on their flank do exist in this study area but they are not associated with linear planform chimneys.

We did not go into detail about the geometry of the PHAAs at the topmost termination of Linear Chimneys as it does affect the interpretation for Linear Chimney's formation and is beyond the scope of this study.

PHAAs in outcrops can take a variety of forms and shapes: nodules, slabs, cements, mounds, etc. However, the PHAAs that are associated with Linear Chimneys in this study, their geometry is not visible within single seismic reflection so we are not able to confirm which shape these PHAAs have (i.e. they are below the vertical resolution of the seismic data). Depressions at the topmost termination of Linear Chimneys, in association with PHAAs or not, are visible on the modern seafloor in some cases (fig. 5b) and no visible carbonate mounds have been observed above Linear Chimneys.

We suggest the reason why Linear Chimneys terminate upward into PHAAs with negative depressions is because the chimney exit was eroded by out-coming gas. The gas concentration or volume was insufficient to allow carbonate mounds development but nodule or cements (Blouet et al., 2017). So we think that the lack of carbonate mounds within depressions was likely related to the intensity of methane flux rather than seismic resolution.

**âËŸA′c The figures are of high quality. However, even if they are over loaded**

**with insets, some times they lack explanations of features that seem relevant for the interpretation. For example, panel b in figure 4 shows a white band braking through a high amplitude reflection. What is that feature? It is really hard to link all the different insets. In figure 5 I don't manage to identify where is the feature pointed with a yellow arrow on 5ai, on 5aiii. Is it correct that the seismic is NW to the right? It is important to find a way of simplifying these figures. Figure 5 could be split into 2 figures. I would suggest selecting 7-8 figures to show the main observations and to illustrate the conceptual models. Despite all the figures in the main text the authors still refer to appendix figures for observations that are key for the paper. The figures in the main text could be used in a more efficient way.**

We will improve the organisation of figures in the revised version of the manuscript.

(For continuation please read Part 2)

———————————————

[Figure]

[Figure]

**Fig. 1.**

---

## Author Comment (AC3) · 30 Jul 2018

**Answers to comments of Reviewer 1 A. Plaza-Faverola**

**Answer Part 2**

Reviewer's questions are in bold and the authors responses are below.

[Figure]

**3. Notes**

**Please fi̧nd below my notes while reading through the text; it includes a few typos. The L refers to the paragraph number and the P to the page number.**

The authors present replies to the specific technical questions and targeted questions only. Reports of mislabelling and grammar will be addressed automatically the revision process of the manuscript.

**- L20/P3: minimum and maximum offsets? I assume these data sets are multi channel with long offsets? It is kind of important to provide this info before it is mentioned that the amplitudes vs. angle were used for verifying the seismic character of the observed features (a sort of undershooting?).**

Addressed previously - see above.

**- L5/P6: The linear features shown in fi̧gure 4a-b are indeed strange features. Are these really along polygonal faults? (PFs?). Polygonal faults usually do not have a preferred orientation, but on the contrary, they consist of fault segments oriented covering the whole azimuth range (closing polygons), right? The linear features seem to follow the circular structures to the north of the syncline. There seems to be an overarching control on the orientation and distribution of these features rather than the polygonal faults as such. I guess I am missing a clear defi̧nition of what the authors are referring to as polygonal faults. For examples, are types 1-3 described by the authors termed polygonal faults because they all formed due to dewatering of fi̧ne-grained sediments? See main comment.**

Question already answered above.

[Figure]

**- L30/P6: Consider making two different figures here. The figures have so many parts that it is actually a struggle to go through and understand everything. Is the statistical part of the figure really relevant? What do we do with the fact that 54 percent are intersecting the lower portion of PF? What seems most relevant in this paper is to get compared the orientation of the fluid flow related features with respect to the orientation local and regional faults and fractures, right?**

We will split the figure into two parts.

This study focuses on the formation of linear planform chimneys, is a detailed seismic study and is not a paper which only describes the parallel relationship between Linear Chimneys and PFs. The statistic is important in this study, we focus on the reason why most of the chimneys emanated from the foot wall area of a fault (i.e. topmost point of a same layer in the up-thrown block).

We do not simply focus on the orientation of Linear Chimneys in comparison with PFs, because the nucleation of Linear Chimney in the up-thrown block caused the linear planform of chimneys and hence the parallelism with PFs. Because there are 54/The distribution of gas among PF blocks, nucleation points of chimneys, the propagation of chimneys related to stress around faults, and the orientation of chimneys and PFs are all interlinked.

**- L20/P7: Is the evidence by Sonnenberg et al., 2016 related to the polygonal faults in this present study? In that case, it would help to see a sentence hinting what is the observation that works as evidence. I got this advice recently and I kind of see now the need for bringing into the current study the key observations rather than referring the reader too often to the previous studies. This degrades the flow of the reading and makes difiňĄcult to follow the paper.**

The reference to Sonnenberg et al,. 2016 was an example reference to remind the reader that polygonal faulting pre-dates chimney development. We acknowledge the reviewers point and will simply relate to the previous section documenting polygonal fault timing. Some expanded comments on polygonal fault time are below.

Due to the size of polygonal faults unambiguous evidence of fault timing is rarely preserved and this is no different to this study. The throw of these small faults is close to the vertical resolution of the seismic data meaning standard techniques such as upper tip gradients and thickness changes across the footwall and hanging wall are not clear cut for advocating if the upper portion of the polygonal faults here are definitively growth rather than blind faults. However, a collection of observations from the literature in addition to the compaction and contraction process which forms the faults all point to an evolution of faulting during early burial with a late phase of growth faulting ceasing at a paleo-seabed (upper tier boundary).

Specific examples across multiple basins suggest polygonal faulting may having this late growth phase in common. The evidences are bullet pointed below. The authors do not feel it is within the scope to cover all of these points explicitly in this manuscript and that simple summary and reference to the literature is adequate.

- Polygonal faulting in London Clay not below a few hundred metres of burial today (Henriet et al., 1982).
- Polygonal faults observed tipping out present seabed in Lake Superior (Berkson et al., 1973, Wattrus et al., 2003).
- Unambiguous growth sequences observed in some large examples of polygonal faults (Carruthers, 2012).
- High upper tip gradients inconsistent with blind faulting (Carruthers, 2012).
- Limited evidence of upper tier boundaries being mechanical barriers to upward propagation (e.g. Laurent et al., 2012, Berndt et al., 2012).
- In many examples upper fault tips occur at a single seismic horizon in systems

which span several hundred sq km and this is best explained with the horizon being a paleo-seabed at which faulting ceased (e.g. Berndt et al., 2012, Christopher et al., 2014).

- Fault tiers very thin in some cases meaning if the upper tier boundaries are paleo-seabed horizons then faulting initiates soon after burial (e.g. PFs in the pinch out of Tier-2's wedge shown in Appendix 5).

**- L30/P7: do polygonal faults really reactivate? How is the accommodation of such movement if the fault planes can converge to each other rather than been parallel? Aren't these kind of faults associated with diagenetic processes and are hence a kind of one-time event?**

This point has been discussed from a general perspective earlier. In this specific example, the horst which has developed onlaps in the section above the upper tier boundary (possible reactivation post-dating polygonal fault development) has a concentric geometry. The concentric geometry is most likely due to the process of pockmark development locally perturbing the horizontal stresses in a way to produce concentric fault strikes (Carruthers, 2012, Ho et al., 2013). The pockmark clearly formed during early tier sedimentation but differential compaction after polygonal faulting above the pockmark may have caused this specific fault to continue forming. Due to the points raised previously on polygonal fault genesis and timing this example shows some specific evidence in this study that even a reactivated polygonal fault still occurred before the chimney formation. All of the other polygonal faults had already formed prior to this phase of reactivation.

**- L20/P8: 60 m deep and 4.5 m wide pockmark??? That is quite deep compare to the with, it is almost a conduit rather than a pockmark.**

We have made a typo here for the unit here, it should be 60cm depth.

**- L30/P8: so there is active gas release at the seafioor at present? Or you mean active in the sense that there is gas filling the near-surface systems through the gas chimney structures?**

When we refer to the chimneys being active, we mean there are still fluid activities on going at the present day or in the very past representing deposition of the seabed surface . The fact that there are interpreted seep carbonates inside the depressions at the top of chimneys indicated the recent activities, although we cannot confirm whether these chimneys represent activity at this present moment because we do not monitor them.

**- L30/P8: this model is hard to digest here since there are so many faults and pre-existing weak planes that one would think that the fiuids would find preferential path- ways without much effort and hence gas chimneys would not be favor?**

The argument that high-density faulting would allow enough fluid migration without chimney formation is wholly dependent on the assumption that the faults are migration pathways. Whilst faults often do provide leakage pathways the extensive literature on fault seal indicate it is far more complicated. In some cases polygonal faults have been shown to be migration pathways whilst in other cases polygonal faults deform important seal packages in many petroleum basins (Carruthers, 2012). We do not feel this assumption is valid because all the available evidence in this study suggests there have been phases of overpressure and fluid venting from intervals within the lower-middle stratigraphic levels of the polygonal fault tier.

A discussion this particular topic in presented in section 5.2.4.

**- L20/P9: The use of appendix figures to illustrate what seems to be the main conceptual model of the paper is not ideal. One figure in the main text should be enough to illustrate description of the model for fluid migration and development of chimneys. Figure 8a doesn't illustrate this, or did I miss something? When you mention PF tier here is it 1 or 2? It is very easy to get lost while reading, I think it is due to the fact that there are so many figures overloaded with details.**

Agree and will be changed into main body.

**- L25/9: typo: the PRESENCE of. . .And please revise this paragraph. These aspects are not necessarily ruling out each other and a combination of them could be a pre-condition for explaining your observations. Consider reformulating the paragraph.**

OK

**- L30/9: where do we see the gas accumulating at the foot wall? The foot wall of a major tectonic fault or are you referring to small faulted compartments resulting from the polygonal faulting? If so, is it really meaningful to talk about footwall if the blocks are somehow both footwalls and hanging walls with respect to each other?**

We will improve the connection between text and figures.

**- L5-10/P10: Why would these areas be subjected to "relative" compressional strain? What do the authors mean here? Would it be more appropriate to say "less" com- pressional state rather than relative? In analogy with the particle motion maps for earthquakes (focal mechanisms) one would expect that the**

**lower part of the hanging wall and the upper part of the footwall would experience more compression while di- lation would dominate the upper hanging wall and the lower footwall (which is indeed consistent with Barnett et al., )**

We will remove relative.

**- L10/P10: again, no iňĄgure 6d and also the interpretation that the gas is accumulated in the footwall of polygonal faults is not easy to digest. A block can be considered a footwall or a hanging wall, depending on which fault plane is used as reference. I can see high amplitudes in both hanging and footwalls in iňĄgure 6c for example.**

Point already raised and addressed in previous replies. Similarly we will improve the clarity of statements and figures.

**- L20/P12: Again, the model of shear stress distribution through the four quadrants of the faulted blocks is very sounding for explaining the distribution of gas into more permeable zones. However, where these more permeable zones are entirely correlated with footwalls and hanging walls in these polygonal fault system is hard to assimilate. Is it really necessary to use this terminology? See main comment.**

Point already addressed above.

**- L25/P12: So the chimneys grow episodically? You foresee that the growth occurred in several episodes of reactivation of the system? It is important to describe this more explicitly in order to be able of comparing to systems from other margins with comparable settings.**

No the authors did not indicate that the chimneys grow episodically. Last point beyond

the scope of the paper.

**- L25/P14: point 6 in the discussion. Check the grammar here. The sentence has a problem. When are the chimneys circular in isotropic stress fields? And when are they linear, isotropic and anisotropic? Clarify.**

We will check the grammar. Chimneys are circular while they are in isotropic stress fields; they are linear when they occur in anisotropic PF network where stress field is anisotropic.

**- Figure 1: great figure. The use of a dip map to show all the elements of the study works extremely well, we can see the flanks of the salt domes and even the fine scale faults and fractures. However, isn't the present day bathymetry important to understand the stress regime?**

We expand briefly on the previous replies to the topic of polygonal faults and their ability to give indications of paleo-stress anisotropy. Improvements in the clarity within the revised manuscript on this topic will hopefully improve the readers understanding of stress throughout. Answering the reviewed specific question here; polygonal faulting represents a brief phase of the basins compaction history and any stress anisotropy that was imparted on the compacted sediments during faulting are reflected in alignment and variety of fault strikes in that tier. The polygonal fault geometry and variations in geometry thus reflect an insight into the paleo-stress state at the time of tier sedimentation. The present day bathymetry and of units above the tier in question may give some broad insight into the structural development of the basin which may in turn have some similarities with the polygonal fault tier. However, the isopach and paleo-bathymetry of the PF tier are more important for understanding what was driving the stress anisotropy and thus the resultant polygonal fault geometry. Carruthers, 2013 discussed this in detail. The authors feel the question of the specific drivers of

the stress anisotropy are beyond the scope of this paper.

**- Figure 2b – typo: linear positive high amplitude. . .It is a bit difficult to read through the symbols of this figure. The idea of overlapping the symbols related to different fluid flow features on the seismic profile is great. However, it is not easy to see the actual seismic feature (in particular the high amplitude anomaly depressional network). I assume the location of the symbols just indicate the interval where each feature is observed rather than the actual feature? Maybe this can be hinted in the caption (as a help for the reader). Is the seismic section a compressed version of the geological section in 2a or C8only part of that transect?**

Improvements to be made during revision phase of manuscript.

**- Figure 5: Can you really tell that the high amplitude analogies in inset ii are gas accumulations, without any clear sign of polarity change?**

Error - it should be water-saturated sand

**References**

Berkson, J. M. and Clay, C.: Possible syneresis origin of valleys on the floor of Lake Superior, Nature, 245, 89-91, 1973.

Berndt, C., Jacobs, C., Evans, A., Gay, A., Elliott, G., Long, D., and Hitchen, K.: Kilometre-scale polygonal seabed depressions in the Hatton Basin, NE Atlantic Ocean: Constraints on the origin of polygonal faulting, Marine Geology, 332, 126-133, 2012.

[Figure]

Blouet, J.-P., Imbert, P., and Foubert, A.: Mechanisms of biogenic gas migration revealed by seep carbonate paragenesis, Panoche Hills, California, AAPG Bulletin, 101, 1309-1340, 2017.

Carruthers, T.: Interaction of polygonal fault systems with salt diapirs, doctoral dissertion, Cardiff University, Cardiff, UK, 489 pp., 2012.

Carruthers, D., Cartwright, J., Jackson, M. P., and Schutjens, P.: Origin and timing of layer-bound radial faulting around North Sea salt stocks: New insights into the evolving stress state around rising diapirs, Marine and Petroleum Geology, 48, 130-148, 2013.

Cartwright, J.: Episodic basin-wide fluid expulsion from geopressured shale sequences in the North Sea basin. Geology, 22, 447-450, 1994.

Jackson, C. A. L., Carruthers, D. T., Mahlo, S. N., and Briggs, O.: Can polygonal faults help locate deep-water reservoirs?, AAPG Bulletin, 98, 1717-1738, 2014.

Henriet, J.-P., D'olier, B., Auffret, J., and Andersen, H.: Seismic tracking of geological hazards related to clay tectonics in the Southern Bight of the North Sea, IZWO Collected Reprints, 12, 1982.

Ho, S., Carruthers, T., Imbert, P., and Cartwright, J.: Spatial Variations in Geometries of Polygonal Faults Due to Stress Perturbations and Interplay with Fluid Venting Features, in: Proceeding of 75th EAGE Conference and Exhibition incorporating SPE EUROPEC 2013, London, UK, 10 - 13 June 2013, 1-6, 2013.

Laurent, D., Gay, A., Baudon, C., Berndt, C., Soliva, R., Planke, S., Mourgues, R., Lacaze, S., Pauget, F., and Mangue, M.: High-resolution architecture of a polygonal fault interval inferred from geomodel applied to 3D seismic data from the Gjallar Ridge, Vøring Basin, Offshore Norway, Marine Geology, 332, 134-151, 2012.

Sonnenberg, S., Underwood, D., Peterson, M., Finley, E., Kernan, N., and Harris, A.: Polygonal faults, Niobrara Formation, Denver Basin, in: Proceeding of AAPG Annual Convention and Exhibition, Calgary, Canada, June 19-22 2016, 51311, 2016.

Wattrus, N. J., D. E. Rausch, and J. Cartwright: Soft-sediment deformation in Lake Superior: evidence for an immature Polygonal Fault System?, Geological Society, London, Special Publications, 216.1, 323-334, 2003.
* * *

---

## Author Comment (AC4) · 31 Jul 2018

**Author's response to general comments**

We kindly appreciate the time and effort of Reviewer 3 in reviewing the manuscript and the technical expertise as well as questions that have been raised. We are particularly thankful for the suggestions which bring new insights into improving the scientific discussion of the gas traps. We will revise the manuscript by including these

suggestions.

Reviewer 3's questions and comments are in bold and are followed by authors comments below.

**1) In the proposed conceptual model, the authors have suggested that gas could not mi-grate further upward the PF plane, the reason for that could simply be the regional seal retains the gas in the lower part of PF tier but has nothing to do with the permeability of PFs? Could it be purely some lithological effects, such as permeable layers occur rather in the lower part of the PF tier and layers in the upper tier layers are less permeable or impermeable?**

Total S.A.'s internal well report describes that the lithology in the upper and lower part of Tier 2 does not have a significant difference in lithology. We do not have direct access to the physical data so we have not been able to assess the possibility of preferential deposition of permeable lithologies in the lower part of Tier-2.

The reviewers point is certainly a valid possibility and accordingly we consider it in addition to the impermeable upper polygonal fault model.

In every scenario, it does not affect the interpretation and the formational model of chimneys, as overpressured fluid is suggested to not be able to escape through the upper part of PF plane, and that the upper PF must be impermeable.

**2) If the more permeable (lower) part of PF tier juxtaposes against the impermeable (upper) part of PF tier, such as the model in Loseth et al. (2011), perhaps this could explain the formation of fault bound traps occurring particularly in the lower part of PF tier?**

The reviewer raises and important point which naturally leads on from the question above. The authors acknowledge this possibility and will incorporate as such into the discussion along side our model.

Given that we do not know the permeability characteristics of the host sediments throughout the tier it is still possible that they are permeable to some degree. Therefore at the very least the upper part of the polygonal fault must be impermeable. Whether this is due to the upper tier sediment properties or not we cannot determine.

**3) Is there any well data which can be used to indicate the lithology of PF tier?**

Please see answer to question 1.

**4) In the statistics there are 7 percent of Linear Chimneys which are not intersecting with any PFs and occur in the middle of the tilted PF blocks. Is the conceptual model of impermeable faults intersecting impermeable layer to form fault bound trap still work for these chimneys?**

This is a very interesting question. Because of the length of the paper and because of the majority of chimneys stemmed from the intersection between PFs and impermeable barrier, therefore we did not explain the minority cases. However, we will address this particular point and add brief interpretation in the revised version of the manuscript.

**5) Is the hypothesis of the upper part of fault is impermeable uniquely based on the number of chimneys emanated along the lower part of PFs?**

[Figure]

Yes, since the vast majority of chimneys emanated from the lower part of PF planes, it is a powerful indicator to the permeability of PFs.

The method of counting the position of chimneys along PFs to estimate the permeability of PF was developed during Ho's 2013 thesis independent of academic supervision.

**6) How if chimneys emanated along the upper part of fault (above the regional impermeable layer) or at the upper tip?**

In the statistics, there is only 1 percent of chimneys emanated from the upper tip of PFs, and no chimneys emanated along the upper part of PF plane, does it mean that among all the studied PFs are only 2 of them permeable?

Yes, as confirmed by our statistics which is consistent with the counting in Ho, 2013. There is "only" 2 chimneys at the upper tip of PF, and all other chimneys do not emanate above the middle level of Tier 2, so we interpret that there are only 2 PFs which were possibly served as fluid migration pathways.

If there were chimneys rooting along the upper part of PFs it means fluid migrated up to their rooting points then broke through at such locations.
We have previously published in a peer reviewed journal the method of using the intersecting position of chimneys and PFs to investigate the permeability of PFs (see Ho et al., 2016).

We would like to take this opportunity to provide some supplementary information.

The interpretation and conceptual model of "fluid did not use the upper portion

of PF to migrate" and "intersection between permeable layer and polygonal fault plane controlling the nucleation of chimney" was proposed in the thesis Ho, 2013. The statistics presented in this manuscript is taken from the thesis has not been altered, and is consistent with the proposed conceptual model – 99% of chimney emanated along the lower part of PFs.

**Author's response to the comments on figures**

We will incorporate the suggestions to improve the figures in the revised version of the manuscript.

**Reference**

Ho, S.: Evolution of complex vertical successions of fluid venting systems during continental margin sedimentation, doctoral dissertion, Cardiff University Cardiff, UK, 2013.
Ho, S., Carruthers, D., and Imbert, P.: Insights into the permeability of polygonal faults from their intersection geometries with Linear Chimneys: a case study from the Lower Congo Basin, Carnets Geol., 16, 17, 2016.

---

## Author Response (AR1)

Dear Topic Editor of Sedimentology section, Prof. Samankasou,

First and foremost, we would like to thank you and the reviewers for the time and efforts that all of you have spent on reading and reviewing our manuscript.

We have now updated a revised version of the manuscript in which all of the reviewer's comments have been accepted or addressed.

- Technical questions on fluid migration, and alternative hypothesis on the formation of Linear Chimneys that have been brought up by Reviewer 3, have now been discussed in the manuscript in sections 5.2.3 and 5.2.4.

- A clear classification on the different types of Linear Chimneys requested by Reviewer 3 and 2 has been added as Figure 8.

- Background information about polygonal faults that has been requested by Reviewer 1 and 3, has been added in Introduction and section 3.2.3.

- Figure reordering as requested by Reviewer 2 and 3 has been updated.

- Clarification for figure citations in the main text requested by Reviewer 2 has been added.

- Figure simplification and modifications as requested by all 3 reviewers have been updated.

- A list describing the purpose for the use of figures is included in the list of corrections for both Reviewers 2 and 3.

- English language in the manuscript has been revised and corrected by our native British co-author as well as by a native American English language editor to ensure the international English writing standard.

We have addressed all the valuable comments provided by our three reviewers, and we hope that have fullfilled all the requirements, the updated and revised manuscript should now meet the publication standard.

Let us thank you again,
On the behalf of all coauthors
Yours sincerely,
Sutieng HO

**List of corrections and accepted suggestions by authors regarding the comments of Reviewer 1: A. Plaza-Faverola**

All suggestions and questions by Reviewer 1 are accepted and answered.

**Summary of corrections**

As requested by Reviewer 1, we have added background information to introduce what is polygonal faults (PFs), describing the main aspects of PF and the use of PF as stress indicators. We have added more clarity and definitions for terminologies in the Introduction as well as section 3.2.3.

We have also simplified figures and split them up to improve the illustration of Linear Chimneys as requested by Reviewer 1.

**Main points:**

The original points of Reviewer 1 are in bold, while the modifications made by the authors are described below each point.

**About the formation of polygonal faults and their difference from radial faults**

The following information has been added into the Introduction section:

*"Polygonal faults are considered as non-tectonic fault systems arising due to compactional-dewatering of very fine-grained sediments during the early stages of burial in passively subsiding sedimentary basins (Henriet et al., 1998).  In the classic examples of these fault systems which show "polygonal" fault arrangements and also contribute to their nomenclature, they were characterized by very small differences between the horizontal principle stresses during their formation (Cartwright, 1994; Carruthers et al., 2013). The examples of polygonal faults in this case study show substantial departures from this classic "polygonal" fault pattern (so called isotropic PFs) to very polarized fault arrangements (so*

*called anisotropic PFs) where the tier is deformed by salt tectonic structures or offset by their associated fault systems (Fig. 1; Carruthers, 2012). These faults can display a variety of intricate patterns ranging from tight radial systems around salt diapir to concentric systems within salt withdrawal basins and spiraling concentric patterns above buried pockmarks (Stewart, 2006; Ho et al., 2013). The preferentially aligned faults are many times longer than the regular faults segments with polygonal alignments but are often still confined to the same "tiers". The observations are consistent with a number of other reported examples of preferred fault alignments within networks of polygonal faults (Stewart, 2006; Ghalayini et al., 2016). The preferred fault alignments are indicative of horizontal stress anisotropy at the time of their formation (Carruthers et al., 2013)."*

See also authors online answers posted on 31[st] July 2018

**Regarding foot wall or hanging wall of the reference fault plane**

The following information has been added into section 3.2.3.:

*""Lower footwall", when not specified, refers to the lower part of tilted PF blocks immediately adjacent to the fault which moved upwards, or referencing the lower part of horsts in this study area.*
*"Lower hanging wall", when not specified refers to the lower part of PF graben."*

See also authors online answers posted on 31st July 2018

**Suggestion about Section 3.2 for adding all the relevant details about the types of polygonal faults**

Suggestion accepted. See answer above.
Information added in Introduction and section 3.2.3 in the revised manuscript.
See also authors online answers posted on 31st July 2018

**In general, the discussion about stresses is hard to follow. I think it would help to use only one figure to project all the relevant stress vectors inferred at the local zones of fluid leakage, together with regional stress vectors from, for example, salt-tectonics.**

Suggestion accepted, Figure 1 has included all necessary stress vectors in the revised manuscript.

**Where do the blue and red vectors in figure 4 and 6 come from? Are these measured orientations of principal stresses or inferred?**

The following information has been added into section 3.2.3.:

*"Preferred fault alignments or "anisotropic fault patterns" within polygonal fault networks have been observed in this study area. Concentric faults surround pockmarks (see fig. 2a in Ho et al., 2013) and are parallel to extensional synclinal faults (red dotted lines in Fig. S3 c; Appendix PF patterns b). Radial faults occur around salt diapirs (Appendix PF patterns c) (Carruthers, 2012) whilst ladder-like fault patterns occur in the center of concentric fault patterns above Syncline-2 (Fig. 6a-b, Appendix 3d).*

*The orientations of the PFs around or above the aforementioned tectonic structures are not unusual as the fault patterns mantle the expected stress state of the structures (Carruthers, 2012; Carruthers et al, 2013). The direction of maximum horizontal stress around the tectonic structures is indicated by the first-order anisotropic PFs while the horizontal minimum stress is indicated by the second-order anisotropic PFs (e.g. stress ellipses in Fig. 1), and hence different PF patterns are considered as indicators of stress state in the host sediments (Carruthers, 2012; Carruthers et al, 2013). "*

See also authors online answers posted on 31st July 2018

**Figure 10 says the vectors are local + in situ. What does in situ mean in this context, how are these stresses estimated? Information about regional and local stress fields are estmated in the region is missing.**

The following information has been added into section 3.2.3.:

*"The orientations of the PFs around or above the aforementioned tectonic structures are not unusual as the fault patterns mantle the expected stress state of the structures (Carruthers, 2012; Carruthers et al, 2013). The direction of maximum horizontal stress around the tectonic structures is indicated by the first-order anisotropic PFs while the horizontal minimum stress is indicated by the second-order anisotropic PFs (e.g. stress ellipses in Fig. 1), and hence different PF patterns are considered as indicators of stress state in the host sediments (Carruthers, 2012; Carruthers et al, 2013)."*

*""Regional stress" refers to stress states in the sub-surface driven by the primary tectonic forces which include gravity and the lateral extension and contraction occurring above the regional salt detachment.*

*"Local stress" refers to stress state at the scale and within close proximity of individual tectonic structures where the regional stress field may be locally perturbed.*

*"In-situ stress" refers to stress conditions in-place at the location of individual polygonal faults, this is particularly relevant when trying to understand the stress conditions at sites of incipient hydraulic fracture developments which lead to the formation of chimneys. "*

See also authors online answers posted on 31st July 2018

**Not sure this is accurate, see comment above. Section 1, second paragraph. Also the figure captions should explain what those vectors signify.**

Suggestion accepted.

The following information has been added into figure captions:

*"Palaeo-stress ellipses show relative directions and magnitudes of the horizontal principal stresses and are constructed from the planform geometry of the polygonal fault networks (see section 3.2.3 for more information). Blue axis and red axes on stress ellipses indicate the palaeo-orientation of the intermediate and minimum stresses, respectively. Location of seismic survey is indicated by a red star on insert map."*

See also authors online answers posted on 31st July 2018

**â ̆A ́c Linear chimneys: Are the "linear chimneys" really chimneys? What is the definition of chimney used here?**

Definition and explanation has already been provided in section 4.1.1. and 5.2.

See also authors online answers posted on 31st July 2018

**Is there evidence of breciation, or hydrofracturing in the regions interpreted as linear chimneys?**

See also authors online answers posted on 31st July 2018

**Actually, the illustration of the chimney features is not that great in the figures. And this brings me to the next concern. In the data section the authors mention that different stacks were produced grouping angle of incidence.**

In order to show the planform geometry of Linear Chimneys, we have mapped high amplitude reflections which cross the body of bright spot chimney columns, and have superimposed it on a dip map of PF. See new figure 9c in the revised manuscript.

See also Figure 5 which shows the 3D chimney bodies which are the 3D mapping of high amplitude anomaly columns on seismic sections.

See also authors online answers posted on 31st July 2018

**It is indicated that the seismic data presented in the paper is from the near offset stack. The seismic profiles shown in figures 3 and 4 are from which stack version?**

Suggestion accepted.

The following information has been added into section 2:

*"The seismic data in both surveys multi-channels, near-offset and has been post-stack time migrated and zero-phased."*

Figure 4 has been updated and became Figure 9.

See also authors online answers posted on 31st July 2018

**â ˘A´c The PHAAs are interpreted as carbonates. Why would they always be associated with depressions rather than mounds?**

See authors online answers posted on 31st July 2018

**â ˘A´c The figures are of high quality. However, even if they are over loaded with insets, some times they lack explanations of features that seem relevant for the interpretation. For example, panel b in figure 4 shows a white band braking through a high amplitude reflection. What is that feature? It is really hard to link all the different insets. In figure 5 I don't manage to identify where is the feature pointed with a yellow arrow on 5ai, on 5aiii. Is it correct that the seismic is NW to the right? It is important to find a way of simplifying these figures. Figure 5 could be split into 2 figures**

Suggestion has been taken into account, old Figure 5 has now became Figure 6 and 7. Figures have been simplified.

The following information has been added into the figure caption:

*"The linear depressions at issue are indicated by yellow arrows and locally interfere with regularly spaced furrows of likely sedimentary origin."*

Please also see authors online answers posted on 31st July 2018

**Please find below my notes while reading through the text; it includes a few typos. The L refers to the paragraph number and the P to the page number.**

**- L20/P3: minimum and maximum offsets? I assume these data sets are multi channel with long offsets? It is kind of important to provide this info before it is mentioned that the amplitudes vs. angle were used for verifying the seismic character of the observed features (a sort of undershooting?).**

Modified. Please see section 2.

**- L25/P3: typo: to map the linear. . .**

Modified.

**- L20/P4: figure 6b referenced before 3, 4 and 5? Check the flow of the figures. It is difficult to see from figure 2b what is stated here: that studies chimneys occur primarily in syncline areas. Maybe indicate the syncline structure in 2B and relate better 2a and 2b?**

Modified.

**- L25/P4 typo: relief instead of relied?**

Modified.

**- L10/P5 typo: check the unit used is it 10 to 100 s, ms or m?**

Modified.

- L5/P6: **The linear features shown in figure 4a-b are indeed strange features. Are these really along polygonal faults? (PFs?). Polygonal faults usually do not have a preferred orientation, but on the contrary, they consist of fault segments oriented covering the whole azimuth range (closing polygons), right?**

Information added in Introduction and section 3.2.3 in the revised manuscript.
See answer above.
See also authors online answers posted on 31st July 2018.

**- L20/P6: typo, 19% FORM**

Modified.

**- L30/P6: Consider making two different figures here. The figures have so many parts that it is actually a struggle to go through and understand everything. Is the statistical part of the figure really relevant? What do we do with the fact that 54% are intersecting the lower portion of PF? What seems most relevant in this paper is to get compared**

the orientation of the fluid flow related features with respect to the orientation local and regional faults and fractures, right?

Suggestion accepted, the figure has been split into two parts.

Information below has been added to section 5.2.3.:

"As supported by the statistical analysis presented herein, over 54\% of chimneys stem from the region around the lower PF footwall, therefore we infer that over 54\% of the time gas accumulated in the footwall, at the base of chimneys. It is also the same for the 19\% of chimneys that stem from the lower PF grabens (hanging wall). As a result, the statistic leads us to the interpretation that 73% of the total time gas preferentially accumulated in the lower part of PF blocks, therefore we investigate the cause of this phenomenon."

Please see 4.1.2. and 5.2.3.

See also authors online answers posted on 31st July 2018.

**- L10/P7: Check the use of tenses in this paragraph.**

Modified.

**- L20/P7: Is the evidence by Sonnenberg et al., 2016 related to the polygonal faults in this present study? In that case, it would help to see a sentence hinting what is the observation that works as evidence. I got this advice recently and I kind of see now the need for bringing into the current study the key observations rather than referring the reader too often to the previous studies. This degrades the flow of the reading and makes difficult to follow the paper.**

See also authors online answers posted on 31st July 2018

**- -L30/P7: do polygonal faults really reactivate? How is the accommodation of such movement if the fault planes can converge to each other rather than been parallel? Aren't these kind of faults associated with diagenetic processes and are hence a kind of one-time event?**

See also authors online answers posted on 31st July 2018

**- -L20/P8: 60 m deep and 4.5 m wide pockmark??? That is quite deep compare to the with, it is almost a conduit rather than a pockmark.**

Typo corrected.

**- -L30/P8: so there is active gas release at the seafloor at present? Or you mean active in the sense that there is gas filling the near-surface systems through the gas chimney structures?**

**- -L30/P8: this model is hard to digest here since there are so many faults and pre-existing weak planes that one would think that the fluids would find preferential pathways without much effort and hence gas chimneys would not be favor?**

Please see section 5.2.4.

**- L5/P9: I could not find figure 6b. In general, it is difficult to find in the figures some of the observations regarding gas chimneys. Again, the figures could be simplified by selecting only key examples.**

Suggestion accepted.
Figure modified.

**- L20/P9: The use of appendix figures to illustrate what seems to be the main conceptual model of the paper is not ideal. One figure in the main text should be enough to illustrate description of the model for fluid migration and development of chimneys. Figure 8a doesn't illustrate this, or did I miss something? When you mention PF tier here is it 1 or 2? It is very easy to get lost while reading, I think it is due to the fact that there are so many figures overloaded with details.**

Suggestion accepted.
Figure modified.

**- L25/9: typo: the PRESENCE of. . .And please revise this paragraph. These aspects are not necessarily ruling out each other and a combination of them could be a precondition for explaining your observations. Consider reformulating the paragraph.**

Suggestion accepted.

The aforementioned sentence in section 5.2.3. has now been rewritten as:

*"We suggest that two hypotheses in combination account for the mechanism of preferential gas accumulation in the lower PF footwalls of tilted blocks, horsts and lower hanging walls/grabens ((1) the presence of an impermeable regional seal, and (2) partial impermeable fault plane)), while two other hypotheses determine together the preferential gas migration to the lower PF footwall ((3) the differential strain in fault blocks, and (4) the stratigraphic position of permeable layers in fault blocks), and finally one for graben hanging wall ((5) the increase of local permeability)."*

**- L30/9: where do we see the gas accumulating at the foot wall? The foot wall of a major tectonic fault or are you referring to small faulted compartments resulting from the polygonal faulting? If so, is it really meaningful to talk about footwall if the blocks are somehow both footwalls and hanging walls with respect to each other?**

Definition and explanation added in section 3.2.3. in the revised manuscript.

**- L5-10/P10: Why would these areas be subjected to "relative" compressional strain? What do the authors mean here? Would it be more appropriate to say "less" compressional state rather than relative? In analogy with the particle motion maps for earthquakes (focal mechanisms) one would expect that the lower part of the hanging wall and the upper part of the footwall would experience more compression while dilation would dominate the upper hanging wall and the lower footwall (which is indeed consistent with Barnett et al., )**

Suggestion has been taken into account.
We have removed "relative".

**- L10/P10: again, no figure 6d and also the interpretation that the gas is accumulated in the footwall of polygonal faults is not easy to digest. A block can be considered a footwall or a hanging wall, depending on which fault plane is used as reference. I can see high amplitudes in both hanging and footwalls in figure 6c for example.**

Modified.
Please see definition and explanation added in section 3.2.3. in the revised manuscript.

**- L20/P11: Typo: BY. . .**
**- L20/P12: Again, the model of shear stress distribution through the four quadrants of the faulted blocks is very sounding for explaining the distribution of gas into more permeable zones. However, where these more permeable zones are entirely correlated with footwalls and hanging walls in these polygonal fault system is hard to assimilate. Is it really necessary to use this terminology? See main comment.**

Please see definition and explanation added in section 3.2.3. in the revised manuscript.
See also authors online answers posted on 31st July 2018

**- L25/P12: So the chimneys grow episodically? You foresee that the growth occurred in several episodes of reactivation of the system? It is important to describe this more explicitly in order to be able of comparing to systems from other margins with comparable settings.**

Please see also authors online answers posted on 31st July 2018

**- L5/P13: typo: may BE because..**

Modified.

- **L10/P14: typo: blocks**

Modified.

- **L25/P14: point 6 in the discussion. Check the grammar here. The sentence has a problem. When are the chimneys circular in isotropic stress fields? And when are they linear, isotropic and anisotropic? Clarify.**

Modified.

- **Figure 1: great figure. The use of a dip map to show all the elements of the study works extremely well, we can see the flanks of the salt domes and even the fine scale faults and fractures. However, isn't the present day bathymetry important to understand the stress regime?**

Please see also authors online answers posted on 31st July 2018

- **Figure 2b – typo: linear positive high amplitude. . .It is a bit difficult to read through the symbols of this figure.**

Suggestion accepted.

Figure modified.

- **Figure 5: Can you really tell that the high amplitude analogies in inset ii are gas accumulations, without any clear sign of polarity change?**

Figure modified.

- **Figure 6: Not convincing with the positive and negative bright spots interpretation. If you think it is key to differentiate between carbonates and gas for the discussion you may need to show a better indication for this, perhaps using wiggles and zooming into the anomalies?**

Figure modified as suggested.

- **Figure A7 typo: Gas MIGRATION into the hanging wall apex WAS likely because of the increase (check the sentence in any case)**

Modified.

**List of corrections and accepted suggestions by authors regarding the comments of Reviewer 2**

Majority of suggestions by Reviewer 2 are accepted.

**Summary of correction**

As requested by Reviewer 2, we have reordered all figures, as well as simplified and improved them to be easier to read. We have clarified the figure citations among the main text.

Our British co-author has checked through all of the English vocabulary in the main text and we have called for a native American English editor to perform proof reading to make sure there is no conflict between the British and American English writing standards.

**1. Main point:**

The original points of Reviewer 1 are in bold while the modifications made in the manuscript by authors are described below each point.

- **About English language**

We have applied all the English corrections given by the reviewers. Our native British co-author has gone through the entire script , we have also asked for English editing services of a native American English editor.

- **Figure reference in the main text**

We have improved all of the figure citations with updated text body.

- **Figure**

We have made improvements for figures by following most of the suggestions of all three reviewers.

**1.1. Figure modifications**

- Figures of appendix 2, 4, 5, 7 have been modified and moved into key figures as requested.

- All labels of figures have been checked and raw data is not covered by interpretation.

- The amplitude scale bar has been annotated in a clearer way.

- Seismic lines showing the three types of chimneys have been added as Figure 8.

- Seismic lines show the three different ways that chimneys intersect polygonal fault plan have been added as Figure 8.

- Images which show the three types of anisotropic PF arrays have been modified and are available as Appendix 2.

- Figure 2 has been updated, and no features of interest are not obscure.

- New Figure 5 has been added to show the 3D morphology of Linear Chimney's, which is issued from the 3D mapping of the high amplitude anomaly columns on seismic sections.

**1.2. Figure orders**

Figures have been reordered as requested by other reviewers. The new figure order is the following:

Fig. 1 has been updated

Fig. 2 has been updated

Appendix 5 became Fig. 3

Fig. 3 became Fig. 4

New Fig. 5

Fig. 5a became Fig. 6

Fig. 5b became Fig. 7

New Fig. 8

Fig. 4 became Fig. 9 and has been simplified

Fig. 6 became Fig. 10 and has been updated

Appendix 4 became Fig. 11

Fig. 7b became Fig. 12

Fig. 8 became Fig. 13

Appendix 7 became Fig. 14

Fig. 9 became Fig. 15

Fig. 10 became Fig. 16

Fig. 11 became Fig. 17

Fig. 12 became Fig. 18

Appendix 1

Appendix 3 became Appendix 2

Appendix 2 became Appendix 3 and has been simplified

Appendix 6 became Appendix 4

**1.3. Purpose of figures**

The list below is the purpose for the use of each figure:

Fig. 1 is a base map of a key seismic horizon in the study area showing the distribution of the relevant structural elements and fluid flow features.

Fig. 2 shows a regional cross section through the study area and seismic-stratigraphic framework illustrating where the different fluid features are situated,

Fig. 3 shows the stratigraphic relationship of polygonal fault tiers with one of the salt diapirs in the study area.

Fig. 4 shows a montage of maps and seismic sections which provide insight into the timing of polygonal faulting.

Fig. 5 illustrates geometry of linear chimneys using a 3D horizon map with geobodies of linear chimneys, and seismic sections.

Fig. 6 uses maps and sections to illustrate the complex, flame-like geometry of PHAAs at the upper termination of linear chimneys

Fig. 7 uses a map and a seismic section to illustrate the geometry of depressions at the top of linear chimneys shallow depressions.

Fig. 8 gives a summary of the three types of Linear Chimneys and their major intersection positions with PFs.

Fig. 9 uses maps and seismic sections to illustrate the relationship of linear chimneys with concentric, synclinal faulting and gas accumulations relative to an impermeable regional horizon.

Fig. 10 shows parallel and non-parallel chimney-fault relationships comprising parallel and non-parallel.

Fig. 11 shows examples where linear chimneys are parallel to second-order polygonal faults.

Fig. 12 shows the percentage abundance of different chimney-PF faults intersecting positions which provides an insight into the formation of chimneys and gas migration history and pathways.

Fig. 13 Images of Syncline-0 show exceptional occurrence of Linear Chimneys in the interval devoid of PFs. Linear Chimneys with the scale of a kilometer in lateral length occur in parallel with tectonic syncline-related faults (For demonstrating the influence of tectonic stresses on the propagation direction of hydraulic fractures).

Fig. 14 shows hypothesis for the location of gas accumulations before the nucleation of chimneys, as well as the hypothesis of what is the cause of gas accumulation in lower grabens.

Fig. 15 shows the hypothesis for the preferential location of gas accumulation in the lower PF tiers and footwalls.

Fig. 16 Demonstrates a conceptual model for the formation of Linear Chimneys.

Fig. 17 Senarios show formations of Linear Chimneys during PF's development in Plio-Quaternary.

Fig. 18 shows different styles of fluid migration in two different geological contexts, in polygonal faulted sediment, and  tectonic faulted interval without polygonal faults.

Appendix 1 Relates to the seismic surveys used in this study, as well as the location of zones which have been previously studied and published.

Appendix 2 Shows different types of anisotropic PF patterns within the study area.

Appendix 3 Shows seismic lines and maps for each group of Linear Chimneys and their statistics as well as the percentage of various intersecting positions which occur within the PF's.

Appendix 4 Shows fault traps which induce gas accumulations at the base of the PF tier as well as the bottom of the PF tier with respect to the geometry of PFs cells.

**2. Other points**

All the comments below have been accepted and requested modifications have been made:

P1 L2: "Angola. These features are termed "Linear Chimneys"."

P1 L3: "Hydrocarbon migration"

P1 L4: Remove "the" (second word in line)

P1 L7: Replace "e.g." with "such as"

P1 L11: "The initiation of polygonal faulting occurred 40 to 80 m"

P1 L12 "The majority of Linear Chimneys nucleated in the lower part of the PF tier

below an impermeable layer within the tier. The filling of lower parts of the polygonal

fault tier demonstrate the presence of pore space within the lower part of the tier. The

PF gas traps restrict the leak points. . ." NOTE it is possible to have porosity / gas

without significant permeability

P1 L17 ". . .polygonal faults coupled with. . ."

P2 L3 "flow directions in the subsurface and the distribution. . ."

P2 L5 ". . .structures formed during fluid leakage records the style. . ."

P2 L8: Replace "leakage" with "leak"

P2 L23: ". . . documented chimneys having elliptical cross-section and described the planform ratio of these chimneys for the first time."

P2 L25 replace "orientations" with "orientation"

P2 L26: Reference in brackets

P2 L28 ". . . align parallel to these"

P2 L32 "However, neither factors that determined the linear planform of these chimneys nor the reason why gas charged fluid migrated into the PF tier have been investigated."

P3 L1 "It has been documented that"

P3 L5: "interactions between the orientation of magna fluid conduits and tectonic stresses. Nakamura established"

P3 L6: ". . .different tectonic regimes, noting, for instance, that aligned. . ."

P3 L11: Delete "Based on seismic observations"

P3 L12: "in shallow buried sediments based on seismic observations, thereby"

P3 L18-19: Statement not supported by fig. 1

P3 L19 "The seismic data has a"

P3 L21 "The dominant frequency is slightly"

P3 L24: Please describe which angles are covered by the near, mid and full stack

P3 L26: "rule out whether the studied features are"

P3 L29: ". . . to map the linear fluid venting structures as accurately as possible"

P3 L30 "are present on the near, mid and far offset volumes"

P4 L4: Fig.ures cited do not support text

P4 L8: Add reference

P4 L14: Which figure in Ho 13?

P4 L20 Ho et al 2012a, fig 6a in this paper OR Fig. 6a in Ho et al 2012a?

P4 L26 Replace "Relied" with "Relief"

P4 L27: Replace "Isopach" with "Constant"

P4 L 28: "Below which a large number of gas accumulations are interpreted" (also

explain why these are interpreted to be gas accumulations)

P5 L2: Spell out that PHAAs are acoustically hard (increase in acoustic impedance)

and NHAs are acoustically soft (decrease in acoustic impedance). Use PHAA and

NHAA or PHA and NHA, don't use what you currently have (PHAA and NHA). Section 4.1.1 I struggled to find the observations in the text in the figures - some figure

references may be wrong. See my separate comments on how to tidy up the figures

to make things easier for the reader Section 4.1.2 I struggled to find the observations

in the text in the figures - some figure references may be wrong. See my separate

comments on how to tidy up the figures to make things easier for the reader

P7 L30: Do not see the described feature on the referenced figure

P8 L2: Same comment as P7 L30

P8 L19: Sentence does not make sense to me

P8 L30: Replace "Some might hypothesize" with "It could be argued" Section 5.2.2 Explain why strong soft anomalies are interpreted as gas. Has the seismic been balanced

correctly?

P9 L28: "1) the presence of an"

P9 L33: Fig.ure reference incorrect

P10 L14: Fig. 6D does not exist

P12 L23: Sentence does not make sense

P13 L8: ". . .. Study area may be because" The notes which follow on the figures are some observations, these may or may not be relevant given my recommendation to reorder and rearrange the figures.

Figure 1: A multi-segment seismic line showing the labelled geometries would be helpful

Figure 2: Features of interest on the seismic line obscured by illustrations

Figure 3: "A few faults propagate above the Tiers-2 (interval A)": This is confusing, interval A is above Tier 1, not Tier 2

Figure 4: Please center amplitude maps on 0. It is difficult to tell which parts of the amplitude maps are negative or positive at the moment.

Figure 5: (A) (i)/(ii) and (iii) should be swapped around (show seismic line first, and then maps of given horizon). Likewise for figure 5B

Figure 6: Labelling of maps is confusing, please label the horizon used for figure 6b on Figure 6c. Amplitude limits of inset to figure 6a are not labelled.

Figure 7: Figure caption does not match figure, check this.

Figure 8: Cannot see location of seismic line on figure 8b. Amplitude map should be centred on 0.

Figure 9: "Low permeable layer" should be "low permeability layer"

Figure 10: Poor grammar in figure caption, suggest rewriting. Not sure what middle block diagram between block diagram to left and seismic line to right adds in a, b and c?

Figure 11: I like this figure, but no caption for figure 11d.

Figure 12: I like this

Figure A1: It is good to have a summary map like this, however I cannot see the grey C8area this study is based on.

Figure A2: This seems to be a key figure, not sure what it is doing in appendices. Ensure amplitude maps are centred on 0 so it is clear what is anomalous.

Figure A3: Would a reference to Ho 2013 suffice instead of reproducing this here?

Figure A4: This seems to be a key figure, not sure why it is in appendices. Ensure amplitudes centred on 0

Figure A5: Where is "appendix A"? Do you mean "Appendix 1"? This seems to be a key figure, why is it in appendices?

Figure A6: The amplitude map sin a(ii) shows PF cells filled by amplitude anomaly which IS INTERPRETED to represent gas fill. Ensure amplitude maps centred on 0.

Figure A7: I like this, is this a key figure?

**List of corrections and accepted suggestions by authors regarding the comments of Reviewer 3**

All suggestions and questions by Reviewer 3 are accepted and answered.

**Summary of correction**

As by requests, we have added in clarification for the conceptual model of fluid migration in polygonal fault tier in section 5.2.3.

A hypothesis for the formation of a sub group of Linear Chimneys has been added in section 5.2.4.

A brief introduction about PF and their usage as palaeo stress indicator has been added in Introduction and section 3.2.3.

Figures have been reordered and updated as requested by the reviewers.

A figure for the classification of different types of Linear Chimneys has been added as Figure 8.

**Main points:**

The original points of Reviewer 1 are in bold while the modifications made in the manuscript by authors are described below each point.

**Regarding the request of adding a brief introduction for polygonal faults**

We have accepted the suggestion and the following information has been added into the Introduction section:

*"Polygonal faults are considered as non-tectonic fault systems arising due to compactional-dewatering of very fine-grained sediments during the early stages of burial in passively subsiding sedimentary basins (Henriet et al., 1998). In the classic examples of these fault systems which show "polygonal" fault arrangements and also contribute to their nomenclature,*

*they were characterized by very small differences between the horizontal principle stresses during their formation (Cartwright, 1994; Carruthers et al., 2013). The examples of polygonal faults in this case study show substantial departures from this classic "polygonal" fault patterns (so called isotropic PFs) to very polarized fault arrangements (so called anisotropic PFs) where the tier is deformed by salt tectonic structures or offset by their associated fault systems (Fig. 1; Carruthers, 2012). These faults can display a variety of intricate patterns ranging from tight radial systems around salt diapir to concentric systems within salt withdrawal basins and spiraling concentric patterns above buried pockmarks (Stewart, 2006; Ho et al., 2013). The preferentially aligned faults are many times longer than the regular faults segments with polygonal alignments but are often still confined to the same "tiers". The observations are consistent with a number of other reported examples of preferred fault alignments within networks of polygonal faults (Stewart, 2006; Ghalayini et al., 2016). The preferred fault alignments are indicative of horizontal stress anisotropy at the time of their formation (Carruthers et al., 2013)."*

**Regarding the explanation of use of polygonal faults as palaeo stress indicators**

The following information has been added into section 3.2.3.:

*"Preferred fault alignments or "anisotropic fault patterns" within polygonal fault networks have been observed in this study area. Concentric faults surround pockmarks (see fig. 2a in Ho et al., 2013) and are parallel to extensional synclinal faults (red dotted lines in Fig. S3 c; Appendix PF patterns b). Radial faults occur around salt diapirs (Appendix PF patterns c) (Carruthers, 2012) whilst ladder-like fault patterns occur in the center of concentric fault patterns above Syncline-2 (Fig. 6a-b, Appendix 3d).*
*The orientations of the PFs around or above the aforementioned tectonic structures are not unusual as the fault patterns mantle the expected stress state of the structures (Carruthers, 2012; Carruthers et al, 2013). The direction of maximum horizontal stress around the tectonic structures is indicated by the first-order anisotropic PFs while the horizontal minimum stress is indicated by the second-order anisotropic PFs (e.g. stress ellipses in Fig. 1), and hence different*

*PF patterns are considered as indicators of stress state in the host sediments (Carruthers, 2012; Carruthers et al, 2013).* "

**In the proposed conceptual model, the authors have suggested that gas could not mi-grate further upward the PF plane, the reason for that could simply be the regional seal retains the gas in the lower part of PF tier but has nothing to do with the permeability of PFs? Could it be purely some lithological effects, such as permeable layers occur rather in the lower part of the PF tier and layers in the upper tier layers are less permeable or impermeable?**

Information below has been added in section 5.2.3. of the revised manuscript:

*"It can be argued that, more permeable deposits preferentially occur in the lower PF tier and lead to preferential occurrence of gas accumulation in such place. This possibility is disregarded because of the indifference between the lithologies in the upper and lower part of the PF tier as indicated by Total's internal well reports, regardless the permeability measurement of the host sediments is unavailable."*

See also authors online answers posted on 31[st] July 2018

**In the statistics there are 7% of Linear Chimneys which are not intersecting with any PFs and occur in the middle of the tilted PF blocks. Is the conceptual model of impermeable faults intersecting impermeable layer to form fault bound trap still work for these chimneys?**

Answer to this question has been added in section 5.2.4. of the revised manuscript:

*"For chimneys that do not intersect with any fault i.e. occur in the middle of PF fault blocks, the illustrated model by Løseth et al. (2009) can be used as a referential analogue (see fig. 21 in Løseth et al., 2009); a lateral contact point between the edge of the gas accumulation and the*

*upper limit of the tilted storage-layer, in the middle of the tilted block, formed a hydrocarbon spill point from where gas chimney nucleated and propagated upward (Løseth et al., 2009). This type of spill point is commonly occurred in structural traps.*"

**Figure modifications**

- Figures have been simplified as requested.

- As requested classification for the three types of chimneys has been added as Figure 8.

- As requested the old Figure 7a has now been integrated with the classification of the three types of chimneys as a whole Figure 8.

- Figures of appendix 2, 4, 5, 7 have been modified and moved into key figures as requested.

- Figures have been reordered as requested by other reviewers. The new figure orders are the following:

  Fig. 1 has been updated
  Fig. 2 has been updated
  Appendix 5 became Fig. 3
  Fig. 3 became Fig. 4
  New Fig. 5
  Fig. 5a became Fig. 6
  Fig. 5b became Fig. 7
  New Fig. 8
  Fig. 4 became Fig. 9 and has been simplified
  Fig. 6 became Fig. 10 and has been updated
  Appendix 4 became Fig. 11

Fig. 7b became Fig. 12

Fig. 8 became Fig. 13

Appendix 7 became Fig. 14

Fig. 9 became Fig. 15

Fig. 10 became Fig. 16

Fig. 11 became Fig. 17

Fig. 12 became Fig. 18

Appendix 1

Appendix 3 became Appendix 2

Appendix 2 became Appendix 3 has been simplified

Appendix 6 became Appendix 4

**Purpose of figures**

The list below is the purpose for the use of each figure:

Fig. 1 is a base map of a key seismic horizon in the study area showing the distribution of the relevant structural elements and fluid flow features.

Fig. 2 shows a regional cross section through the study area and seismic-stratigraphic framework illustrating where the different fluid features are situated.

Fig. 3 shows the stratigraphic relationship of polygonal fault tiers with one of the salt diapirs in the study area.

Fig. 4 shows a montage of maps and seismic sections which provide insight into the timing of polygonal faulting.

Fig. 5 illustrates geometry of linear chimneys using a 3D horizon map with geobodies of linear chimneys, and seismic sections.

Fig. 6 uses maps and sections to illustrate the complex, flame-like geometry of PHAAs at the upper termination of linear chimneys

Fig. 7 uses a map and a seismic section to illustrate the geometry of depressions at the top of linear chimneys shallow depressions.

Fig. 8 gives a summary of the three types of Linear Chimneys and their major intersection positions with PFs.

Fig. 9 uses maps and seismic sections to illustrate the relationship of linear chimneys with concentric, synclinal faulting and gas accumulations relative to an impermeable regional horizon.

Fig. 10 shows parallel and non-parallel chimney-fault relationships comprising parallel and non-parallel.

Fig. 11 shows examples where linear chimneys are parallel to second-order polygonal faults.

Fig.. 12 shows the percentage abundance of different chimney-PF faults intersecting positions which provides an insight into the formation of chimneys and gas migration history and pathways.

Fig. 13 Images of Syncline-0 show exceptional occurrence of Linear Chimneys in the interval devoid of PFs. Linear Chimneys with the scale of a kilometer in lateral length occur in parallel with tectonic syncline-related faults (demonstrating the influence of tectonic stresses on the propagation direction of hydraulic fractures).

Fig. 14 shows hypothesis for the location of gas accumulations before the nucleation of chimneys, as well as the hypothesis of what is the cause of gas accumulation in lower grabens.

Fig. 15 shows the hypothesis for the preferential location of gas accumulation in the lower PF tiers and footwalls.

Fig. 16 Demonstrates a conceptual model for the formation of Linear Chimneys.

Fig. 17 Senarios show formations of Linear Chimneys during PF's development in Plio-Quaternary.

Fig. 18 shows different styles of fluid migration in two different geological contexts, in polygonal faulted sediment, and tectonic faulted interval without polygonal faults.

Appendix 1 Relates to the seismic surveys used in this study, as well as the location of zones which have been previously studied and published.

Appendix 2 Shows different types of anisotropic PF patterns within the study area.

Appendix 3 Shows seismic lines and maps for each group of Linear Chimneys and their statistics as well as the percentage of various intersecting positions occuring within the PF's.

Appendix 4 Shows fault traps which induce gas accumulations at the base of the PF tier as well as the bottom of the PF tier with respect to the geometry of PFs cells.

[revised manuscript text omitted]

---

## Referee Report (RR1)

Formation of linear planform chimneys controlled by preferential

hydrocarbon leakage and anisotropic stresses in faulted fine-grained

sediments, Offshore Angola

Ho et al.,

The paper is significantly easier to follow. I must admit that it is still a bit difficult to cope with all the figures included in the manuscript. Nevertheless, I reckon that the features documented are so amazing that is difficult to cut the number of figures. In this version the information provided in the figures is more synthesized and does a better job of illustrating what is described in the text.

I could identify a few typos and a few parts that would benefit from clarifying the ideas. Please find below the summary of minor points that I think would improve the manuscript even more (specific annotations can be found in the PFD file as comments):

-        Minor typos identified

-        It is said that near-offset, middle offset and far offset stacks were used to investigate the origin of the chimneys. It is not clear whether the interpretation of chimneys has been done indistinctively over the three type of stacks. The chimneys look different in a near-offset vs. a far offset stack. It would help if the type of stack is indicated in the figure caption of figures showing seismic sections with chimneys. Otherwise, mentioning that several offset stacks were used makes no sense.

-        Shearing of the basal part of footwall – if compaction is an issue then wouldn't it be so for the entire hanging and footwall? Can this be clarified?

-        What if the gas was already distributed along the reservoir layer before polygonal faulting? Then there would be gas available for generating chimneys that originate at the hanging and foot walls indistinctively. Isn't this a plausible scenario?

-        Section 5.2.5 – This section is still hard to follow. It is not clear whether the authors propose that 1) generally the regional stress field controls the orientation of chimneys while there are exceptions where a local modification of the stress field becomes dominant and controls the orientation of certain chimneys; or 2) whether an interaction of both regional and local stress patterns is always a requirement to trigger the development of chimneys. I think all the info is there but it is just hard to follow up. My feeling is that a little rewording and restructuring of the ideas would be enough to improve this section. It comes clearer in the abstract.

-        Terminology "in-situ stress" to refer to local stress – is the use of in situ here correct? If we go to the field and measure stress (in-situ) wouldn't we measure a stress quantity that is the summation of different sources of stress (regional + local)? I tend to think that referring to "local" stress fields when describing the stress field dominated by the small scale faults and pore-fluid pressure interactions, is more appropriate.

-        Section 5.3.2 is very interesting however, it is still not entirely clear why the authors argue for a shift in the orientation of the stress field. Different stress fields may characterize the north and the

east of the salt feature at a contemporaneous period. Why the observation of chimneys toward the astern edge is used as evidence of a shift in the stress field with time? Can this be clarified?

-        The conclusion would benefit from avoiding repeating the details of chimney development as presented already in section 5.2.6

---

## Referee Report (RR2)

**Figure orders**

In the text timing of PF is discussed after chimney observations are presented. Therefore Fig. 4 (which deals with timing of PF) should be presented after the present Fig. 13.

**Page 2**

L13: Replace "observed" with "described"

L21: Replace "(Hovland, 1983)" with "Hovland (1983)"

L23: Remove "are"

**Page 3**

L8-11: Revise grammar, too many brackets. Consider "They were later called polygonal fault (PF) systems by Cartwright (1994). These faults were further described by Clausen et al. (1999) and Goulty (2008), although other observations show that these faults can host a whole range of different plan form geometries including concentric patterns (cf. Stewart , 2006, Chopra and Marfurt, 2007).

L27-29: Revise grammar. Consider "Particularly the following questions are addressed; (a) why are chimneys linear in planform and not circular or elliptical as observed elsewhere? (b) why do they occur specifically along certain parts of PF planes?"

**Page 4**

L16: replace "basins" with "basin's"

L23: remove the comma

**Page 5**

L24: replace "reorganizes" with "reorganising"

L29: replace "(Ho et al., 2013, see fig. 2a in)" with (see fig. 2a in Ho et al., 2013)

L30-31: replace "(Appendix 2c) (Carruthers, 2012)" with "(Appendix 2c and Carruthers, 2012)

**Page 6**

L1 and 4: check this "(Carruthers, 2012, **?;** Carruthers et al., 2013)"

**Page 7**

L18-19: replace "(Hustoft et al., 2007, 2010)" with "Hustoft et al., (2007, 2010)"

**Page 8**

L30: replace "fault's" with "fault"

**Page 9**

L2: missing full stop

L3-8: replace with "Geometries of PFs in the Neogene-Quaternary deposits of Lake Superior, Hatton Basin and Vøring Basin indicate that their growth is very recent and could occur to the present day seafloor (Berkson et al., 1973; Jacobs, 2006; Berndt et al., 2012; Laurent et al., 2012). Recently Sonnenberg et al., (2016) confirmed that PFs grew close to the seafloor with evidence of fault scarps filled by onlapping syn-sedimentary strata."

L9 replace "new evidence has shown to support" with "new evidence supports"

L9-10 delete "A previous study"

L19: Replace "15ms" with "15 ms". Check the rest of the text for errors of this type.

L32: delete space after "carbonates"

**Page 10**

L34:  replace "can not" with "cannot"

**Page 11**

L11: there is something odd about this line, check it

L17: why does line start "2)"?

L26: why does line start "3)"? This appears to be a list of paragraphs, but I do not see 1), unless this is in L11?

**Page 12**

L6: replace "19%" with "19 %". Check the rest of this document for errors of this type.

L22: check this: "In literature, numerical models of **?** shows that fluid pressure"

**Page 15**

L27: remove the comma

**Page 16**

L2: remove the comma

**Fig 9c**

If this is a map of a NHAA, why is amplitude scale on map only positive?

**Fig 16**

In caption replace "right column)" and "left column)" with "right column:-" and "left column:-" respectively

---

## Referee Report (RR3)

**General comments for the revised version of the manuscript "Formation of linear planform chimneys controlled by preferential hydrocarbon leakage and anisotropic stresses in faulted fine-grained sediments, Offshore Angola"**

The overall manuscript is greatly improved and now meets the publication requirements, aside a very few minor adjustments. It can be seen that the authors have put in great amount of effort. The figures are much clearer now, especially the 3D illustration for the morphology of the Linear Chimney and the figure of classification. Each section has a much better flow for reading. The introduction is very well written as well. The section of geological setting has been greatly improved. The discussion sections are much more focused and detailed, it may be a bit lengthy which may be in part do to adding in the extra information as requested by reviewers (including myself), but since the manuscript is addressing a relatively new and very technical topic, it is reasonable and perfectly acceptable.

There are just some minor mistakes left in the main text, some sentences are repetitive (please see the list below), and there are two places that need clarification: please specify that polygonal faults were impermeable "at the time" when gas migration happened and when chimneys formed.

Generally speaking the manuscript is ready for publication. After applying the minor corrections in the list below the article can be published at its present state.

P2/L12 add reference
P4/L2 typo "being"
P4/L2 typo "an to"
P4/L19 redundancy remove "salt"
P9/L31 and P10/L8 repetition
L10/L28 Remove "In contrast, residual gas is still present in the reservoir of Type 2 chimneys" put in L24 and change it to "….at their downward termination (Fig. 10c) which are interpreted as residual gas accumulation"
P11/L12 what is 731) ?? something missing?
P11 please put in subtitle for each point
P11/L20 please clarify that the PF planes were impermeable "at the time" when gas migrated

P12/L8 redundancy, remove "where gas is likely to have accumulated before exceeding the lithostatic pressure."

P12/L13 redundancy, remove "In the upper parts of a graben, extensional phenomena dominate, while"

P12/L17 remove "Based on seismic observations and the hypothesis of gas migration into the specific part of a PF interval, as established above"

P12/L20 please clarify that the PF planes were impermeable "at the time" when chimneys formed

P13/L18 change sigma-2 to intermediate stress, please be consistent with the term

P14/L4 is a bit contradicting with L8-10. Please change L8-10 into something like "…the planform and orientation of chimneys can be affected simply by the stress field of a "neighboring" tectonic fault when PFs are absent."

P15-P16 in 5.3.2 please put in subtitle for the two sub sections, such as: 1) palaeo stress indicated by linear chimneys and 2) palaeo stress indicated by linear PHAAs

P16-L13 please move point 4 before 1

P16-L20 remove "strongly"

Figure 3 where is SE?

Figure 4 redundancy - remove "(interval A)"

Figure 5 excellent figure! Which reflection on section (b) and (c) the map (a) corresponds to?

Figure 8 excellent figure! May consider to remove the grey lines between blocks in the right column

Figure 10c please label which is peak or trough for the seismic traces

---

## Author Response (AR2)

**Author's answers to Reviewer's comments on "Formation of linear planform chimneys controlled by preferential hydrocarbon leakage and anisotropic stresses in faulted fine-grained sediments, Offshore Angola" – Ho et al. (2018)**

We would like to thank the three reviewers for their precious time, effort and helpful comments given to this manuscript.

**- Answer to Reviewer 3**

We thank Reviewer 3 for the very important point that has risen for the correction.

Now we have specified that it was at the time when fluid leakage happened the impermeability occurred in the polygonal faults. We have added "at least during the gas migrations" and "at the moment when chimneys formed" in P11 L28 and P12 L29.

We have also put in short title for every bullet point in section 5.2.3.

We have removed the repetitive sentences and corrected all the typos.

**- Answer to Reviewer 2**

We thank Reviewer 2 for the impressive precision in listing the typo corrections.

We have corrected all of the typos, and we have requested our English editor to check through the corrections again.

We keep Figure 4 in this same figure order because it is a part of the stratigraphy description, although it has been discussed later in the discussion section. We do not feel it is necessary to discuss about the stratigraphic element without making the description upfront.

**- Answer to Reviewer 1**

We would like to thank Reviewer 1 Plaza-Faverola for her precious time spent on reviewing our manuscript and her useful comments.

All typos have now been corrected in the manuscript.
Points 2, 3, 4, 6 have been clarified in the manuscript.
Points 1, 5 have already been clarified previously.

Author's answers (green text) to Reviewer 1's questions (in black) are below:

- It is said that near-offset, middle offset and far offset stacks were used to investigate the origin of the chimneys. It is not clear whether the interpretation of chimneys has been done indistinctively over the three type of stacks. The chimneys look different in a near-offset vs. a far offset stack. It would help if the type of stack is indicated in the figure caption of figures showing seismic sections with chimneys. Otherwise, mentioning that several offset stacks were used makes no sense.

As indicated in Methodology in the first and second version of manuscript (P3 L26-27; P3 L32 & P2 L4), the near offset survey was used for interpretations and illustrations in this study.

The middle and far-offset volumes were only used for verifying the authenticity of studied chimneys to make sure that they are not artefacts, as explained in the previous manuscript (P3 L25).

- Shearing of the basal part of footwall – if compaction is an issue then wouldn't it be so for the entire hanging and footwall? Can this be clarified?

The above model (by Barnett et al., 1987) is for unlithified shallow sediments only, it demonstrates that the different parts of footwall/hanging wall do not compress in the same way when faulting happens in shallow depths. We applied this model because Tier-2 was buried only a few tens of meters when PFs formed.

We have now clarified with key words "…in shallow buried depths" and "…for uncompacted shallow buried sediments" in Point C of section 5.2.3.

- What if the gas was already distributed along the reservoir layer before polygonal faulting? Then there would be gas available for generating chimneys that originate at the hanging and foot walls indistinctively. Isn't this a plausible scenario?

The actual polygonal fault tier and its overlaying sediment are not very thick, so before polygonal faulting and before the deposition of the fault tier, the sediment column above the reservoir layer was even thinner (<100m) and it was not likely able to generate enough vertical pressure to retain overpressured gas.

This is now clarified in section 5.2.2.

- Section 5.2.5 – This section is still hard to follow. It is not clear whether the authors propose that 1) generally the regional stress field controls the orientation of chimneys while there are exceptions where a local modification of the stress field becomes dominant and controls the orientation of certain chimneys; or 2) whether an interaction of both regional and local stress patterns is always a requirement to trigger the development of chimneys. I think all the info is there but it is just hard to follow up. My feeling is that a little rewording and restructuring of the ideas would be enough to improve this section. It comes clearer in the abstract.

We have improved the wording into: "...*at tier-fault scale, the in-situ anisotropic stress of the nearest PFs has major influence on the orientation of Linear Chimneys than the local tectonic fault stress field. Nevertheless, as the majority of Linear Chimneys are aligned parallel to both tectonic and polygonal faults, as Linear Chimneys do not occur in areas where isotropic PFs are solely present, therefore the combination of both anisotropic stress fields of tectonic and polygonal faults is suggested to be the main cause of linear planform chimneys with preferential orientations.*"

- Terminology "in-situ stress" to refer to local stress – is the use of in situ here correct? If we go to the field and measure stress (in-situ) wouldn't we measure a stress quantity that is the summation of different sources of stress (regional + local)? I tend to think that referring to "local" stress fields when describing the stress field dominated by the small scale faults and pore-fluid pressure interactions, is more appropriate.

As the definition has already been provided in the previous correction, in section 3.2.3 we distinguish "regional", "local" and "in-situ" stress fields.

The variations of polygonal fault patterns from isotropic to anisotropic indicate that the regional stress state (offshore Angola) is different from the local one (surrounding salt structures), and that the orientation of some Linear Chimneys does not follow the horizontal direction of local tectonic stress but the one of less-anisotropic polygonal faults indicate that stress conditions at the location of individual polygonal faults (at sites of incipient hydraulic fractures), is different. Fault formations can induce a stress field around them which has orientations different from the local and regional ones (Kattenhorn et al., 2000; Hu, 1995). Therefore we consider that the use of "in-situ" is appropriate.

- Section 5.3.2 is very interesting however, it is still not entirely clear why the authors argue for a shift in the orientation of the stress field. Different stress fields may characterize the north and the east of the salt feature at a contemporaneous period. Why the observation of chimneys toward the astern edge is used as evidence of a shift in the stress field with time? Can this be clarified?

This has now been clarified in the main text.

Knowing that both anisotropic long polygonal faults (PFs) and Linear Chimneys have to form along the intermediate principal stress direction, as the PFs formed before the Linear Chimneys, and as these PFs are parallel to the northern side of syncline while the Linear Chimneys are paralleled to the eastern side; we conclude that the intermediate stress direction probably altered from North to East during the time between the formation of both features.

- The conclusion would benefit from avoiding repeating the details of chimney development as presented already in section 5.2.6

We thank the reviewer for this suggestion. We have only provided 9 main points as the core conclusions. We believe that these are not excessive and are necessary to summarise for "hurry reading" readers.

**References**

Hu, J.C., 1995. Numerical modeling and regional tectonic analyse : the case of Taiwan; Modélisation numérique et analyse tectonique régionale: le cas de Taiwan (Doctoral dissertation, Paris 6).

[revised manuscript text omitted]